# Are Two Datasets Close Enough With Statistical Significance?
# A Kernel Distributional Closeness Testing Approach

**Zhijian Zhou** [1]  **Liuhua Peng** [1]  **Xunye Tian** [1]  **Mingming Gong** [1]  **Feng Liu** [1]

## Abstract

Are two distributions close to each other with statistical significance? *Distribution closeness testing* (DCT) formalizes this question by testing whether the distance between a distribution pair is at least $\epsilon$-far. Existing DCT methods mainly measure discrepancies between a distribution pair defined on discrete spaces (e.g., using total variation), which limits their applications to complex data (e.g., images). To extend DCT to more types of data, a natural idea is to introduce *maximum mean discrepancy* (MMD), a powerful measurement of the distributional discrepancy between two complex distributions, into DCT scenarios. However, the empirical results indicate that many distribution pairs can have the same MMD value despite having different norms in the same *reproducing kernel Hilbert space* (RKHS), and these pairs may exhibit different finite-sample distinguishability and reflect different practical closeness levels, making MMD less informative in DCT. To mitigate the issue, we design a new measurement of distributional discrepancy, *norm-adaptive MMD* (NAMMD), which scales MMD's value using the RKHS norms of distributions. Based on the asymptotic distribution of NAMMD, we finally propose the NAMMD-based DCT to assess the closeness level of a distribution pair. Theoretically, we prove that NAMMD-based DCT has higher test power compared to MMD-based DCT, with bounded type-I error, which is also validated by extensive experiments on many types of data (e.g., synthetic noise, real images). Our code is available at: https://github.com/zhijianzhouml/NAMMD.

[1]The University of Melbourne. Correspondence to: Feng Liu <fengliu.ml@gmail.com>.

*Proceedings of the 43rd International Conference on Machine Learning*, Seoul, South Korea. PMLR 306, 2026. Copyright 2026 by the author(s).

## 1. Introduction

Distribution shift between training and test sets often exists in many real-world scenarios where machine learning methods are used (Dietterich, 1998; Wang et al., 2019). According to the classical machine learning theory (Quiñonero-Candela et al., 2022), it is well-known that such a shift will influence the performance on the test set. In a worst case: having a very large distributional discrepancy between training and test data, we might have poor performance on test data for a model trained on the training data (Ben-David et al., 2010; Rabanser et al., 2019). The obtained poor performance can be explained by many theoretical results (Ben-David et al., 2010; Fang et al., 2022). However, we can also observe the other interesting phenomenon: it is also empirically proved that models trained on a large dataset (e.g., ImageNet (Deng et al., 2009)) can have good performance on relevant/similar test data (e.g., Pascal VOC (Hoiem et al., 2009)) that is different from training dataset (Oquab et al., 2014). This means that, even if training and test data are from different distributions, we can still expect good performance as they might be close to each other.

Therefore, *seeing to what statistically significant extent* two distributions are close to each other is important and might help us decide if we really need to adapt a model when we observe upcoming data that follow a different distribution from training data. Two-sample testing (TST) can naturally see if training and test data are from the same distribution (Gretton et al., 2006), but it is less useful in the phenomenon above as we might also have good performance when the training and test data are close to each other. Fortunately, in theoretical computer science, researchers have proposed *distribution closeness testing* (DCT) to see if the distance between a distribution pair is at least $\epsilon$-far, including TST as a specific case with $\epsilon = 0$ (Li, 1996; Acharya et al., 2012; Levi et al., 2013; Canonne, 2020). The DCT fits the aim of *seeing to what statistically significant extent* two distributions are close to each other, and has been used to evaluate Markov chain mixing time (Batu et al., 2000), test language membership (Bathie & Starikovskaya, 2021) and analyze feature combinations (Mehrabi & Rossi, 2023).

However, existing DCT methods mainly measure closeness using total variation (Batu et al., 2013; Bhattacharya

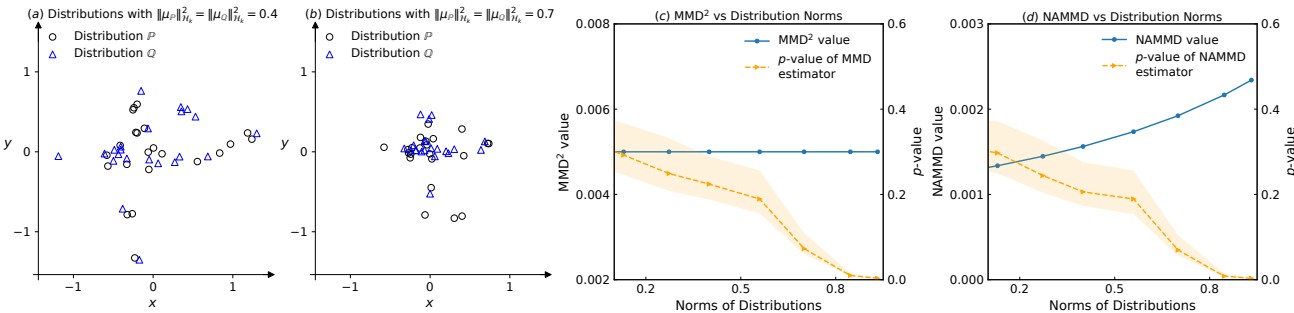

**Figure 1.** All visualizations are based on 30-dimensional Gaussian mixture distributions with 30 mixture components. The population MMD value is fixed as $\|\boldsymbol{\mu}_{\mathbb{P}} - \boldsymbol{\mu}_{\mathbb{Q}}\|_{\mathcal{H}_\kappa}^2 = 0.005$ under the Gaussian kernel with bandwidth 1. This experiment fixes the population MMD value, varies the RKHS norms of the distribution embeddings, and uses TST with sample size 50 to examine finite-sample distinguishability. Subfigures (a) and (b) show two-dimensional projections of distributions $\mathbb{P}$ and $\mathbb{Q}$ with different RKHS norms ($\|\boldsymbol{\mu}_{\mathbb{P}}\|_{\mathcal{H}_\kappa}^2$ and $\|\boldsymbol{\mu}_{\mathbb{Q}}\|_{\mathcal{H}_\kappa}^2$), while yielding the same MMD value. Subfigure (c) presents the MMD value and the $p$-values of its estimator in TST. Subfigure (d) presents the NAMMD value and the $p$-values of its estimator in TST. NAMMD exhibits a stronger empirical correlation with the $p$-value than MMD: *larger NAMMD corresponds to smaller $p$-value*, while MMD *keeps the same value* as the $p$-value changes. Although this example uses a Gaussian kernel, the same phenomenon can be observed for the bounded, shift-invariant kernels considered in this paper, namely kernels of the form $\kappa(\boldsymbol{x}, \boldsymbol{x}') = \Psi(\boldsymbol{x} - \boldsymbol{x}') \leq K$, where $K > 0$, $\Psi(\cdot)$ is positive-definite, and $\Psi(\boldsymbol{0}) = K$; see Appendix F for kernel limitations and Appendix D.1.1 for experimental details.

& Valiant, 2015; Acharya et al., 2015; Diakonikolas et al., 2015), and primarily focus on the theoretical analysis of the sample complexity of sub-linear algorithms applied to *discrete distributions* defined on a support set only containing finite elements (e.g., distribution defined on a positive-integer domain $\{1, 2, ..., n\}$). This limits their applications to complex data, which is often used in machine learning tasks (e.g., image classification). Although it is possible to discretize complex data to a simple support set (then using existing DCT (Diakonikolas et al., 2024)), it is not easy to maintain intrinsic structures and patterns of complex data after discretization (Liu & Rong, 2006; Silverman, 2018).

To extend DCT to more types of data, a natural idea is to introduce *maximum mean discrepancy* (MMD), a powerful kernel-based measurement of distributional discrepancy between two complex distributions (Gretton et al., 2012a; Liu et al., 2020), into DCT scenarios. MMD provides a versatile approach across discrete and continuous domains, and many approaches have extended it to various scenarios, including mean embeddings with test locations (Chwialkowski et al., 2015; Jitkrittum et al., 2016), local difference exploration (Zhou et al., 2023), stochastic process (Salvi et al., 2021), adversarial learning (Gao et al., 2021), and domain adaptation (Zhou et al., 2021). Yet, no one has explored how to extend DCT to complex data with MMD.

**Limitations of Using MMD Directly in DCT.** In this paper, however, we empirically find that many distribution pairs can have the same MMD value despite having different norms in the same *reproducing kernel Hilbert space* (RKHS), and these pairs may exhibit different finite-sample distinguishability and reflect different practical closeness levels, making MMD less informative in DCT. We present an example to illustrate the above issue in Figure 1. The empirical results show that distribution pairs $(\mathbb{P}, \mathbb{Q})$ with

the same MMD value but larger RKHS norms tend to yield smaller $p$-values in assessing equivalence between two distributions, which indicates that $\mathbb{P}$ and $\mathbb{Q}$ are less likely to satisfy the null (i.e., $\mathbb{P} = \mathbb{Q}$). This suggests that, under the given kernel and sample size (i.e., 50), identical MMD values can still correspond to different finite-sample distinguishability, as reflected by the $p$-values, making MMD alone less informative for DCT[1].

**A Norm-Adaptive MMD for DCT.** Motivated by the above *empirical* observation, we propose a new discrepancy measure, *norm-adaptive MMD* (NAMMD), which rescales MMD using the RKHS norms of distributions. In our experiments, among distribution pairs with identical MMD values, those with larger RKHS norms tend to be more distinguishable in finite samples; NAMMD is designed to reflect this variation by assigning larger discrepancy values to such pairs. As shown in Figure 1c and Figure 1d, NAMMD exhibits a stronger empirical correlation with the $p$-value than MMD: *larger NAMMD corresponds to smaller $p$-value* (Figure 1d), whereas MMD *remains unchanged* as the $p$-value varies (Figure 1c).

Eventually, we propose a new NAMMD-based DCT, which derives its testing threshold from the analytical asymptotic null distribution of NAMMD and guarantees type-I error control. Beyond the empirical limitation of MMD discussed above, we further show that NAMMD can provide advantages even in settings where that limitation does not arise, provided a certain norm condition holds. In particular, Theorem 9 establishes that, whenever MMD-based DCT correctly rejects the null, NAMMD-based DCT also rejects it with high probability; moreover, NAMMD-based DCT

---

[1]Appendix D.1.2 associates the decreased $p$-values in Figure 1c with changes in the standard deviation of the MMD estimator.

can reject in cases where MMD-based DCT fails, yielding higher power. We also provide an analysis regarding the sample complexity of NAMMD-based DCT (Theorem 7), i.e., how many samples we need to correctly reject a null hypothesis with high probability.

In experiments, we validate NAMMD-based DCT on benchmark datasets in comparison with state-of-the-art methods. Furthermore, considering practical scenarios in testing distribution closeness, we might use a reference (known) pair of distributions $\mathbb{P}_1$ and $\mathbb{Q}_1$, with their distance serving as the $\epsilon$; and we then test whether the distance between an unknown distribution pair $\mathbb{P}_2$ and $\mathbb{Q}_2$ exceeds that between $\mathbb{P}_1$ and $\mathbb{Q}_1$. Given this, we conduct experiments in three practical case studies to demonstrate the effectiveness of our NAMMD test in evaluating whether a classifier performs relatively similarly across training and test datasets, compared to a prespecified level, without labels (Section 5.2).

## 2. Preliminaries

**Distribution Closeness Testing (DCT).** Denote by $\mathbb{P}$ and $\mathbb{Q}$ two unknown distributions over an instance space $\mathcal{X} \subseteq \mathbb{R}^d$. The DCT assesses whether $\mathbb{P}$ and $\mathbb{Q}$ are $\epsilon$-far from each other under a closeness measurement $d$, where $d$ can be any distance or metric that quantifies the closeness or difference between probability distributions. For convenience, we assume that $d$ is scaled to $[0, 1]$. Formally, given $d$, DCT aims to test between the null and alternative hypotheses as

$$\boldsymbol{H}_0 : d(\mathbb{P}, \mathbb{Q}) \leq \epsilon \quad \text{and} \quad \boldsymbol{H}_1 : d(\mathbb{P}, \mathbb{Q}) > \epsilon \,,$$

where $\epsilon \in [0, 1)$ is the predetermined closeness parameter.

**Maximum Mean Discrepancy (MMD).** The MMD (Gretton et al., 2012a) is a typical kernel-based distance between two distributions. Let $\kappa : \mathcal{X} \times \mathcal{X} \to \mathbb{R}$ be the kernel of a reproducing kernel Hilbert space $\mathcal{H}_\kappa$, with feature map $\kappa(\cdot, \boldsymbol{x}) \in \mathcal{H}_\kappa$ and $0 \leq \kappa(\boldsymbol{x}, \boldsymbol{y}) \leq K$. The kernel mean embeddings (Berlinet & Thomas-Agnan, 2004; Muandet et al., 2017) of distributions $\mathbb{P}$ and $\mathbb{Q}$ are given as

$$\boldsymbol{\mu}_\mathbb{P} = E_{\boldsymbol{x} \sim \mathbb{P}}[\kappa(\cdot, \boldsymbol{x})] \quad \text{and} \quad \boldsymbol{\mu}_\mathbb{Q} = E_{\boldsymbol{y} \sim \mathbb{Q}}[\kappa(\cdot, \boldsymbol{y})] \,.$$

We now define the MMD of $\mathbb{P}$ and $\mathbb{Q}$ as

$$\begin{aligned} \mathrm{MMD}^2(\mathbb{P}, \mathbb{Q}; \kappa) &= \|\boldsymbol{\mu}_\mathbb{P} - \boldsymbol{\mu}_\mathbb{Q}\|_{\mathcal{H}_\kappa}^2 \qquad\qquad (1) \\ &= \|\boldsymbol{\mu}_\mathbb{P}\|_{\mathcal{H}_\kappa}^2 + \|\boldsymbol{\mu}_\mathbb{Q}\|_{\mathcal{H}_\kappa}^2 - 2\langle \boldsymbol{\mu}_\mathbb{P}, \boldsymbol{\mu}_\mathbb{Q} \rangle_{\mathcal{H}_\kappa} \,, \end{aligned}$$

where $\| \cdot \|_{\mathcal{H}_\kappa}^2 = \langle \cdot, \cdot \rangle_{\mathcal{H}_\kappa}$ is the inner product in RKHS $\mathcal{H}_\kappa$.

**Two-sample Testing (TST).** TST aims to assess a fundamentally different hypothesis testing problem compared to DCT, with the null and the alternative hypotheses as follows

$$\boldsymbol{H}_0^{\mathrm{TST}} : \mathbb{P} = \mathbb{Q} \quad \text{and} \quad \boldsymbol{H}_1^{\mathrm{TST}} : \mathbb{P} \neq \mathbb{Q} \,,$$

For characteristic kernels, $\mathrm{MMD}(\mathbb{P}, \mathbb{Q}; \kappa) = 0$ if and only if $\mathbb{P} = \mathbb{Q}$. Hence, MMD can be readily applied to the two-sample testing. A wide range of discrepancy measurements has also been developed for comparing probability distributions in the TST setting, including integral probability metrics (IPMs) such as total variation and Wasserstein distance, and $f$-divergences such as KL and $\chi^2$.

**DCT vs. TST: what changes, and why it matters.** The key difference lies in the null hypothesis. TST tests a *point null* $\mathbb{P} = \mathbb{Q}$, while DCT tests a *margin/composite null* $d(\mathbb{P}, \mathbb{Q}) \leq \epsilon$. When $\epsilon = 0$, DCT reduces to TST; for $\epsilon > 0$, "failing to reject" in DCT should be read as "within tolerance" rather than "identical". This perspective is often more aligned with practice, where small distribution shifts may still be acceptable, and the goal is to distinguish *different levels of closeness* rather than detect any difference.

**Why DCT cannot reuse TST procedures.** A common way to implement TST is the permutation test: under $\boldsymbol{H}_0^{\mathrm{TST}} : \mathbb{P} = \mathbb{Q}$, the pooled sample is exchangeable, so randomly permuting the sample labels leaves the joint distribution invariant. As a result, the permutation distribution of a test statistic provides valid *critical values* (i.e., testing thresholds) that control the type-I error without requiring an explicit asymptotic null distribution. In contrast, under the DCT null $d(\mathbb{P}, \mathbb{Q}) \leq \epsilon$, $\mathbb{P}$ and $\mathbb{Q}$ may differ, so samples from $\mathbb{P}$ and $\mathbb{Q}$ are generally not interchangeable and label permutations change the data-generating mechanism. Hence, the permutation distribution no longer represents the null distribution, and permutation-based critical values are invalid for DCT; one therefore needs a tractable characterization of the test statistic under the composite null to determine valid rejection thresholds.

**Why we focus on MMD.** This requirement rules out many discrepancies that are popular in TST. For example, total variation and KL divergence typically do not yield convenient asymptotic null distributions for plug-in estimators. Moreover, while general nonparametric frameworks exist for estimating such discrepancies in continuous settings (Sriperumbudur et al., 2009; Nowozin et al., 2016), certain metrics (e.g., total variation) often require additional structural assumptions for consistency (Devroye & Györfi, 1990; Barron et al., 2002); without them, they may fail to reliably order distribution pairs by closeness, which is central to DCT. In contrast, MMD yields well-behaved estimators and tractable asymptotic null theory, making it a practical and theoretically sound discrepancy for DCT.

## 3. NAMMD-based DCT

As discussed in the introduction and Figure 1, MMD is effective for detecting whether two distributions are identical, but MMD alone may be less informative for DCT when dis-

tribution pairs with identical MMD values exhibit different finite-sample distinguishability. This motivates us to incorporate the RKHS norms of distributions into the discrepancy measure, so that pairs with the same MMD value can be further distinguished when their RKHS norms differ.

**NAMMD and Its Asymptotic Property.** We define the norm-adaptive maximum mean discrepancy (NAMMD) by rescaling MMD with the RKHS norms of the distributions. This rescaling allows distribution pairs with identical MMD values to have different NAMMD values when their RKHS norms differ; in particular, pairs with larger RKHS norms are assigned larger NAMMD values.

**Definition 1.** For a kernel $\kappa$ with $\mathcal{H}_\kappa$ and $0 \leq \kappa(\boldsymbol{x}, \boldsymbol{y}) \leq K$, we define the *norm-adaptive maximum mean discrepancy* (NAMMD) w.r.t. distributions $\mathbb{P}$ and $\mathbb{Q}$ as follows:

$$\text{NAMMD}(\mathbb{P}, \mathbb{Q}; \kappa) = \frac{\|\boldsymbol{\mu}_\mathbb{P} - \boldsymbol{\mu}_\mathbb{Q}\|^2_{\mathcal{H}_\kappa}}{4K - \|\boldsymbol{\mu}_\mathbb{P}\|^2_{\mathcal{H}_\kappa} - \|\boldsymbol{\mu}_\mathbb{Q}\|^2_{\mathcal{H}_\kappa}} . \quad (2)$$

In this paper, we focus on bounded, shift-invariant kernels of the form $\kappa(\boldsymbol{x}, \boldsymbol{x}') = \Psi(\boldsymbol{x} - \boldsymbol{x}')$, where $\Psi(\boldsymbol{0}) = K > 0$, $\Psi(\cdot) \leq K$, and $\Psi(\cdot)$ is positive-definite. For this kernel class, the numerator of NAMMD is $\text{MMD}^2(\mathbb{P}, \mathbb{Q}; \kappa)$, which lies in $[0, 2K]$, while the denominator $4K - \|\boldsymbol{\mu}_\mathbb{P}\|^2_{\mathcal{H}_\kappa} - \|\boldsymbol{\mu}_\mathbb{Q}\|^2_{\mathcal{H}_\kappa}$ lies in $[2K, 4K]$. Consequently, $0 \leq \text{NAMMD}(\mathbb{P}, \mathbb{Q}; \kappa) \leq 1$. Moreover, NAMMD approaches 1 when two distributions $\mathbb{P}$ and $\mathbb{Q}$ are *well-separated* from each other and both are *highly concentrated*[2].

In NAMMD, we essentially capture differences between two distributions using their characteristic kernel mean embeddings (i.e. $\boldsymbol{\mu}_\mathbb{P}$ and $\boldsymbol{\mu}_\mathbb{Q}$), which uniquely represent distributions and capture distinct characteristics for effective comparison (Sriperumbudur et al., 2011). Equivalently, NAMMD can be viewed as MMD scaled by the RKHS variances of $\mathbb{P}$ and $\mathbb{Q}$. Specifically, for the kernel class considered above, the variances are $\text{Var}(\mathbb{P}; \kappa) = E_{\boldsymbol{x} \sim \mathbb{P}}[\kappa(\boldsymbol{x}, \boldsymbol{x})] - \|\boldsymbol{\mu}_\mathbb{P}\|^2_{\mathcal{H}_\kappa} = K - \|\boldsymbol{\mu}_\mathbb{P}\|^2_{\mathcal{H}_\kappa}$ and $\text{Var}(\mathbb{Q}; \kappa) = K - \|\boldsymbol{\mu}_\mathbb{Q}\|^2_{\mathcal{H}_\kappa}$. Hence, we have $\text{NAMMD}(\mathbb{P}, \mathbb{Q}; \kappa) = \text{MMD}^2(\mathbb{P}, \mathbb{Q}; \kappa)/(2K + \text{Var}(\mathbb{P}; \kappa) + \text{Var}(\mathbb{Q}; \kappa))$.

Several prior normalization strategies for kernel tests have been developed for TST, but their objectives differ from ours. For example, Kernel FDA (Harchaoui et al., 2007) focuses on testing equality of the mean and covariance of two distributions; Spectral-MMD (Hagrass et al., 2024) introduces normalization to achieve minimax separation rates under $\Delta$-separated alternatives; and analytic or feature-based approaches (Chwialkowski et al., 2015; Jitkrittum et al., 2016) are constructed to improve sensitivity to local discrepancies. These methods are highly effective for detecting differences under $P = Q$, the classical TST null. However, *DCT places*

*more emphasis on distinguishing different closeness levels for different distribution pairs*, and NAMMD is introduced to mitigate a specific limitation of MMD in this regard.

In practice, $\mathbb{P}$ and $\mathbb{Q}$ are generally unknown, and we can only observe two i.i.d. samples[3]

$$X = \{\boldsymbol{x}_i\}_{i=1}^m \sim \mathbb{P}^m \text{ and } Y = \{\boldsymbol{y}_j\}_{j=1}^m \sim \mathbb{Q}^m .$$

Based on two samples $X$ and $Y$, we introduce the empirical estimator of our NAMMD as follows

$$\widehat{\text{NAMMD}}(X, Y; \kappa) = \frac{\sum_{i \neq j} H_{i,j}}{\sum_{i \neq j}[4K - \kappa(\boldsymbol{x}_i, \boldsymbol{x}_j) - \kappa(\boldsymbol{y}_i, \boldsymbol{y}_j)]},$$

where $H_{i,j} = \kappa(\boldsymbol{x}_i, \boldsymbol{x}_j) + \kappa(\boldsymbol{y}_i, \boldsymbol{y}_j) - \kappa(\boldsymbol{x}_i, \boldsymbol{y}_j) - \kappa(\boldsymbol{y}_i, \boldsymbol{x}_j)$. Then, we prove an asymptotic distribution of NAMMD when two distributions are different in following theorem.

**Lemma 2.** *If* $\text{NAMMD}(\mathbb{P}, \mathbb{Q}; \kappa) = \epsilon > 0$*, we have*

$$\sqrt{m}(\widehat{\text{NAMMD}}(X, Y; \kappa) - \epsilon) \xrightarrow{d} \mathcal{N}(0, \sigma^2_{\mathbb{P},\mathbb{Q}}) ,$$

*where* $\sigma_{\mathbb{P},\mathbb{Q}} = \sqrt{4E[H_{1,2}H_{1,3}] - 4(E[H_{1,2}])^2}/(4K - \|\boldsymbol{\mu}_\mathbb{P}\|^2_{\mathcal{H}_\kappa} - \|\boldsymbol{\mu}_\mathbb{Q}\|^2_{\mathcal{H}_\kappa})$*, and the expectation are taken over* $\boldsymbol{x}_1, \boldsymbol{x}_2, \boldsymbol{x}_3 \sim \mathbb{P}^3$ *and* $\boldsymbol{y}_1, \boldsymbol{y}_2, \boldsymbol{y}_3 \sim \mathbb{Q}^3$.

We now present the NAMMD-based DCT, along with an appropriately estimated testing threshold from the above analytical and asymptotic distribution.

**NAMMD-DCT Testing Procedure.** In the following, we instantiate the distribution closeness testing in Section 2 using NAMMD as the closeness measurement.

**Definition 3.** Given the closeness parameter $\epsilon \in (0, 1)$, the goal is to test between hypotheses

$$\boldsymbol{H}_0 : \text{NAMMD}(\mathbb{P}, \mathbb{Q}; \kappa) \leq \epsilon$$
$$\boldsymbol{H}_1 : \text{NAMMD}(\mathbb{P}, \mathbb{Q}; \kappa) > \epsilon ,$$

with the significance level $\alpha \in (0, 1)$.

To conduct a hypothesis testing procedure for distribution closeness, we first estimate the testing threshold $\hat{\tau}_\alpha$ under the null hypothesis $\boldsymbol{H}_0 : \text{NAMMD}(\mathbb{P}, \mathbb{Q}; \kappa) \leq \epsilon$ at significance level $\alpha$. The null hypothesis is composite, consisting of the case $\text{NAMMD}(\mathbb{P}, \mathbb{Q}; \kappa) = \epsilon$ and the case $\text{NAMMD}(\mathbb{P}, \mathbb{Q}; \kappa) < \epsilon$. Since the value $\text{NAMMD}(\mathbb{P}, \mathbb{Q}; \kappa)$ is unknown, we set the testing threshold $\hat{\tau}_\alpha$ as the estimated $(1 - \alpha)$-quantile of the the asymptotic Gaussian

---

[2]See Appendix B.1, which provides further details on the conditions under which NAMMD approaches 1.

[3]Following Liu et al. (2020), we assume equal sample sizes for the two samples to simplify notation, while our results can be extended to unequal sample sizes using multi-sample $U$-statistics (Korolyuk & Borovskich, 2013); see Appendix B.2 for details. The i.i.d. assumption is required for following asymptotic analysis. Extending the proposed test to *non-i.i.d.* data requires dependence-aware calibration and is left for future work.

distribution of $\widehat{\text{NAMMD}}(X, Y; \kappa)$ under the case where $\text{NAMMD}(\mathbb{P}, \mathbb{Q}; \kappa) = \epsilon$ (i.e., the least-favorable boundary of the composite null hypothesis) as shown in Lemma 2. For the asymptotic distribution, the term $\sigma_{\mathbb{P},\mathbb{Q}}^2$ is unknown and we use its estimator

$$\sigma_{X,Y} = \tag{3}$$
$$\frac{\sqrt{((4m-8)\hat{\zeta}_1 + 2\hat{\zeta}_2)/(m-1)}}{(m^2-m)^{-1}\sum_{i \neq j} 4K - \kappa(\boldsymbol{x}_i, \boldsymbol{x}_j) - \kappa(\boldsymbol{y}_i, \boldsymbol{y}_j)} \,,$$

where $\hat{\zeta}_1$ and $\hat{\zeta}_2$ are estimated standard variance components of the MMD (Serfling, 2009; Sutherland, 2019) (See Appendix B.3). Lemma 4 shows that the estimator $\sigma_{X,Y}^2$ converges to $\sigma_{\mathbb{P},\mathbb{Q}}^2$ at a rate of $O(1/\sqrt{m})$.

We now have the testing threshold for null hypothesis $\boldsymbol{H}_0$ : $\text{NAMMD}(\mathbb{P}, \mathbb{Q}; \kappa) \leq \epsilon$ with $\epsilon \in (0, 1)$ as

$$\hat{\tau}_\alpha = \epsilon + \sigma_{X,Y} \mathcal{N}_{1-\alpha}/\sqrt{m} \,, \tag{4}$$

where $\mathcal{N}_{1-\alpha}$ is the $(1 - \alpha)$-quantile of the standard normal distribution $\mathcal{N}(0, 1)$.

Finally, we have the following test with testing threshold $\hat{\tau}_\alpha$

$$h(X, Y; \kappa) = \mathbb{I}[\widehat{\text{NAMMD}}(X, Y; \kappa) > \hat{\tau}_\alpha] \,. \tag{5}$$

**Performing DCT in Practice.** We have demonstrated the NAMMD-based DCT above, yet it is still not clear how the $\epsilon$ of Definition 3 should be set in practice. Normally, when we want to test the closeness, we often have a reference pair of distributions $\mathbb{P}_1$ and $\mathbb{Q}_1$ where the closeness between $\mathbb{P}_1$ and $\mathbb{Q}_1$ is acceptable/satisfactory. For example, although ImageNet and Pascal VOC are from different distributions, the model trained on ImageNet can still have good performance on Pascal VOC. Thus, we can use the NAMMD's empirical value between ImageNet and Pascal VOC as the prespecified $\epsilon$ in this case. Then, given two samples $X$ and $Y$ drawn from an unknown pair of distributions $\mathbb{P}_2$ and $\mathbb{Q}_2$ respectively, we seek to determine whether the distance between $\mathbb{P}_2$ and $\mathbb{Q}_2$ is as close or closer to that between $\mathbb{P}_1$ and $\mathbb{Q}_1$, by applying distribution closeness testing. Here, given the specified $\epsilon$, this DCT problem can be formalized by Definition 3 with hypotheses as

$$\boldsymbol{H}_0 : \text{NAMMD}(\mathbb{P}_2, \mathbb{Q}_2; \kappa) \leq \epsilon$$
$$\boldsymbol{H}_1 : \text{NAMMD}(\mathbb{P}_2, \mathbb{Q}_2; \kappa) > \epsilon \,.$$

Finally, we can perform testing procedure with $X$ and $Y$.

In practice, the value of $\epsilon$ is typically estimated from a sufficiently large amount of reference data and used as a plug-in tolerance level. When no obvious reference pair is available, one practical heuristic is to construct a synthetic reference pair by adding controlled perturbations, such as Gaussian noise, to one distribution. The resulting perturbed pairs provide a range of discrepancies for calibrating $\epsilon$; for example, $\epsilon$ may be chosen according to the largest perturbation level still regarded as negligible for the application.

**Kernel Selection and related Works.** For kernel selection in DCT, we select a fixed global kernel for all distribution pairs, which is essential for effectively comparing their closeness levels under a unified measurement. However, existing kernel selections are primarily designed for TST (Liu et al., 2020; Sutherland et al., 2017), selecting a kernel to maximize the $t$-statistic in test power estimation to distinguish a fixed distribution pair. In DCT, deriving a test power estimator with several different distribution pairs remains an open question and we follow the TST approaches to select a kernel to distinguish between $\mathbb{P}_1$ and $\mathbb{Q}_1$ in practice (see Appendix B.5)[4]. See Appendix B.4 for more related works, including those on testing-threshold estimation, kernel selection, etc.

**Applying NAMMD to Two-Sample Testing.** Although the NAMMD is specially designed for DCT, it is still a statistic to measure the distributional discrepancy between two distributions. Thus, it is interesting to apply it to two-sample testing (TST) scenarios. In TST, we aim to assess the equivalence between distributions $\mathbb{P}$ and $\mathbb{Q}$, where the null hypothesis assumes $\mathbb{P} = \mathbb{Q}$ and is tested against the alternative hypothesis $\mathbb{P} \neq \mathbb{Q}$. Following MMD-based approaches to TST (Sutherland et al., 2017), we use a standard permutation test to estimate the test threshold $\hat{\tau}_\alpha$, which estimate the null distribution by repeatedly re-computing estimator with the samples randomly re-assigned to $X$ or $Y$ (see Appendix B.6 for details).

## 4. Theoretical Analysis

In this section, we make theoretical investigations regarding NAMMD-based DCT and compare NAMMD and the MMD in addressing the DCT problem. All the proofs are presented in Appendix C. We first provide theoretical guarantees for the variance estimation and concentration properties of the NAMMD estimator. Specifically, for the variance estimator $\sigma_{X,Y}$ in Eqn. (3), we have

**Lemma 4.** *Given samples $X$ and $Y$ with size $m$, we have that $\left| E[\sigma_{X,Y}^2] - \sigma_{\mathbb{P},\mathbb{Q}}^2 \right| = O(1/\sqrt{m})$.*

We now present the large deviation bound for our NAMMD.

**Lemma 5.** *Over samples $X$ and $Y$ of size $m$, the probability $\Pr\left( \left| \widehat{\text{NAMMD}}(X, Y; \kappa) - \text{NAMMD}(\mathbb{P}, \mathbb{Q}; \kappa) \right| \geq t \right)$ is at*

---

[4]A possible direction is to optimize an aggregate DCT criterion over multiple candidate pairs, rather than tailoring kernel selection to a single pair; for example, the criterion may aggregate NAMMD-based testing criteria across reference and test pairs, but its precise formulation and theoretical analysis remain largely open.

most $4 \exp(-mt^2/9)$ *for* $t > 0$.

Next, we study type-I error for NAMMD-based DCT.

**Theorem 6.** *Under* $\boldsymbol{H}_0$ : $\mathrm{NAMMD}(\mathbb{P}, \mathbb{Q}; \kappa) \leq \epsilon$ *with* $\epsilon \in (0, 1)$, *the type-I error of NAMMD-based DCT is asymptotically bounded by* $\alpha$, *i.e.,* $\mathrm{Pr}_{\boldsymbol{H}_0}(h(X, Y; \kappa) = 1) \leq \alpha$.

We then analyze the sample complexity regarding NAMMD-based DCT to correctly reject the null hypothesis.

**Theorem 7.** *For NAMMD-based DCT, formally defined in Eqn.* (5), *we correctly reject null hypothesis* $\boldsymbol{H}_0$ : $\mathrm{NAMMD}(\mathbb{P}, \mathbb{Q}; \kappa) \leq \epsilon \in (0, 1)$ *with probability at least* $1 - \upsilon$ *given the sample size*

$$m \geq \frac{\left(2 * \mathcal{N}_{1-\alpha} + \sqrt{9 \log 2/\upsilon}\right)^2}{(\mathrm{NAMMD}(\mathbb{P}, \mathbb{Q}; \kappa) - \epsilon)^2} .$$

This theorem shows that the ratio $1/(\mathrm{NAMMD}(\mathbb{P}, \mathbb{Q}; \kappa) - \epsilon)^2$ is the main quantity dictating the sample complexity of our NAMMD test under $\boldsymbol{H}_1$ : $\mathrm{NAMMD}(\mathbb{P}, \mathbb{Q}; \kappa) > \epsilon$.

**Comparison between NAMMD- and MMD-based DCT.** As demonstrated in Section 3, in practice, we might often need a reference pair to confirm the value of $\epsilon$, thus, we first reformalize the DCT testing procedure with the reference pair, which is shown in the following definition.

**Definition 8.** Given the reference distributions $\mathbb{P}_1$ and $\mathbb{Q}_1$, and samples $X$ and $Y$ drawn from unknown distributions $\mathbb{P}_2$ and $\mathbb{Q}_2$, the goal of DCT is to correctly determine whether the distance between $\mathbb{P}_2$ and $\mathbb{Q}_2$ is larger than that between $\mathbb{P}_1$ and $\mathbb{Q}_1$. To compare the test power, we perform NAMMD-based DCT and MMD-based DCT separately, under scenarios where the following two null hypotheses for NAMMD- and MMD-based DCT are simultaneously false:

$$\boldsymbol{H}_0^N : \mathrm{NAMMD}(\mathbb{P}_2, \mathbb{Q}_2; \kappa) \leq \epsilon^N$$
$$\boldsymbol{H}_0^M : \mathrm{MMD}^2(\mathbb{P}_2, \mathbb{Q}_2; \kappa) \leq \epsilon^M ,$$

and the following alternatives hold simultaneously:

$$\boldsymbol{H}_1^N : \mathrm{NAMMD}(\mathbb{P}_2, \mathbb{Q}_2; \kappa) > \epsilon^N$$
$$\boldsymbol{H}_1^M : \mathrm{MMD}^2(\mathbb{P}_2, \mathbb{Q}_2; \kappa) > \epsilon^M ,$$

with the tolerance levels $\epsilon^N = \mathrm{NAMMD}(\mathbb{P}_1, \mathbb{Q}_1; \kappa)$ and $\epsilon^M = \mathrm{MMD}^2(\mathbb{P}_1, \mathbb{Q}_1; \kappa)$.

Based on *the specific setting in the above definition*, we show in the following theoretical analysis that NAMMD-based DCT can still provide advantages *even when the limitation of MMD (see Section 1) does not arise*, provided that a certain norm condition holds.

**Theorem 9.** *Under* $\boldsymbol{H}_1^N$ : $\mathrm{NAMMD}(\mathbb{Q}_2, \mathbb{P}_2; \kappa) > \epsilon^N$ *and* $\boldsymbol{H}_1^M$ : $\mathrm{MMD}^2(\mathbb{Q}_2, \mathbb{P}_2; \kappa) > \epsilon^M$, *assume further that* $\|\boldsymbol{\mu}_{\mathbb{P}_1}\|_{\mathcal{H}_\kappa}^2 + \|\boldsymbol{\mu}_{\mathbb{Q}_1}\|_{\mathcal{H}_\kappa}^2 < \|\boldsymbol{\mu}_{\mathbb{P}_2}\|_{\mathcal{H}_\kappa}^2 + \|\boldsymbol{\mu}_{\mathbb{Q}_2}\|_{\mathcal{H}_\kappa}^2$.

*Then the following relation holds with probability at least*
$$1 - \exp\left(\frac{-m\Delta^2(4K - \|\boldsymbol{\mu}_{\mathbb{P}_2}\|_{\mathcal{H}_\kappa}^2 - \|\boldsymbol{\mu}_{\mathbb{Q}_2}\|_{\mathcal{H}_\kappa}^2)^2}{4K^2(1 - \Delta)^2}\right).$$

$$\sqrt{m}\widehat{\mathrm{MMD}}(X, Y; \kappa) > \tau_\alpha^M \ \Rightarrow \ \sqrt{m}\widehat{\mathrm{NAMMD}}(X, Y; \kappa) > \tau_\alpha^N,$$

*where* $\tau_\alpha^M$ *and* $\tau_\alpha^N$ *are asymptotic* $(1 - \alpha)$-*thresholds of the null distributions of* $\sqrt{m}\widehat{\mathrm{MMD}}$ *and* $\sqrt{m}\widehat{\mathrm{NAMMD}}$. *Given* $\sigma_M$ *in Eqn.* (7) *(Appendix C.6.1),* $\Delta \in (0, 1/2)$ *is given by*

$$\Delta = \sqrt{m}\mathrm{NAMMD}(\mathbb{P}_1, \mathbb{Q}_1; \kappa)$$
$$\times \frac{\|\boldsymbol{\mu}_{\mathbb{P}_2}\|_{\mathcal{H}_\kappa}^2 + \|\boldsymbol{\mu}_{\mathbb{Q}_2}\|_{\mathcal{H}_\kappa}^2 - \|\boldsymbol{\mu}_{\mathbb{P}_1}\|_{\mathcal{H}_\kappa}^2 - \|\boldsymbol{\mu}_{\mathbb{Q}_1}\|_{\mathcal{H}_\kappa}^2}{\sqrt{m}\mathrm{MMD}^2(\mathbb{P}_1, \mathbb{Q}_1; \kappa) + \sigma'_M \mathcal{N}_{1-\alpha}}.$$

*Furthermore, the following relation holds with probability* $\varsigma \geq 1/65$ *over samples* $X$ *and* $Y$,

$$\sqrt{m}\widehat{\mathrm{MMD}}(X, Y; \kappa) \leq \tau_\alpha^M \ \text{yet} \ \sqrt{m}\widehat{\mathrm{NAMMD}}(X, Y; \kappa) > \tau_\alpha^N,$$

*if* $C_1 \leq m \leq C_2$, *where* $C_1$ *and* $C_2$ *are dependent on distributions* $\mathbb{P}$ *and* $\mathbb{Q}$, *and probability* $\varsigma$.

This theorem provides a conditional power comparison between MMD-based DCT and NAMMD-based DCT. Under the same kernel and the stated norm condition $\|\boldsymbol{\mu}_{\mathbb{P}_1}\|_{\mathcal{H}_\kappa}^2 + \|\boldsymbol{\mu}_{\mathbb{Q}_1}\|_{\mathcal{H}_\kappa}^2 < \|\boldsymbol{\mu}_{\mathbb{P}_2}\|_{\mathcal{H}_\kappa}^2 + \|\boldsymbol{\mu}_{\mathbb{Q}_2}\|_{\mathcal{H}_\kappa}^2$, if the MMD test rejects the null hypothesis under the alternative, then the NAMMD test also rejects the null hypothesis with high probability. Furthermore, the NAMMD test can reject the null hypothesis even in cases where the original MMD test fails to do so. This result should not be interpreted as a uniform dominance claim: it relies on the stated norm condition and the use of asymptotic thresholds under the same kernel.

The stated norm condition is used to characterize the setting where the reference pair is closer than the test pair under MMD, i.e., $\mathrm{MMD}^2(\mathbb{P}_1, \mathbb{Q}_1; \kappa) < \mathrm{MMD}^2(\mathbb{P}_2, \mathbb{Q}_2; \kappa)$. This condition is related to the MMD decomposition in Eqn. (1), where the RKHS norms contribute to the MMD value together with the negative inner-product term $-2\langle\boldsymbol{\mu}_{\mathbb{P}}, \boldsymbol{\mu}_{\mathbb{Q}}\rangle_{\mathcal{H}_\kappa}$. In particular, when the inner-product terms are comparable across distribution pairs, larger RKHS norms can correspond to larger MMD values. In Appendix C.6.2, we further discuss this norm condition and the constants $C_1$ and $C_2$ in Theorem 9. While the analysis is asymptotic, the empirical results in Section 5 illustrate the finite-sample behavior and potential benefits of NAMMD.

Although Theorem 9 assumes the same kernel for NAMMD- and MMD-based DCT, it also provides insight into the case where each test uses its own optimal kernel. Here, an optimal kernel refers to an unknown oracle kernel within a given kernel class that maximizes the asymptotic power of the corresponding DCT procedure for fixed distributions, tolerance level, sample size, and significance level. Such a kernel is generally unavailable in practice because it depends on the underlying distributions. Let $\kappa_*^M$ denote such

**Table 1.** Test power (mean±std) on DCT versus different total variation values $\epsilon'$, and the bold denotes the highest mean between our NAMMD and Canonne's tests. The experiments are conducted on discrete distributions defined over the same support set.

| Dataset | $\epsilon' = 0.1$ | | $\epsilon' = 0.3$ | | $\epsilon' = 0.5$ | | $\epsilon' = 0.7$ | |
| | Canonne's | NAMMD | Canonne's | NAMMD | Canonne's | NAMMD | Canonne's | NAMMD |
|---|---|---|---|---|---|---|---|---|
| blob | .856±.023 | **.968±.022** | .809±.014 | **.912±.053** | .944±.013 | **.960±.020** | **.998±.002** | .961±.029 |
| higgs | .883±.015 | **.908±.050** | .825±.010 | **.947±.027** | .960±.005 | **.962±.023** | .994±.003 | **.995±.005** |
| hdgm | .861±.011 | **.942±.023** | .888±.016 | **.946±.017** | .937±.014 | **.965±.014** | .987±.004 | **.989±.004** |
| mnist | .715±.021 | **.931±.024** | .786±.026 | **.965±.007** | .896±.013 | **.997±.001** | .971±.008 | **1.00±.000** |
| cifar10 | .686±.030 | **.919±.017** | .751±.021 | **.923±.021** | .917±.006 | **.997±.002** | .981±.004 | **.999±.001** |
| Average | .800±.020 | **.934±.027** | .812±.017 | **.939±.025** | .931±.010 | **.976±.012** | .986±.004 | **.989±.008** |

an optimal kernel for MMD-based DCT. Under the norm condition in Theorem 9, NAMMD-based DCT with the same kernel $\kappa_*^{\mathrm{M}}$ has higher asymptotic power than MMD-based DCT with $\kappa_*^{\mathrm{M}}$. Therefore, if $\kappa_*^{\mathrm{N}}$ denotes an optimal kernel for NAMMD-based DCT, the asymptotic power of NAMMD-based DCT with $\kappa_*^{\mathrm{N}}$ is no smaller than that with $\kappa_*^{\mathrm{M}}$. This gives an oracle-level comparison suggesting that the advantage indicated by Theorem 9 is not restricted to using a common kernel.

## 5. Experiments

We perform DCT and TST on five benchmark datasets used by previous hypothesis testing studies (Liu et al., 2020; Zhou et al., 2023). Specifically, "blob" and "hdgm" are synthetic Gaussian mixtures with dimensions 2 and 10. The "higgs" are tabular dataset consisting of the 4 dimension $\phi$-momenta distributions of Higgs-producing and background processes. "mnist" and "cifar" are image datasets consisting of original and generative images. We also conduct experiments on practical tasks related to domain adaptation using ImageNet and its variants, and evaluating adversarial perturbations on CIFAR10. More experiments, including **type-I error** for both DCT and TST, can be found in Appendix E.

### 5.1. Experiments on Benchmark Datasets

**First,** we compare the test power of DCTs using our NAMMD and the statistic based on total variation introduced by Canonne et al. (2022), and the experiments are conducted on *discrete distributions with the same support set containing only finite elements*. For each dataset, we randomly draw 50 elements $Z = \{z_1, z_2, ..., z_{50}\}$, and denote by $\mathbb{P}_{50}$ the uniform distribution over domain $Z$. We further construct distributions $\mathbb{Q}_{50}$ and $\mathbb{Q}_{50}^A$ for null and alternative hypotheses respectively, which satisfies $\mathrm{TV}(\mathbb{P}_{50}, \mathbb{Q}_{50}) = \epsilon'$ and $\mathrm{TV}(\mathbb{P}_{50}, \mathbb{Q}_{50}^A) = \epsilon' + 0.2$ (Details are provided in Appendix D.2). In experiments, we draw two i.i.d samples from $\mathbb{P}_{50}$ and $\mathbb{Q}_{50}^A$ to evaluate if the distance between $\mathbb{P}_{50}$ and $\mathbb{Q}_{50}^A$ is larger than that between $\mathbb{P}_{50}$ and $\mathbb{Q}_{50}$, i.e, $\epsilon'$. Table 1 summarizes the average test powers and standard deviations of NAMMD-based DCT and Canonne's DCT (Appendix D.2) based on total variaton.

For comparison, we set $\epsilon' \in \{0.1, 0.3, 0.5, 0.7\}$[5]. From Table 1, NAMMD-based DCT generally performs better than Canonne's DCT, except on 2-dimensional blob dataset with $\epsilon' = 0.7$, where Canonne's DCT has lower variance and captures fine-grained distributional difference. See Appendix E.2 for NAMMD versus Wasserstein experimental results under the same setting.

**Second,** we compare NAMMD with more baselines (Appendix D.4) on TST, include: 1) MMDFuse (Biggs et al., 2023); 2) MMD-D (Liu et al., 2020); 3) MMDAgg (Schrab et al., 2022); 4) AutoTST (Kübler et al., 2022); 5) ME$_{\mathrm{MaBiD}}$ (Zhou et al., 2023); 6) ACTT (Domingo-Enrich et al., 2023). Although we discuss NAMMD with a fixed kernel in this paper, it is compatible with various kernel selection frameworks as MMD. To illustrate this, we adapt NAMMD with multiple kernels using the fusion method (Biggs et al., 2023) and refer to it as NAMMDFuse (Appendix D.5). From Figure 2, it is observed that NAMMDFuse achieves test power that is either higher or comparable to other methods. Besides the multiple kernel scheme, we also empirically demonstrate that NAMMD can be applied with various kernels (Gaussian, Laplace, Mahalanobis, and deep kernels) and achieves better performance than MMD under the same kernel, as shown in Table 13 (Appendix E).

**Third,** to compare our NAMMD and original MMD in DCT, we first *select the kernel* $\kappa$ based on the original distribution pair $(\mathbb{P}, \mathbb{Q})$ of the dataset, following the TST approach (Liu et al., 2020). Based on the selected kernel $\kappa$ and following the setup in Definition 8, we construct two pairs of distributions: $\mathbb{P}_1$ and $\mathbb{Q}_1$, and $\mathbb{P}_2$ and $\mathbb{Q}_2$, where $\mathrm{NAMMD}(\mathbb{P}_1, \mathbb{Q}_1; \kappa) = \epsilon$ and $\mathrm{NAMMD}(\mathbb{Q}_2, \mathbb{P}_2; \kappa) = \epsilon + 0.01$, and $\mathrm{MMD}^2(\mathbb{P}_1, \mathbb{Q}_1; \kappa) < \mathrm{MMD}^2(\mathbb{Q}_2, \mathbb{P}_2; \kappa)$. Details are provided in Appendix D.3.

For comparison, we set $\epsilon \in \{0.1, 0.3, 0.5, 0.7\}$. We randomly draw two samples from $\mathbb{Q}_2$ and $\mathbb{P}_2$ evaluate if distance between $\mathbb{P}_2$ and $\mathbb{Q}_2$ is larger than that between $\mathbb{P}_1$ and $\mathbb{Q}_1$. Table 2 summarizes the average test powers and standard deviations of our NAMMD distance and original

---

[5]Although $\epsilon'$ increases, the difference between two total variation values, namely the ground-truth total variation between $\mathbb{P}_{50}$ and $\mathbb{Q}_{50}^A$ minus that between $\mathbb{P}_{50}$ and $\mathbb{Q}_{50}$, remains fixed at 0.2.

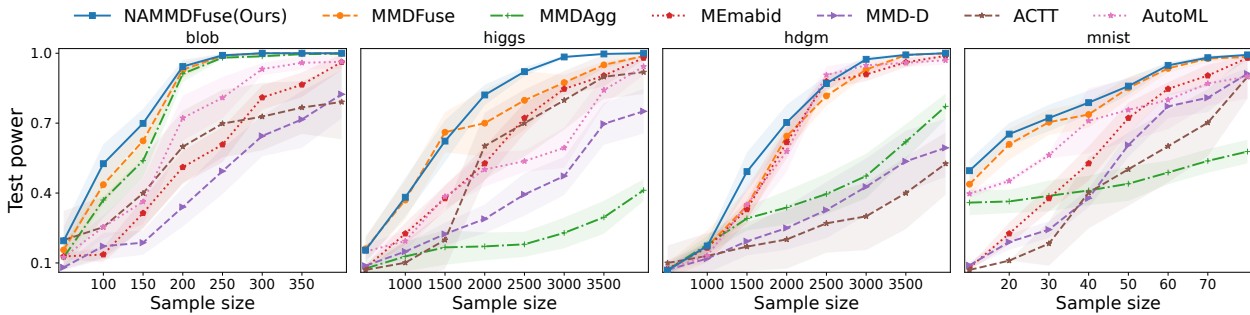

**Figure 2.** The comparisons of test power vs sample size for our NAMMDFuse and SOTA two-sample tests.

**Table 2.** Comparisons of test power (mean±std) on DCT with respect to different NAMMD values, and the bold denotes the highest mean between tests with our NAMMD and original MMD. Notably, the same selected kernel is applied for both NAMMD and MMD in this table. The experiments are not limited to discrete distributions defined over the same support set, which is different from those in Table 1.

| Dataset | $\epsilon = 0.1$ | | $\epsilon = 0.3$ | | $\epsilon = 0.5$ | | $\epsilon = 0.7$ | |
|---|---|---|---|---|---|---|---|---|
| | MMD | NAMMD | MMD | NAMMD | MMD | NAMMD | MMD | NAMMD |
| blob | .974±.009 | **.978±.008** | .890±.030 | **.923±.025** | .902±.032 | **.924±.021** | .909±.024 | **.933±.011** |
| higgs | .998±.002 | **.999±.001** | .938±.020 | **.965±.013** | .975±.012 | **.993±.003** | .978±.010 | **.996±.002** |
| hdgm | .980±.007 | **.984±.007** | .883±.027 | **.921±.021** | .901±.025 | **.941±.013** | **1.00±.000** | **1.00±.000** |
| mnist | **.982±.004** | **.982±.004** | .961±.006 | **.974±.004** | .946±.014 | **.983±.005** | .962±.010 | **.991±.003** |
| cifar10 | .932±.007 | **.938±.007** | .968±.019 | **.994±.003** | .898±.054 | **.912±.041** | **1.00±.000** | **1.00±.000** |
| Average | .973±.006 | **.976±.005** | .928±.020 | **.955±.013** | .924±.027 | **.951±.017** | .970±.009 | **.984±.003** |

MMD distance in DCT for *distributions over continuous domains*. It is evident that our NAMMD-based DCT achieves better performances than the original MMD-based DCT with respect to different datasets.

### 5.2. Performing DCT in Practical Tasks

We present three practical case studies demonstrating the effectiveness of NAMMD-based DCT. First, given a pre-trained ResNet50 that performs well on ImageNet, we evaluate its performance on several ImageNet variants, including ImageNet-A, ImageNet-V2, ImageNet-R, and ImageNet-Sketch. As a reference, we consider the accuracy gap (Eqn. 14 in Appendix D.7), defined as the difference in model accuracy between ImageNet and its variant, where a smaller gap indicates more comparable performance. Using *ground-truth labels*, the accuracy gaps for {ImageNet-Sketch, ImageNet-R, ImageNet-V2, ImageNet-A} are {0.529, 0.564, 0.751, 0.827}.

However, *in practice*, obtaining ground-truth labels for ImageNet variants is often costly or infeasible. We therefore use NAMMD-based DCT to assess the relative distributional closeness of these variants to ImageNet *without labels*, and compare its conclusions with the label-based ranking above. For DCT, the actual inputs are deep image features extracted by the pre-trained ResNet50, rather than labels or accuracy gaps. We use a deep kernel on the extracted features, with the kernel bandwidth selected by the median heuristic on pairwise distances. Following Definition 8, ImageNet is used as the shared source

distribution, so we set $\mathbb{P}_1 = \mathbb{P}_2 = $ ImageNet. We then choose a reference variant $\mathbb{Q}_1$ to define the acceptable shift, with $\epsilon = $ NAMMD$(\mathbb{P}_1, \mathbb{Q}_1; \kappa)$, and test whether another variant $\mathbb{Q}_2$ is farther from ImageNet than $\mathbb{Q}_1$, i.e., whether NAMMD$(\mathbb{P}_2, \mathbb{Q}_2; \kappa) > \epsilon$. In the experiments, $\mathbb{Q}_1$ is selected sequentially from {ImageNet-V2, ImageNet-R, ImageNet-Sketch, slightly perturbed ImageNet}, and $\mathbb{Q}_2$ is selected sequentially from {ImageNet-A, ImageNet-V2, ImageNet-R, ImageNet-Sketch}. We use sample size 150.

Figure 3 shows that NAMMD-based DCT achieves higher test power than MMD-based DCT and is consistent with the closeness relationships indicated by the accuracy-gap ranking. In the testing procedure, *the ground truth about whether $\mathbb{Q}_2$ is actually closer or farther than $\mathbb{Q}_1$ is not known in advance*. While Figure 3 reports results for cases where $\mathbb{Q}_2$ is indeed farther, the complementary case, where $\mathbb{Q}_2$ is not farther, is shown in the type-I error results in Table 9 to Table 12 (Appendix E.4).

In practical scenarios with limited samples, the accuracy gap can be noisy and may not reliably reflect performance differences. In this setting, we therefore use the confidence gap (Eqn. 13 in Appendix D.7) as an alternative *validation* metric. The confidence gap measures the absolute difference in the model's expected prediction confidence between ImageNet and ImageNetv2 for each class, where a smaller gap indicates more similar predictive behavior. This metric is used only to construct and validate the reference ordering, and is not used as an input feature for NAMMD-based DCT.

Specifically, we first compute the confidence gap for each

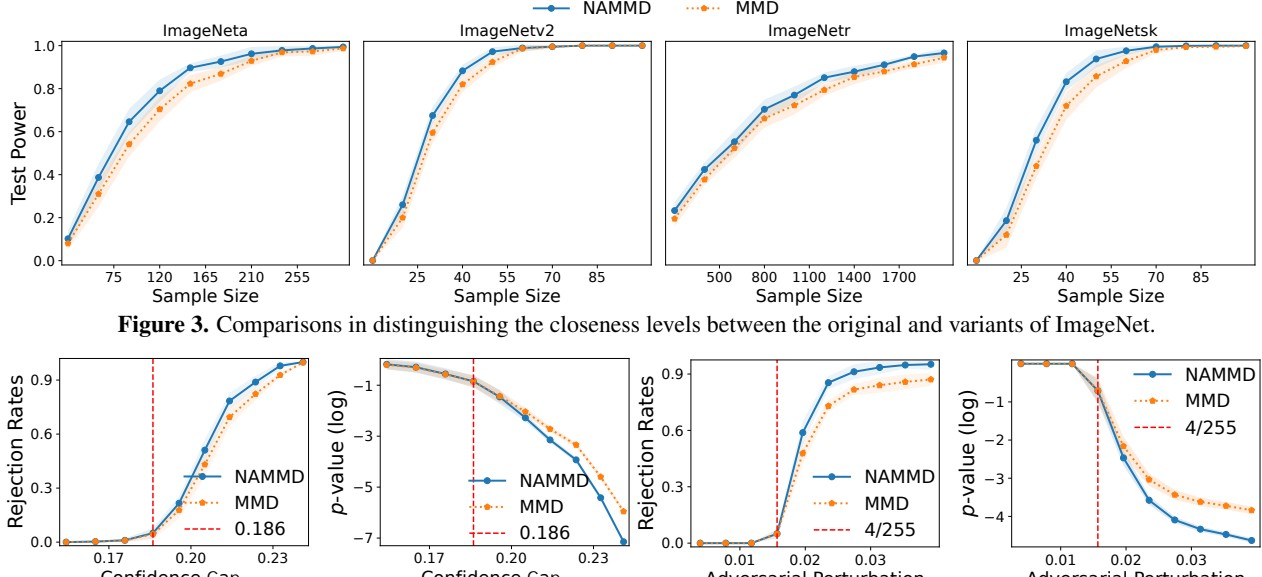

**Figure 3.** Comparisons in distinguishing the closeness levels between the original and variants of ImageNet.

**Figure 4.** Comparison of NAMMD- and MMD-based DCT in detecting the confidence gap between ImageNet and ImageNetv2.

**Figure 5.** Comparison of NAMMD- and MMD-based DCT in detecting adversarial perturbations on the cifar10.

of the 1,000 classes, sort these class-level gaps from small to large, and split them into 10 intervals with 100 classes each. We then compute the mean confidence gap within each interval, yielding the interval-level gaps {0.154, 0.165, 0.176, 0.186, 0.196, 0.205, 0.214, 0.224, 0.233, 0.241}. Following Definition 8, we use the interval with mean gap 0.186 as the reference pair $(\mathbb{P}_1, \mathbb{Q}_1)$, where $\mathbb{P}_1$ and $\mathbb{Q}_1$ are ImageNet and ImageNetv2 distributions for the classes in this interval, respectively. Each interval is then used to form a test pair $(\mathbb{P}_2, \mathbb{Q}_2)$ in the same way.

For DCT, the actual inputs are deep image features rather than labels or confidence gaps. We extract these features using a pre-trained ResNet-50 and use a deep kernel, with the kernel bandwidth selected by the median heuristic on pairwise distances between extracted features. We test each interval-level pair against the reference pair with sample size 150 and report the rejection rates and $p$-values in Figure 4. For intervals with mean confidence gap no larger than 0.186 (left side of the red line), the rejection rates are controlled around the significance level $\alpha = 0.05$. For intervals with mean confidence gap larger than 0.186 (right side of the red line), NAMMD-based DCT yields higher rejection rates and lower $p$-values, which is consistent with the ordering suggested by the confidence-gap validation metric.

Similarly, we validate that our NAMMD can be used to assess the level of adversarial perturbation over the cifar10 dataset. Using ResNet18 as the base model, we apply the PGD attack (Madry et al., 2018) with perturbations $\{i/255\}_{i=1}^{[10]}$. As expected, a larger perturbation generally result in poor model performance on the perturbed cifar10 dataset, indicating that the perturbed cifar10 is farther from

the original cifar10. Following Definition 8, we define the original cifar10 as $\mathbb{P}_1 = \mathbb{P}_2$ and the cifar10 dataset with $4/255$ perturbation as $\mathbb{Q}_1$. We further set $\mathbb{Q}_2$ as the cifar10 after applying perturbations $\{i/255\}_{i=1}^{[10]}$, and perform testing with sample size 1500. It is evident that our NAMMD performs better than MMD and effectively assesses the levels of adversarial perturbations, as shown in Figure 5.

## 6. Conclusion

We proposed a kernel-based method for DCT, which decides whether two distributions are within a prescribed tolerance rather than merely testing equality. Built on an MMD-style discrepancy with asymptotic calibration, our test provides type-I error control and strong power in practice. Experiments on synthetic and real-world benchmarks show reliable performance, and suggest promising directions for improving scalability and sharpening sample-complexity guarantees. Future work includes extending DCT to non-i.i.d. data and developing kernel selection criteria for DCT.

## Acknowledgements

This research was supported by the University of Melbourne's Research Computing Services and the Petascale Campus Initiative. ZJZ and XYT are supported by the Melbourne Research Scholarship and the ARC with grant number DE240101089. LHP is supported by the ARC with grant number LP240100101. MG is supported by ARC with grant number DP240102088. FL is supported by the ARC with grant number DE240101089, LP240100101, DP230101540 and the NSF&CSIRO Responsible AI program with grant

number 2303037.

## Impact Statement

This paper studies *distribution closeness testing*, which asks whether two distributions are within a pre-specified tolerance with statistical guarantees, rather than requiring exact equality. We propose a kernel-based approach built on *norm-adaptive MMD (NAMMD)* to make closeness comparisons more informative across different distribution pairs.

Tolerance-based tests are often more aligned with deployment settings than two-sample tests that flag any difference: small shifts may be acceptable, while larger ones may warrant model adaptation, data collection, or additional validation. By enabling statistically grounded assessment of how close is close enough without labels, NAMMD-based DCT can support dataset curation, monitoring distribution shift across time and domains, and prioritizing when to retrain or adapt models, which may reduce unnecessary retraining and improve reproducibility in distributional comparisons.

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

# Appendix

## Appendix Contents

# A. Notations

In this section, we summarize important notations in Tables 3 and 4.

**Table 3.** Notation (Part 1)

| Symbol | Description |
|---|---|
| **• Basic Notations in Setting** | |
| $\mathcal{X} \subseteq \mathbb{R}^d$ | Instance space / domain of data |
| $\mathbb{P}, \mathbb{Q}, \mathbb{P}_1, \mathbb{Q}_1, \mathbb{P}_2, \mathbb{Q}_2$ | Borel probability measures on $\mathcal{X}$ |
| $\mathbb{P}_n, \mathbb{Q}_n$ | Discrete distributions over domain $Z = \{\boldsymbol{z}_1, \boldsymbol{z}_2, ..., \boldsymbol{z}_n\} \subseteq \mathbb{R}^d$ |
| $\kappa : \mathcal{X} \times \mathcal{X} \to \mathbb{R}$ | Positive-definite kernel, with $0 \le \kappa(\boldsymbol{x}, \boldsymbol{x}') \le K$ for any $\boldsymbol{x}, \boldsymbol{x}' \in \mathcal{X}$ |
| $\mathcal{H}_\kappa$ | Reproducing kernel Hilbert space (RKHS) associated to $\kappa$ |
| $\|\cdot\|_{\mathcal{H}_\kappa}$ | Norm in the RKHS $\mathcal{H}_\kappa$ |
| $\boldsymbol{\mu}_\mathbb{P}, \boldsymbol{\mu}_\mathbb{Q}$ | Kernel mean embeddings of $\mathbb{P}$ and $\mathbb{Q}$ |
| $\sigma^2_{\mathbb{P},\mathbb{Q}}$ | Asymptotic variance of $\sqrt{m}\,\widehat{\mathrm{NAMMD}}(X, Y; \kappa)$ under $\mathbb{P}, \mathbb{Q}$ |
| $\sigma^2_M$ | Asymptotic variance of $\sqrt{m}\,\widehat{\mathrm{MMD}}(X, Y; \kappa)$ under $\mathbb{P}, \mathbb{Q}$ or $\mathbb{P}_1, \mathbb{Q}_1$ |
| $\epsilon$ | Closeness parameter in NAMMD-based DCT with $\boldsymbol{H}_0$ and $\boldsymbol{H}_1$ |
| $\mathcal{N}$ | The standard normal distribution $\mathcal{N}(0, 1)$ |
| $\mathcal{N}_{1-\alpha}$ | The $(1 - \alpha)$-quantile of $\mathcal{N}$ |
| $B$ | The iteration number of permutation test in TST |
| **• Distances** | |
| $\mathrm{TV}(\mathbb{P}_n, \mathbb{Q}_n)$ | Total variation distance between $\mathbb{P}_n$ and $\mathbb{Q}_n$ |
| $\mathrm{MMD}^2(\mathbb{P}, \mathbb{Q}; \kappa)$ | MMD distance between $\mathbb{P}$ and $\mathbb{Q}$ |
| $\mathrm{NAMMD}(\mathbb{P}, \mathbb{Q}; \kappa)$ | NAMMD distance between $\mathbb{P}$ and $\mathbb{Q}$ |
| **• Hypotheses** | |
| $\boldsymbol{H}_0, \boldsymbol{H}_1$ | Null and alternative hypotheses of NAMMD-based DCT with a given $\epsilon$ |
| $\boldsymbol{H}_0^N, \boldsymbol{H}_1^N$ | Hypotheses of MMD-based DCT with $\epsilon^N = \mathrm{NAMMD}(\mathbb{P}_1, \mathbb{Q}_1; \kappa)$ |
| $\boldsymbol{H}_0^M, \boldsymbol{H}_1^M$ | Hypotheses of MMD-based DCT with $\epsilon^M = \mathrm{MMD}^2(\mathbb{P}_1, \mathbb{Q}_1; \kappa)$ |
| $\boldsymbol{H}_0', \boldsymbol{H}_1'$ | Null and alternative hypotheses of TST |
| **• Estimations** | |
| $m$ | Sample size |
| $X, Y$ | Two independent samples of size $m$ from $\mathbb{P}, \mathbb{Q}$ or $\mathbb{P}_2, \mathbb{Q}_2$ |
| $X_{\boldsymbol{\pi}}, Y_{\boldsymbol{\pi}}$ | Permuted two samples |
| $H_{ij}$ | Pairwise function used in the NAMMD estimator |
| $\hat{\sigma}_{X,Y}$ | Plug-in estimator of $\sigma_{\mathbb{P},\mathbb{Q}}$ via $U$-statistics |
| $\widehat{\tau}_\alpha$ | Threshold of NAMMD-based DCT from asymptotic normal estimation |
| $\widehat{\tau}_\alpha'$ | Testing threshold of NAMMD-based TST from permutation test |

**Table 4.** Notation (Part 2)

| Symbol | Description |
|---|---|
| **• Estimations** | |
| $h(X, Y; \kappa)$ | Decision rule of the NAMMD-based DCT |
| $h'(X, Y; \kappa)$ | Decision rule of the NAMMD-based TST |
| $\widehat{\text{NAMMD}}(X, Y; \kappa)$ | Estimator of NAMMD$(\mathbb{P}, \mathbb{Q}; \kappa)$ or NAMMD$(\mathbb{P}_2, \mathbb{Q}_2; \kappa)$ |
| $\widehat{\text{NAMMD}}(X_{\boldsymbol{\pi}}, Y_{\boldsymbol{\pi}}; \kappa)$ | Estimator of NAMMD distance based on permuted samples $X_{\boldsymbol{\pi}}$ and $Y_{\boldsymbol{\pi}}$ |
| $\widehat{\text{MMD}}(X, Y; \kappa)$ | Estimator of MMD$^2(\mathbb{P}, \mathbb{Q}; \kappa)$ or MMD$^2(\mathbb{P}_2, \mathbb{Q}_2; \kappa)$ |
| **• Key Elements in Theoretical Results** | |
| $\tau_{\alpha}^{M}$ | Asymptotic $(1-\alpha)$-quantile of the distribution of the MMD estimator $\sqrt{m}\widehat{\text{MMD}}(X, Y; \kappa)$ under $\mathbb{P}_1$ and $\mathbb{Q}_1$, used in Theorem 9 |
| $\tau_{\alpha}^{N}$ | Asymptotic $(1-\alpha)$-quantile of the distribution of the NAMMD estimator $\sqrt{m}\widehat{\text{NAMMD}}(X, Y; \kappa)$ under $\mathbb{P}_1$ and $\mathbb{Q}_1$, used in Theorem 9 |
| $\epsilon^{M}$ | Closeness parameter for MMD-based DCT with hypotheses $\boldsymbol{H}_0^M$ and $\boldsymbol{H}_1^M$, defined as $\epsilon^M = \text{MMD}^2(\mathbb{P}_1, \mathbb{Q}_1; \kappa)$ and used in Theorem 9 |
| $\epsilon^{N}$ | Closeness parameter for NAMMD-based DCT with hypotheses $\boldsymbol{H}_0^N$ and $\boldsymbol{H}_1^N$, defined as $\epsilon^N = \text{NAMMD}(\mathbb{P}_1, \mathbb{Q}_1; \kappa)$ and used in Theorem 9 |
| $C_1, C_2$ | Constants that bound the sample complexity in Theorem 9 |

# B. Further Details on NAMMD and the NAMMD-based Test

## B.1. Conditions under which NAMMD approaches to 1

Recall that the NAMMD is defined as:

$$
\begin{aligned}
\mathrm{NAMMD}(\mathbb{P}, \mathbb{Q}; \kappa) &= \frac{\|\boldsymbol{\mu}_{\mathbb{P}} - \boldsymbol{\mu}_{\mathbb{Q}}\|_{\mathcal{H}_{\kappa}}^2}{4K - \|\boldsymbol{\mu}_{\mathbb{P}}\|_{\mathcal{H}_{\kappa}}^2 - \|\boldsymbol{\mu}_{\mathbb{Q}}\|_{\mathcal{H}_{\kappa}}^2} \\
&= \frac{\|\boldsymbol{\mu}_{\mathbb{P}}\|_{\mathcal{H}_{\kappa}}^2 + \|\boldsymbol{\mu}_{\mathbb{Q}}\|_{\mathcal{H}_{\kappa}}^2 - 2\langle \boldsymbol{\mu}_{\mathbb{P}}, \boldsymbol{\mu}_{\mathbb{Q}} \rangle_{\mathcal{H}_{\kappa}}}{4K - \|\boldsymbol{\mu}_{\mathbb{P}}\|_{\mathcal{H}_{\kappa}}^2 - \|\boldsymbol{\mu}_{\mathbb{Q}}\|_{\mathcal{H}_{\kappa}}^2} \\
&= \frac{E_{\boldsymbol{x},\boldsymbol{x}' \sim \mathbb{P}^2}[\kappa(\boldsymbol{x}, \boldsymbol{x}')] + E_{\boldsymbol{y},\boldsymbol{y}' \sim \mathbb{Q}^2}[\kappa(\boldsymbol{y}, \boldsymbol{y}')] - 2E_{\boldsymbol{x} \sim \mathbb{P}, \boldsymbol{y} \sim \mathbb{Q}}[\kappa(\boldsymbol{x}, \boldsymbol{y}')]}{4K - E_{\boldsymbol{x},\boldsymbol{x}' \sim \mathbb{P}^2}[\kappa(\boldsymbol{x}, \boldsymbol{x}')] - E_{\boldsymbol{y},\boldsymbol{y}' \sim \mathbb{Q}^2}[\kappa(\boldsymbol{y}, \boldsymbol{y}')]} \,,
\end{aligned}
$$

where the kernel $\kappa(\boldsymbol{x}, \boldsymbol{x}') = \Psi(\boldsymbol{x} - \boldsymbol{x}')$ is positive-definite with $\Psi(\boldsymbol{0}) = K$ and $\Psi(\boldsymbol{x} - \boldsymbol{x}') \leq K$ for all $\boldsymbol{x}, \boldsymbol{x}'$, and $K > 0$.
The value $\mathrm{NAMMD}(\mathbb{P}, \mathbb{Q}; \kappa) \to 1$ (i.e., maximum) is attained when:

- $\|\boldsymbol{\mu}_{\mathbb{P}}\|_{\mathcal{H}_{\kappa}}^2 = \|\boldsymbol{\mu}_{\mathbb{Q}}\|_{\mathcal{H}_{\kappa}}^2 = K$,

- $\langle \boldsymbol{\mu}_{\mathbb{P}}, \boldsymbol{\mu}_{\mathbb{Q}} \rangle_{\mathcal{H}_{\kappa}} \to 0$ (which essentially indicates that the two distributions have disjoint support).

Here, as an example, we consider two Dirac distributions $P$ and $Q$ over distinct supports $\boldsymbol{z}$ and $\boldsymbol{w}$, respectively, and use a Gaussian kernel with parameter $\eta$. In this case:

$$
\|\boldsymbol{\mu}_{\mathbb{P}}\|_{\mathcal{H}_{\kappa}}^2 = \|\boldsymbol{\mu}_{\mathbb{Q}}\|_{\mathcal{H}_{\kappa}}^2 = \Psi(\boldsymbol{0}) = K, \quad \text{and} \quad \langle \boldsymbol{\mu}_{\mathbb{P}}, \boldsymbol{\mu}_{\mathbb{Q}} \rangle_{\mathcal{H}_{\kappa}} = \Psi(\boldsymbol{x} - \boldsymbol{y}) = \exp(-\|\boldsymbol{x} - \boldsymbol{y}\|_2^2 / \eta^2) \,.
$$

As $\eta \to 0$, $\Psi(\boldsymbol{x} - \boldsymbol{y}) \to 0$, causing $\mathrm{NAMMD}(\mathbb{P}, \mathbb{Q}; \kappa) \to 1$.

We also present an empirical example for illustration. Specifically, we consider two Gaussian distributions $\mathbb{P} = \mathcal{N}(-1000, \sigma^2)$ and $\mathbb{Q} = \mathcal{N}(1000, \sigma^2)$, and compute NAMMD using a Gaussian kernel with bandwidth 1. When $\sigma$ is small, the distributions are both sharply concentrated around their respective means and have negligible overlap, effectively resulting in near-disjoint support. This setting closely approximates the idealized condition for maximizing $\mathrm{NAMMD}(\mathbb{P}, \mathbb{Q}; \kappa)$. In the following experiment, we compare the value of $\mathrm{NAMMD}(\mathbb{P}, \mathbb{Q}; \kappa)$ under varying $\sigma$ to empirically verify this behavior.

**Table 5.** Comparison of NAMMD and MMD across $\sigma$.

| $\sigma$ | $10^0$ | $10^{-1}$ | $10^{-2}$ | $10^{-3}$ | |
|---|---|---|---|---|---|
| NAMMD | 0.2679 | $1 - 4.5 \times 10^{-2}$ | $1 - 9.9 \times 10^{-5}$ | $1 - 2.1 \times 10^{-7}$ | |

| $\sigma$ | $10^{-4}$ | $10^{-5}$ | $10^{-6}$ | $10^{-7}$ | $10^{-8}$ |
|---|---|---|---|---|---|
| NAMMD | $1 - 2.1 \times 10^{-8}$ | $1 - 6.4 \times 10^{-10}$ | $1 - 7.0 \times 10^{-12}$ | $1 - 1.1 \times 10^{-16}$ | 1 |

When $\sigma = 10^{-8}$, the kernel value $\kappa(\boldsymbol{x}, \boldsymbol{x}')$ is close to 1 when $\boldsymbol{x}$ and $\boldsymbol{x}'$ are drawn from the same distribution, and close to 0 when they are drawn from different distributions. Consequently, $\mathrm{NAMMD}(\mathbb{P}, \mathbb{Q}; \kappa)$ approaches its maximum value 1.

## B.2. Extension to unequal sample sizes

Recall that

$$
\mathrm{NAMMD}(\mathbb{P}, \mathbb{Q}; \kappa) = \frac{\mathrm{MMD}^2(\mathbb{P}, \mathbb{Q}; \kappa)}{4K - \|\boldsymbol{\mu}_{\mathbb{P}}\|_{\mathcal{H}_{\kappa}}^2 - \|\boldsymbol{\mu}_{\mathbb{Q}}\|_{\mathcal{H}_{\kappa}}^2} \,.
$$

To estimate $\mathrm{NAMMD}(\mathbb{P}, \mathbb{Q}; \kappa)$ from two samples of unequal sizes,

$$
X = \{\boldsymbol{x}_i\}_{i=1}^m \sim \mathbb{P}^m \text{ and } Y = \{\boldsymbol{y}_j\}_{j=1}^n \sim \mathbb{Q}^n \,,
$$

We analyze the behavior of NAMMD estimator by examining its numerator, corresponding to the MMD statistic, and its denominator, which depends on the RKHS norms of $\mathbb{P}$ and $\mathbb{Q}$, separately. The numerator, $\mathrm{MMD}^2(\mathbb{P}, \mathbb{Q}; \kappa)$, can be estimated

using a $U$-statistic. When moving from equal to unequal sample sizes, the estimator changes from a one-sample $U$-statistic to a two-sample statistic as follows

$$U_{m,n} = \frac{1}{\binom{m}{2}\binom{n}{2}} \sum_{1 \le i < i' \le m} \sum_{1 \le j < j' \le n} h(\boldsymbol{x}_i, \boldsymbol{x}_{i'}; \boldsymbol{y}_j, \boldsymbol{y}_{j'}),$$

where

$$h(\boldsymbol{x}_1, \boldsymbol{x}_2; \boldsymbol{y}_1, \boldsymbol{y}_2) = \kappa(\boldsymbol{x}_1, \boldsymbol{x}_2) + \kappa(\boldsymbol{y}_1, \boldsymbol{y}_2) - \kappa(\boldsymbol{x}_1, \boldsymbol{y}_2) - \kappa(\boldsymbol{x}_2, \boldsymbol{y}_1).$$

Despite this modification, both the equal-sample and unequal-sample versions exhibit similar asymptotic properties (Lee, 2019). In particular, when $\mathrm{MMD}^2(\mathbb{P}, \mathbb{Q}; \kappa) = 0$, the statistic converges in distribution to an (often infinite) weighted sum of $\chi^2$ random variables, where the weights are given by the eigenvalues of the covariance operator on $\mathcal{H}_\kappa \to \mathcal{H}_\kappa$.

On the other hand, the estimator of the denominator term

$$4K - \frac{1}{m(m-1)} \sum_{i \ne i'}^{m} \kappa(\boldsymbol{x}_i, \boldsymbol{x}_{i'}) - \frac{1}{n(n-1)} \sum_{j \ne j'}^{n} \kappa(\boldsymbol{y}_j, \boldsymbol{y}_{j'}),$$

remains unchanged regardless of whether the sample sizes are equal or unequal, since the RKHS norms $\|\boldsymbol{\mu}_{\mathbb{P}}\|_{\mathcal{H}_\kappa}^2$ and $\|\boldsymbol{\mu}_{\mathbb{Q}}\|_{\mathcal{H}_\kappa}^2$ can be estimated independently from each sample.

### B.3. Details of Variance Estimator

We adhere to the results of empirical variance estimators provided by Sutherland (2019). For simplicity, we first introduce the uncentred covariance operator as follows:

$$C_X = E_{\boldsymbol{x} \sim \mathbb{P}}[\varphi(\boldsymbol{x}) \otimes \varphi(\boldsymbol{x})],$$

where $\varphi(\cdot)$ is the feature map of the corresponding RKHS $\mathcal{H}_\kappa$.

For simplicity, we define the $m \times m$ matrix $\mathbf{K_{XY}}$ with $(\mathbf{K_{XY}})_{ij} = \kappa(\boldsymbol{x}_i, \boldsymbol{y}_j)$. Let $\tilde{\mathbf{K}}_{\mathbf{XY}}$ be $\mathbf{K_{XY}}$ with diagonals set to zero. In a similar manner, we have $\mathbf{K_{XX}}$ and $\mathbf{K_{YY}}$, and $\tilde{\mathbf{K}}_{\mathbf{XX}}$ and $\tilde{\mathbf{K}}_{\mathbf{YY}}$. Let $\mathbf{1}$ be the $m$-vector of all ones. Denote by $(m)_k := m(m-1)\cdots(m-k+1)$.

Throughout this section, $\hat{\zeta}_1$ and $\hat{\zeta}_2$ denote the population quantities appearing in the asymptotic variance, while $\widehat{\zeta}_1$ and $\widehat{\zeta}_2$ denote their empirical estimators computed from the samples. Specifically, the population quantity $\hat{\zeta}_1$ is given by

$$
\begin{aligned}
\zeta_1 =\ & \langle \boldsymbol{\mu}_X, C_X \boldsymbol{\mu}_X \rangle - \langle \boldsymbol{\mu}_X, \boldsymbol{\mu}_X \rangle^2 + \langle \boldsymbol{\mu}_Y, C_Y \boldsymbol{\mu}_Y \rangle - \langle \boldsymbol{\mu}_Y, \boldsymbol{\mu}_Y \rangle^2 \\
& + \langle \boldsymbol{\mu}_Y, C_X \boldsymbol{\mu}_Y \rangle + \langle \boldsymbol{\mu}_X, C_Y \boldsymbol{\mu}_X \rangle - \langle \boldsymbol{\mu}_X, \boldsymbol{\mu}_Y \rangle^2 - \langle \boldsymbol{\mu}_Y, \boldsymbol{\mu}_X \rangle^2 \\
& - 2\langle \boldsymbol{\mu}_X, C_X \boldsymbol{\mu}_Y \rangle + 2\langle \boldsymbol{\mu}_X, \boldsymbol{\mu}_X \rangle \langle \boldsymbol{\mu}_X, \boldsymbol{\mu}_Y \rangle - 2\langle \boldsymbol{\mu}_Y, C_Y \boldsymbol{\mu}_X \rangle + 2\langle \boldsymbol{\mu}_Y, \boldsymbol{\mu}_Y \rangle \langle \boldsymbol{\mu}_X, \boldsymbol{\mu}_Y \rangle.
\end{aligned}
$$

The corresponding empirical estimator $\widehat{\zeta}_1$ is

$$
\begin{aligned}
\widehat{\zeta}_1 =\ & \frac{1}{(m)_3}\left[\left\|\tilde{\mathbf{K}}_{\mathbf{XX}}\mathbf{1}\right\|^2 - \left\|\tilde{\mathbf{K}}_{\mathbf{XX}}\right\|_F^2\right] - \frac{1}{(m)_4}\left[\left(\mathbf{1}^\top \tilde{\mathbf{K}}_{\mathbf{XX}}\mathbf{1}\right)^2 - 4\left\|\tilde{\mathbf{K}}_{\mathbf{XX}}\mathbf{1}\right\|^2 + 2\left\|\tilde{\mathbf{K}}_{\mathbf{XX}}\right\|_F^2\right] \\
& + \frac{1}{(m)_3}\left[\left\|\tilde{\mathbf{K}}_{\mathbf{YY}}\mathbf{1}\right\|^2 - \left\|\tilde{\mathbf{K}}_{\mathbf{YY}}\right\|_F^2\right] - \frac{1}{(m)_4}\left[\left(\mathbf{1}^\top \tilde{\mathbf{K}}_{\mathbf{YY}}\mathbf{1}\right)^2 - 4\left\|\tilde{\mathbf{K}}_{\mathbf{YY}}\mathbf{1}\right\|^2 + 2\left\|\tilde{\mathbf{K}}_{\mathbf{YY}}\right\|_F^2\right] \\
& + \frac{1}{m^2(m-1)}\left[\left\|\mathbf{K_{XY}}\mathbf{1}\right\|^2 - \left\|\mathbf{K_{XY}}\right\|_F^2\right] + \frac{1}{m^2(m-1)}\left[\left\|\mathbf{K_{XY}}^\top\mathbf{1}\right\|^2 - \left\|\mathbf{K_{XY}}\right\|_F^2\right] \\
& - \frac{2}{m^2(m-1)^2}\left[\left(\mathbf{1}^\top \mathbf{K_{XY}}\mathbf{1}\right)^2 - \left\|\mathbf{K_{XY}}^\top\mathbf{1}\right\|^2 - \left\|\mathbf{K_{XY}}\mathbf{1}\right\|^2 + \left\|\mathbf{K_{XY}}\right\|_F^2\right] \\
& - \frac{2}{m^2(m-1)}\mathbf{1}^\top\tilde{\mathbf{K}}_{\mathbf{XX}}\mathbf{K_{XY}}\mathbf{1} + \frac{2}{m(m)_3}\left[\mathbf{1}^\top\tilde{\mathbf{K}}_{\mathbf{XX}}\mathbf{11}^\top\mathbf{K_{XY}}\mathbf{1} - 2\mathbf{1}^\top\tilde{\mathbf{K}}_{\mathbf{XX}}\mathbf{K_{XY}}\mathbf{1}\right] \\
& - \frac{2}{m^2(m-1)}\mathbf{1}^\top\tilde{\mathbf{K}}_{\mathbf{YY}}\mathbf{K_{XY}}^\top\mathbf{1} + \frac{2}{m(m)_3}\left[\mathbf{1}^\top\tilde{\mathbf{K}}_{\mathbf{YY}}\mathbf{11}^\top\mathbf{K_{XY}}^\top\mathbf{1} - 2\mathbf{1}^\top\tilde{\mathbf{K}}_{\mathbf{YY}}\mathbf{K_{XY}}^\top\mathbf{1}\right].
\end{aligned}
$$

Similarly, the population quantity $\hat{\zeta}_2$ is given by

$$
\begin{aligned}
\zeta_2 = \; & \mathbb{E}\left[\kappa\left(\boldsymbol{x}_1, \boldsymbol{x}_2\right)^2\right] - \langle \boldsymbol{\mu}_X, \boldsymbol{\mu}_X \rangle^2 + \mathbb{E}\left[\kappa\left(\boldsymbol{y}_1, \boldsymbol{y}_2\right)^2\right] \\
& - \langle \boldsymbol{\mu}_Y, \boldsymbol{\mu}_Y \rangle^2 + 2\mathbb{E}\left[\kappa(\boldsymbol{x}, \boldsymbol{y})^2\right] - 2\langle \boldsymbol{\mu}_X, \boldsymbol{\mu}_Y \rangle^2 \\
& - 4\langle \boldsymbol{\mu}_X, C_X \boldsymbol{\mu}_Y \rangle + 4\langle \boldsymbol{\mu}_X, \boldsymbol{\mu}_X \rangle \langle \boldsymbol{\mu}_X, \boldsymbol{\mu}_Y \rangle - 4\langle \boldsymbol{\mu}_Y, C_Y \boldsymbol{\mu}_X \rangle + 4\langle \boldsymbol{\mu}_Y, \boldsymbol{\mu}_Y \rangle \langle \boldsymbol{\mu}_X, \boldsymbol{\mu}_Y \rangle \; .
\end{aligned}
$$

The corresponding empirical estimator $\widehat{\zeta}_2$ is

$$
\begin{aligned}
\widehat{\zeta}_2 = \; & \frac{1}{m(m-1)}\left\|\tilde{\mathbf{K}}_{\mathbf{XX}}\right\|_F^2 - \frac{1}{(m)_4}\left[\left(\mathbf{1}^\top \tilde{\mathbf{K}}_{\mathbf{XX}}\mathbf{1}\right)^2 - 4\left\|\tilde{\mathbf{K}}_{\mathbf{XX}}\mathbf{1}\right\|^2 + 2\left\|\tilde{\mathbf{K}}_{\mathbf{XX}}\right\|_F^2\right] \\
& + \frac{1}{m(m-1)}\left\|\tilde{\mathbf{K}}_{\mathbf{YY}}\right\|_F^2 - \frac{1}{(m)_4}\left[\left(\mathbf{1}^\top \tilde{\mathbf{K}}_{\mathbf{YY}}\mathbf{1}\right)^2 - 4\left\|\tilde{\mathbf{K}}_{\mathbf{YY}}\mathbf{1}\right\|^2 + 2\left\|\tilde{\mathbf{K}}_{\mathbf{YY}}\right\|_F^2\right] \\
& + \frac{2}{m^2}\|\mathbf{K}_{\mathbf{XY}}\|_F^2 - \frac{2}{m^2(m-1)^2}\left[\left(\mathbf{1}^\top \mathbf{K}_{\mathbf{XY}}\mathbf{1}\right)^2 - \|\mathbf{K}_{\mathbf{XY}}^\top\mathbf{1}\|^2 - \|\mathbf{K}_{\mathbf{XY}}\mathbf{1}\|^2 + \|\mathbf{K}_{\mathbf{XY}}\|_F^2\right] \\
& - \frac{4}{m^2(m-1)}\mathbf{1}^\top \tilde{\mathbf{K}}_{\mathbf{XX}}\mathbf{K}_{\mathbf{XY}}\mathbf{1} + \frac{4}{m(m)_3}\left[\mathbf{1}^\top \tilde{\mathbf{K}}_{\mathbf{XX}}\mathbf{1}\mathbf{1}^\top \mathbf{K}_{\mathbf{XY}}\mathbf{1} - 2\mathbf{1}^\top \tilde{\mathbf{K}}_{\mathbf{XX}}\mathbf{K}_{\mathbf{XY}}\mathbf{1}\right] \\
& - \frac{4}{m^2(m-1)}\mathbf{1}^\top \tilde{\mathbf{K}}_{\mathbf{YY}}\mathbf{K}_{\mathbf{XY}}^\top\mathbf{1} + \frac{4}{m(m)_3}\left[\mathbf{1}^\top \tilde{\mathbf{K}}_{\mathbf{YY}}\mathbf{1}\mathbf{1}^\top \mathbf{K}_{\mathbf{XY}}^\top\mathbf{1} - 2\mathbf{1}^\top \tilde{\mathbf{K}}_{\mathbf{YY}}\mathbf{K}_{\mathbf{XY}}^\top\mathbf{1}\right] \; .
\end{aligned}
$$

where $\langle \cdot, \cdot \rangle$ denotes the inner product in RKHS $\mathcal{H}_\kappa$. Here, we denote by

$$
\boldsymbol{\mu}_X = \boldsymbol{\mu}_\mathbb{P} = E_{\boldsymbol{x}\sim\mathbb{P}}[\kappa(\cdot, \boldsymbol{x})] \quad \text{and} \quad \boldsymbol{\mu}_Y = \boldsymbol{\mu}_\mathbb{Q} = E_{\boldsymbol{y}\sim\mathbb{Q}}[\kappa(\cdot, \boldsymbol{y})] \; .
$$

**Convergence of the estimators.** Having established that the constituent estimators in $\widehat{\zeta}_1$ and $\widehat{\zeta}_2$ are unbiased (Sutherland, 2019), we now prove their convergence by analyzing each constituent term separately with bounded kernel $\kappa(\cdot, \cdot) \leq K$, as follows.

- The term $\langle \boldsymbol{\mu}_X, C_X \boldsymbol{\mu}_X \rangle$ is estimated by

$$
A = \frac{1}{(n)_3}\sum_i \sum_{j\neq\ell}\sum_{\ell\notin\{i,j\}}\kappa(\boldsymbol{x}_i, \boldsymbol{x}_j)\kappa(\boldsymbol{x}_i, \boldsymbol{x}_\ell) \; .
$$

  It is evident that

$$
|A - \langle \boldsymbol{\mu}_X, C_X \boldsymbol{\mu}_X \rangle| \leq |A - B| + |B - \langle \boldsymbol{\mu}_X, C_X \boldsymbol{\mu}_X \rangle| \; ,
$$

  with

$$
\begin{aligned}
B &= \frac{1}{n}\sum_i E_{\boldsymbol{x}}[\kappa(\boldsymbol{x}_i, \boldsymbol{x})]E_{\boldsymbol{x}}[\kappa(\boldsymbol{x}_i, \boldsymbol{x})] \\
&= \frac{1}{n}\sum_i \langle \kappa(\boldsymbol{x}_i, \cdot), \boldsymbol{\mu}_X \rangle^2 \; .
\end{aligned}
$$

  As we can see, $B$ is a $U$-statistic. By the large deviation bound (Theorem 11) for $U$-statistic, we have that

$$
B \xrightarrow{p} \langle \boldsymbol{\mu}_X, C_X \boldsymbol{\mu}_X \rangle \; .
$$

  For the term $|A - B|$, we have that

$$
\begin{aligned}
&|A - B| \\
&= \frac{1}{n}\sum_i \left[\frac{1}{(n-1)(n-2)}\sum_{j\neq\ell}\sum_{\ell\notin\{i,j\}}\kappa(\boldsymbol{x}_i, \boldsymbol{x}_j)\kappa(\boldsymbol{x}_i, \boldsymbol{x}_\ell) - E_{\boldsymbol{x}}[\kappa(\boldsymbol{x}_i, \boldsymbol{x})]E_{\boldsymbol{x}}[\kappa(\boldsymbol{x}_i, \boldsymbol{x})]\right] \; ,
\end{aligned}
$$

where the term $\frac{1}{(n-1)(n-2)}\sum_{j\neq\ell}\sum_{\ell\notin\{i,j\}}\kappa(\boldsymbol{x}_i,\boldsymbol{x}_j)\kappa(\boldsymbol{x}_i,\boldsymbol{x}_\ell)$ can also be viewed as a $U$-statistic, and it follows that

$$\frac{1}{(n-1)(n-2)}\sum_{j\neq\ell}\sum_{\ell\notin\{i,j\}}\kappa(\boldsymbol{x}_i,\boldsymbol{x}_j)\kappa(\boldsymbol{x}_i,\boldsymbol{x}_\ell)\xrightarrow{p}E_{\boldsymbol{x}}[\kappa(\boldsymbol{x}_i,\boldsymbol{x})]E_{\boldsymbol{x}}[\kappa(\boldsymbol{x}_i,\boldsymbol{x})]\ ,$$

by the large deviation bound (Theorem 11) for $U$-statistic.

Combining these results, we have that

$$\frac{1}{(n)_3}\sum_i\sum_{j\neq\ell}\sum_{\ell\notin\{i,j\}}\kappa(\boldsymbol{x}_i,\boldsymbol{x}_j)\kappa(\boldsymbol{x}_i,\boldsymbol{x}_\ell)\xrightarrow{p}\langle\boldsymbol{\mu}_X,C_X\boldsymbol{\mu}_X\rangle\ .$$

- The term $\langle\boldsymbol{\mu}_X,\boldsymbol{\mu}_X\rangle^2$ is estimated by

$$A=\frac{1}{(n)_4}\sum_i\sum_{j\neq i}\kappa(\boldsymbol{x}_i,\boldsymbol{x}_j)\sum_{a\notin\{i,j\}}\sum_{b\notin\{i,j,a\}}\kappa(\boldsymbol{x}_a,\boldsymbol{x}_b)\ .$$

It is evident that

$$\left|A-\langle\boldsymbol{\mu}_X,\boldsymbol{\mu}_X\rangle^2\right|\leq|A-B|+\left|B-\langle\boldsymbol{\mu}_X,\boldsymbol{\mu}_X\rangle^2\right|\ ,$$

with

$$B=\frac{1}{n(n-1)}\sum_i\sum_{j\neq i}\kappa(\boldsymbol{x}_i,\boldsymbol{x}_j)E_{\boldsymbol{x},\boldsymbol{x}'}[\kappa(\boldsymbol{x},\boldsymbol{x}')]\ .$$

Building on this, we can prove that

$$\frac{1}{(n)_4}\sum_i\sum_{j\neq i}\kappa(\boldsymbol{x}_i,\boldsymbol{x}_j)\sum_{a\notin\{i,j\}}\sum_{b\notin\{i,j,a\}}\kappa(\boldsymbol{x}_a,\boldsymbol{x}_b)\xrightarrow{p}\langle\boldsymbol{\mu}_X,\boldsymbol{\mu}_X\rangle^2\ ,$$

using a similar argument as in the convergence proof for the estimator of $\langle\boldsymbol{\mu}_X,C_X\boldsymbol{\mu}_X\rangle$.

- The term $\langle\boldsymbol{\mu}_Y,C_X\boldsymbol{\mu}_Y\rangle$ is estimated by

$$A=\frac{1}{n^2(n-1)}\sum_i\sum_j\sum_{\ell\neq j}\kappa(\boldsymbol{x}_i,\boldsymbol{y}_j)\kappa(\boldsymbol{x}_i,\boldsymbol{y}_\ell)\ .$$

It is evident that

$$|A-\langle\boldsymbol{\mu}_Y,C_X\boldsymbol{\mu}_Y\rangle|\leq|A-B|+|B-\langle\boldsymbol{\mu}_Y,C_X\boldsymbol{\mu}_Y\rangle|\ ,$$

with

$$B=\frac{1}{n}\sum_i E_{\boldsymbol{y}}[\kappa(\boldsymbol{x}_i,\boldsymbol{y})]E_{\boldsymbol{y}}[\kappa(\boldsymbol{x}_i,\boldsymbol{y})]\ .$$

Building on this, we can prove that

$$\frac{1}{n^2(n-1)}\sum_i\sum_j\sum_{\ell\neq j}\kappa(\boldsymbol{x}_i,\boldsymbol{y}_j)\kappa(\boldsymbol{x}_i,\boldsymbol{y}_\ell)\xrightarrow{p}\langle\boldsymbol{\mu}_Y,C_X\boldsymbol{\mu}_Y\rangle\ ,$$

using a similar argument as in the convergence proof for the estimator of $\langle\boldsymbol{\mu}_X,C_X\boldsymbol{\mu}_X\rangle$.

- The term $\langle\boldsymbol{\mu}_X,C_X\boldsymbol{\mu}_Y\rangle$ is estimated by

$$A=\frac{1}{n^2(n-1)}\sum_i\sum_{j\neq i}\sum_\ell\kappa(\boldsymbol{x}_i,\boldsymbol{x}_j)\kappa(\boldsymbol{x}_i,\boldsymbol{y}_\ell)\ .$$

It is evident that

$$|A-\langle\boldsymbol{\mu}_X,C_X\boldsymbol{\mu}_Y\rangle|\leq|A-B|+|B-\langle\boldsymbol{\mu}_X,C_X\boldsymbol{\mu}_Y\rangle|\ ,$$

with

$$B = \frac{1}{n} \sum_i E_{\boldsymbol{x}}[\kappa(\boldsymbol{x}_i, \boldsymbol{x})] E_{\boldsymbol{y}}[\kappa(\boldsymbol{x}_i, \boldsymbol{y})] .$$

Building on this, we can prove that

$$\frac{1}{n^2(n-1)} \sum_i \sum_{j \neq i} \sum_\ell \kappa(\boldsymbol{x}_i, \boldsymbol{x}_j)\kappa(\boldsymbol{x}_i, \boldsymbol{y}_\ell) \xrightarrow{p} \langle \boldsymbol{\mu}_X, C_X \boldsymbol{\mu}_Y \rangle ,$$

using a similar argument as in the convergence proof for the estimator of $\langle \boldsymbol{\mu}_X, C_X \boldsymbol{\mu}_X \rangle$.

- The term $\langle \boldsymbol{\mu}_X, \boldsymbol{\mu}_X \rangle \langle \boldsymbol{\mu}_X, \boldsymbol{\mu}_Y \rangle$ is estimated by

$$A = \frac{1}{n(n)_3} \sum_i \sum_{j \neq i} \kappa(\boldsymbol{x}_i, \boldsymbol{x}_j) \sum_{\ell \notin \{i,j\}} \sum_a \kappa(\boldsymbol{x}_\ell, \boldsymbol{y}_a) .$$

It is evident that

$$|A - \langle \boldsymbol{\mu}_X, \boldsymbol{\mu}_X \rangle \langle \boldsymbol{\mu}_X, \boldsymbol{\mu}_Y \rangle| \leq |A - B| + |B - \langle \boldsymbol{\mu}_X, \boldsymbol{\mu}_X \rangle \langle \boldsymbol{\mu}_X, \boldsymbol{\mu}_Y \rangle| ,$$

with

$$B = \frac{1}{n(n-1)} \sum_i \sum_{j \neq i} \kappa(\boldsymbol{x}_i, \boldsymbol{x}_j) E_{\boldsymbol{x}, \boldsymbol{y}}[\kappa(\boldsymbol{x}, \boldsymbol{y})] .$$

Building on this, we can prove that

$$\frac{1}{n(n)_3} \sum_i \sum_{j \neq i} \kappa(\boldsymbol{x}_i, \boldsymbol{x}_j) \sum_{\ell \notin \{i,j\}} \sum_a \kappa(\boldsymbol{x}_\ell, \boldsymbol{y}_a) \xrightarrow{p} \langle \boldsymbol{\mu}_X, \boldsymbol{\mu}_X \rangle \langle \boldsymbol{\mu}_X, \boldsymbol{\mu}_Y \rangle ,$$

using a similar argument as in the convergence proof for the estimator of $\langle \boldsymbol{\mu}_X, C_X \boldsymbol{\mu}_X \rangle$.

- The term $\langle \boldsymbol{\mu}_X, \boldsymbol{\mu}_Y \rangle^2$ is estimated by

$$A = \frac{1}{n^2} \sum_{i,j} \kappa(\boldsymbol{x}_i, \boldsymbol{y}_j) \frac{1}{(n-1)^2} \sum_{i' \neq i} \sum_{j' \neq j} \kappa(\boldsymbol{x}_{i'}, \boldsymbol{y}_{j'}) .$$

It is evident that

$$\left| A - \langle \boldsymbol{\mu}_X, \boldsymbol{\mu}_Y \rangle^2 \right| \leq |A - B| + \left| B - \langle \boldsymbol{\mu}_X, \boldsymbol{\mu}_Y \rangle^2 \right| ,$$

with

$$B = \frac{1}{n^2} \sum_{i,j} \kappa(\boldsymbol{x}_i, \boldsymbol{y}_j) E_{\boldsymbol{x}, \boldsymbol{y}}[\kappa(\boldsymbol{x}, \boldsymbol{y})] .$$

Building on this, we can prove that

$$\frac{1}{n^2} \sum_{i,j} \kappa(\boldsymbol{x}_i, \boldsymbol{y}_j) \frac{1}{(n-1)^2} \sum_{i' \neq i} \sum_{j' \neq j} \kappa(\boldsymbol{x}_{i'}, \boldsymbol{y}_{j'}) \xrightarrow{p} \langle \boldsymbol{\mu}_X, \boldsymbol{\mu}_Y \rangle^2 ,$$

using a similar argument as in the convergence proof for the estimator of $\langle \boldsymbol{\mu}_X, C_X \boldsymbol{\mu}_X \rangle$.

- The term $\mathbb{E}\left[ \kappa(\boldsymbol{x}_1, \boldsymbol{x}_2)^2 \right]$ is estimated by

$$\frac{1}{n(n-1)} \sum_{i \neq j} \kappa(\boldsymbol{x}_i, \boldsymbol{x}_j)^2 ,$$

which can also be viewed as a $U$-statistic, and it follows that

$$\frac{1}{n(n-1)} \sum_{i \neq j} \kappa(\boldsymbol{x}_i, \boldsymbol{x}_j)^2 \xrightarrow{p} \mathbb{E}\left[ \kappa(\boldsymbol{x}_1, \boldsymbol{x}_2)^2 \right] ,$$

by the large deviation bound (Theorem 11) for $U$-statistic.

Based on the convergence of each constituent term, it follows that the empirical estimators $\widehat{\zeta}_1$ and $\widehat{\zeta}_2$ converge in probability to their respective population quantities $\hat{\zeta}_1$ and $\hat{\zeta}_2$ by an application of the *continuous mapping theorem.*

## B.4. Relevant Works

A well-known class of two-sample testing (TST) constructs kernel embeddings for each distribution and then test the differences between these embeddings (Zaremba et al., 2013; Wynne & Duncan, 2022; Shekhar et al., 2022; Scetbon & Varoquaux, 2019; Tian et al., 2025; Zhou et al., 2025b). Another relevant approach assesses the differences between distributions with classification performance (Golland & Fischl, 2003; Lopez-Paz & Oquab, 2017; Cheng & Cloninger, 2019; Cai et al., 2020; Kim et al., 2021; Jang et al., 2022; Cheng & Cloninger, 2022; Kübler et al., 2022; Hediger et al., 2022). Kernel-based MMD has been one of the most important statistic for TST, which includes popular classifier-based TST approaches as a special case (Liu et al., 2020).

Previous distribution closeness testing approaches primarily study sample-complexity guarantees for sub-linear algorithms, and often focus on total variation over discrete distributions with finite support (Batu et al., 2000; 2013; Bhattacharya & Valiant, 2015; Acharya et al., 2015; Diakonikolas et al., 2015). Other notions of closeness have also been considered, including $\ell_2$ distance (Diakonikolas & Kane, 2016; Chan et al., 2014; Luo et al., 2024), entropy (Valiant, 2008), and probability difference (Li, 1996; Blum & Hu, 2018). In statistics, related perspectives include equivalence or similarity testing, such as the nonparametric Mallows-distance test of Munk & Czado (1998) for one-dimensional continuous distributions. Optimal transport also provides an important class of distributional discrepancies, including Wasserstein distances, and has been widely used for distribution comparison and two-sample testing (Peyré & Cuturi, 2019; Ramdas et al., 2017). In comparison, our focus is on developing a kernel-based DCT framework for complex and high-dimensional data, where MMD yields tractable estimators and asymptotic calibration.

Permutation tests are widely used in statistics for testing equality of distributions, providing a finite-sample guarantee on the type-I error under the null hypothesis that assumes $\mathbb{P} = \mathbb{Q}$ (Hoeffding, 1952; Hemerik & Goeman, 2018; Hall & Tajvidi, 2002; Kim et al., 2022). For DCT with null hypothesis $\boldsymbol{H}_0 : \text{NAMMD}(\mathbb{P}, \mathbb{Q}; \kappa) \leq \epsilon$ and $\epsilon \in (0, 1)$, the empirical estimator of our NAMMD distance, i.e., $\text{NAMMD}(\mathbb{P}, \mathbb{Q}; \kappa) = \epsilon$, has an asymptotic Gaussian distribution as shown in Lemma 2. Consequently, the testing threshold can be easily estimated as the $(1 - \alpha)$-quantile of this asymptotic Gaussian distribution, following (Zhou et al., 2023; Shekhar et al., 2022; Scetbon & Varoquaux, 2019).

Some approaches select kernels in a supervised manner using held-out data (Jitkrittum et al., 2016; Sutherland et al., 2017; Zhou et al., 2025a), while others rely on unsupervised methods, such as the median heuristic (Gretton et al., 2012a), or adaptively combine multiple kernels (Schrab et al., 2022; Biggs et al., 2023). Our NAMMD is compatible with these methods; for instance, the kernel can be selected by maximizing the $t$-statistic for test power estimation derived from Lemma 2 (details are provided in Appendix B.5). However, these approaches are primarily designed for distinguishing between a fixed distribution pair in two-sample testing. It remains an open question and an important future work to select an optimal global kernel for distribution closeness testing with multiple distribution pairs.

## B.5. Details of Optimization for Kernel Selecting

---
**Algorithm 1** Kernel Selection

**Input**: Two samples $X$ and $Y$, a kernel $\kappa$, step size $\eta$, iteration number $N$
**Output**: Two samples $X$ and $Y$
1: **for** $\ell = 1, 2, \cdots, N$ **do**
2:     Calculate the estimator $\widehat{\text{NAMMD}}(X, Y; \kappa)/\sigma_{X,Y}$ according to Eqn. 6
3:     Calculate gradient $\nabla \cdot \left( \widehat{\text{NAMMD}}(X, Y; \kappa)/\sigma_{X,Y} \right)$
4:     Gradient ascend with step size $\eta$ by the Adam method
5: **end for**

---

Recall Lemma 2, if $\text{NAMMD}(\mathbb{P}, \mathbb{Q}; \kappa) = \epsilon$ with $\epsilon \in (0, 1)$, we have

$$\sqrt{m}(\widehat{\text{NAMMD}}(X, Y; \kappa) - \epsilon) \xrightarrow{d} \mathcal{N}(0, \sigma_{\mathbb{P},\mathbb{Q}}^2) \, ,$$

where $\sigma_{\mathbb{P},\mathbb{Q}} = \sqrt{4E[H_{1,2}H_{1,3}] - 4(E[H_{1,2}])^2}/(4K - \|\boldsymbol{\mu}_{\mathbb{P}}\|_{\mathcal{H}_\kappa}^2 - \|\boldsymbol{\mu}_{\mathbb{Q}}\|_{\mathcal{H}_\kappa}^2)$, and the expectation are taken over $\boldsymbol{x}_1, \boldsymbol{x}_2, \boldsymbol{x}_3 \sim \mathbb{P}^3$ and $\boldsymbol{y}_1, \boldsymbol{y}_2, \boldsymbol{y}_3 \sim \mathbb{Q}^3$.

We can find the approximate test power by using the asymptotic testing threshold $\tau_\alpha^N$ as follows:

$$\Pr\left(m\widehat{\mathrm{NAMMD}}(X, Y; \kappa) \geq \tau_\alpha^N\right) - \Phi\left(\frac{m\mathrm{NAMMD}(\mathbb{P}, \mathbb{Q}; \kappa) - \tau_\alpha^N}{\sqrt{m}\sigma_{\mathbb{P},\mathbb{Q}}}\right) \to 0.$$

It is evident that maximizing the test power is equivalent to optimizing the following term

$$\frac{\mathrm{NAMMD}(\mathbb{P}, \mathbb{Q}; \kappa)}{\sigma_{\mathbb{P},\mathbb{Q}}} = \frac{\mathrm{MMD}^2(\mathbb{P}, \mathbb{Q}; \kappa)}{\sqrt{4E[H_{1,2}H_{1,3}] - 4(E[H_{1,2}])^2}}.$$

Recall that

$$\widehat{\mathrm{NAMMD}}(X, Y; \kappa) = \sum_{i \neq j} H_{i,j} / \sum_{i \neq j}(4K - \kappa(\boldsymbol{x}_i, \boldsymbol{x}_j) - \kappa(\boldsymbol{y}_i, \boldsymbol{y}_j)),$$

with $H_{i,j} = \kappa(\boldsymbol{x}_i, \boldsymbol{x}_j) + \kappa(\boldsymbol{y}_i, \boldsymbol{y}_j) - \kappa(\boldsymbol{x}_i, \boldsymbol{y}_j) - \kappa(\boldsymbol{y}_i, \boldsymbol{x}_j)$ and

$$\sigma_{X,Y} = \frac{\sqrt{((4m - 8)\hat{\zeta}_1 + 2\hat{\zeta}_2)/(m - 1)}}{(m^2 - m)^{-1}\sum_{i \neq j} 4K - \kappa(\boldsymbol{x}_i, \boldsymbol{x}_j) - \kappa(\boldsymbol{y}_i, \boldsymbol{y}_j)},$$

where $\hat{\zeta}_1$ and $\hat{\zeta}_2$ are standard variance components of the MMD (Serfling, 2009; Sutherland, 2019). The details of the $\hat{\zeta}_1$ and $\hat{\zeta}_2$ are provided in Appendix B.3.

We have the empirical $t$-statistic for test power estimation as follows

$$\frac{\widehat{\mathrm{NAMMD}}(X, Y; \kappa)}{\sigma_{X,Y}} = \frac{\widehat{\mathrm{MMD}}(X, Y; \kappa)}{\sqrt{((4m - 8)\hat{\zeta}_1 + 2\hat{\zeta}_2)/(m - 1)}}, \tag{6}$$

It is evident that the $t$-statistic for test power estimation of NAMMD is equal to the $t$-statistic for test power estimation of MMD (Sutherland et al., 2017). We take gradient method (Boyd & Vandenberghe, 2004) for the optimization of Eqn. 6. Algorithm 1 presents the detailed description on optimization.

### B.6. Methodology of NAMMD-based Two-Sample Test

Although the NAMMD is specially designed for DCT, it is still a statistic to measure the distributional discrepancy between two distributions. Thus, it is interesting to see how it performs in two-sample testing (TST) scenarios. In TST, we aim to assess the equivalence between distributions $\mathbb{P}$ and $\mathbb{Q}$ with null and alternative hypotheses as follows

$$\boldsymbol{H}_0' : \mathbb{P} = \mathbb{Q} \quad \text{and} \quad \boldsymbol{H}_1' : \mathbb{P} \neq \mathbb{Q}.$$

Following MMD-based TST (Sutherland et al., 2017), we implement our NAMMD-based TST via a permutation test, which estimates the null distribution by repeatedly re-computing the estimator with samples randomly reassigned to $X$ or $Y$. Specifically, denote by $B$ the iteration number of permutation test. Let $\boldsymbol{\Pi}_{2m}$ be the set of all possible permutations of $\{1, \ldots, 2m\}$ over the pooled sample $Z = \{\boldsymbol{x}_1, \ldots, \boldsymbol{x}_m, \boldsymbol{y}_1, \ldots, \boldsymbol{y}_m\} = \{\boldsymbol{z}_1, \ldots, \boldsymbol{z}_m, \boldsymbol{z}_{m+1}, \ldots, \boldsymbol{z}_{2m}\}$. In $b$-th iteration $(b \in [B])$, we generate a permutation $\boldsymbol{\pi} = (\pi_1, \ldots, \pi_{2m}) \in \boldsymbol{\Pi}_{2m}$ and then calculate the empirical estimator of NAMMD statistic as follows

$$T_b = \widehat{\mathrm{NAMMD}}(X_{\boldsymbol{\pi}}, Y_{\boldsymbol{\pi}}; \kappa),$$

where $X_{\boldsymbol{\pi}} = \{\boldsymbol{z}_{\pi_1}, \boldsymbol{z}_{\pi_2}, ..., \boldsymbol{z}_{\pi_m}\}$ and $Y_{\boldsymbol{\pi}} = \{\boldsymbol{z}_{\pi_{m+1}}, \boldsymbol{z}_{\pi_{m+2}}, ..., \boldsymbol{z}_{\pi_{2m}}\}$.

During such process, we obtain $B$ statistics $T_1, T_2, ..., T_B$ and introduce the testing threshold for the null hypothesis $\boldsymbol{H}_0 : \mathrm{NAMMD}(\mathbb{P}, \mathbb{Q}; \kappa) = 0$ as follows

$$\hat{\tau}_\alpha' = \arg\min_\tau \left\{ \sum_{b=1}^B \frac{\mathbb{I}[T_b \leq \tau]}{B} \geq 1 - \alpha \right\}.$$

Finally, we have the following test with the testing threshold $\tau_\alpha$ as follows

$$h'(X, Y; \kappa) = \mathbb{I}[\widehat{\mathrm{NAMMD}}(X, Y; \kappa) > \hat{\tau}_\alpha'].$$

## C. Detailed Proofs of Theoretical Results

To begin, we define the concept of the $U$-statistic, which is a key statistical tool.

**Definition 10.** (Serfling, 2009) Let $h(\boldsymbol{x}_1, \boldsymbol{x}_2, \ldots, \boldsymbol{x}_r)$ be a symmetric function of $r$ arguments. Suppose we have a random sample $\boldsymbol{x}_1, \boldsymbol{x}_2, \ldots, \boldsymbol{x}_m$ from some distribution. The U-statistic is given by:

$$U_m = \binom{m}{r}^{-1} \sum_{1 \leq i_1 < i_2 < \cdots < i_r \leq m} h(\boldsymbol{x}_{i_1}, \boldsymbol{x}_{i_2}, ..., \boldsymbol{x}_{i_r}) \, .$$

Here, $\binom{m}{r}$ is the number of ways to choose $r$ distinct indices from $m$, i.e., the binomial coefficient, and the summation is taken over all possible $r$-tuples from the sample.

We further present the large deviation for U-statistic as follows.

**Theorem 11.** *(Hoeffding, 1963) If the function $h$ is bounded, $a \leq h(\boldsymbol{x}_{i_1}, \boldsymbol{x}_{i_2}, ..., \boldsymbol{x}_{i_r}) \leq b$, we have*

$$\Pr(|U_m - \theta| \geq t) \leq 2 \exp\left(-2\lfloor m/r \rfloor t^2/(b-a)^2\right) ,$$

*where $\theta = E[h(\boldsymbol{x}_{i_1}, \boldsymbol{x}_{i_2}, ..., \boldsymbol{x}_{i_r})]$.*

### C.1. Detailed Proofs of Lemma 2

We begin with the empirical estimator of MMD as

$$\widehat{\mathrm{MMD}}^2(X, Y; \kappa) = \frac{1}{m(m-1)} \sum_{i \neq j} [\kappa(\boldsymbol{x}_i, \boldsymbol{x}_j) + \kappa(\boldsymbol{y}_i, \boldsymbol{y}_j) - \kappa(\boldsymbol{x}_i, \boldsymbol{y}_j) - \kappa(\boldsymbol{y}_i, \boldsymbol{x}_j)] \, .$$

Given this, we introduce a useful lemma as follows.

**Lemma 12.** *Assume that $X = \{\boldsymbol{x}_i\}_{i=1}^m$ and $Y = \{\boldsymbol{y}_i\}_{i=1}^m$ are independent i.i.d. samples from $\mathbb{P}$ and $\mathbb{Q}$, respectively. Suppose that $\mathbb{P} \neq \mathbb{Q}$, the kernel $\kappa$ is measurable and bounded, and the non-degenerate variance $\sigma_M^2 > 0$. Then a standard central limit theorem for non-degenerate $U$-statistics holds (Serfling, 2009, Section 5.5.1):*

$$\sqrt{m}\left(\widehat{\mathrm{MMD}}^2(X, Y; \kappa) - \mathrm{MMD}^2(\mathbb{P}, \mathbb{Q}; \kappa)\right) \xrightarrow{d} \mathcal{N}\left(0, \sigma_M^2\right) ,$$

$$\sigma_M^2 := 4E[H_{1,2} H_{1,3}] - 4(E[H_{1,2}])^2 ,$$

*where $H_{i,j} = \kappa(\boldsymbol{x}_i, \boldsymbol{x}_j) + \kappa(\boldsymbol{y}_i, \boldsymbol{y}_j) - \kappa(\boldsymbol{x}_i, \boldsymbol{y}_j) - \kappa(\boldsymbol{y}_i, \boldsymbol{x}_j)$, and the expectations are taken with respect to $\boldsymbol{x}_1, \boldsymbol{x}_2, \boldsymbol{x}_3 \overset{i.i.d.}{\sim} \mathbb{P}$ and $\boldsymbol{y}_1, \boldsymbol{y}_2, \boldsymbol{y}_3 \overset{i.i.d.}{\sim} \mathbb{Q}$.*

We now present the proofs of Lemma 2 as follows.

*Proof.* Recall the empirical estimator of our NAMMD distance

$$
\begin{aligned}
\widehat{\mathrm{NAMMD}}(X, Y; \kappa) &= \frac{\sum_{i \neq j}[\kappa(\boldsymbol{x}_i, \boldsymbol{x}_j) + \kappa(\boldsymbol{y}_i, \boldsymbol{y}_j) - \kappa(\boldsymbol{x}_i, \boldsymbol{y}_j) - \kappa(\boldsymbol{y}_i, \boldsymbol{x}_j)]}{\sum_{i \neq j}[4K - \kappa(\boldsymbol{x}_i, \boldsymbol{x}_j) - \kappa(\boldsymbol{y}_i, \boldsymbol{y}_j)]} \\
&= \frac{\widehat{\mathrm{MMD}}^2(X, Y; \kappa)}{\frac{1}{m(m-1)} \sum_{i \neq j}[4K - \kappa(\boldsymbol{x}_i, \boldsymbol{x}_j) - \kappa(\boldsymbol{y}_i, \boldsymbol{y}_j)]} \, .
\end{aligned}
$$

As a U-statistic, by the large deviation bound (Theorem 11), it is easy to see that,

$$\frac{1}{m(m-1)} \sum_{i \neq j}[4K - \kappa(\boldsymbol{x}_i, \boldsymbol{x}_j) - \kappa(\boldsymbol{y}_i, \boldsymbol{y}_j)] \xrightarrow{p} 4K - \|\boldsymbol{\mu}_\mathbb{P}\|_{\mathcal{H}_\kappa}^2 - \|\boldsymbol{\mu}_\mathbb{Q}\|_{\mathcal{H}_\kappa}^2 ,$$

where $\xrightarrow{p}$ denotes convergence in probability.

If $\text{NAMMD}(\mathbb{P}, \mathbb{Q}; \kappa) = \epsilon > 0$, we have $\text{MMD}^2(\mathbb{P}, \mathbb{Q}; \kappa) > 0$. Furthermore, from Lemma 12, we have, for $\mathbb{P} \neq \mathbb{Q}$,

$$\sqrt{m} \left( \widehat{\text{MMD}}^2(X, Y; \kappa) - \text{MMD}^2(\mathbb{P}, \mathbb{Q}; \kappa) \right) \xrightarrow{d} \mathcal{N}(0, \sigma_M^2) .$$

We use the first-order variance simplification for the ratio statistic.[6] Then, by applying the delta method for $U$-statistics together with Slutsky's theorem (Papoulis & Pillai, 2002), we obtain

$$\sqrt{m} \left( \frac{\widehat{\text{MMD}}^2(X, Y; \kappa)}{\frac{1}{m(m-1)} \sum_{i \neq j} [4K - \kappa(\boldsymbol{x}_i, \boldsymbol{x}_j) - \kappa(\boldsymbol{y}_i, \boldsymbol{y}_j)]} - \frac{\text{MMD}^2(\mathbb{P}, \mathbb{Q}; \kappa)}{4K - \|\boldsymbol{\mu}_{\mathbb{P}}\|_{\mathcal{H}_\kappa}^2 - \|\boldsymbol{\mu}_{\mathbb{Q}}\|_{\mathcal{H}_\kappa}^2} \right) \xrightarrow{d} \mathcal{N} \left( 0, \frac{\sigma_M^2}{\left( 4K - \|\boldsymbol{\mu}_{\mathbb{P}}\|_{\mathcal{H}_\kappa}^2 - \|\boldsymbol{\mu}_{\mathbb{Q}}\|_{\mathcal{H}_\kappa}^2 \right)^2} \right) .$$

Recalling the definition of NAMMD, we have

$$\sqrt{m} \left( \widehat{\text{NAMMD}}(X, Y; \kappa) - \text{NAMMD}(\mathbb{P}, \mathbb{Q}; \kappa) \right) \xrightarrow{d} \mathcal{N} \left( 0, \frac{\sigma_M^2}{\left( 4K - \|\boldsymbol{\mu}_{\mathbb{P}}\|_{\mathcal{H}_\kappa}^2 - \|\boldsymbol{\mu}_{\mathbb{Q}}\|_{\mathcal{H}_\kappa}^2 \right)^2} \right) ,$$

which can be expressed as

$$\sqrt{m} \left( \widehat{\text{NAMMD}}(X, Y; \kappa) - \epsilon \right) \xrightarrow{d} \mathcal{N} \left( 0, \frac{4E[H_{1,2} H_{1,3}] - 4(E[H_{1,2}])^2}{\left( 4K - \|\boldsymbol{\mu}_{\mathbb{P}}\|_{\mathcal{H}_\kappa}^2 - \|\boldsymbol{\mu}_{\mathbb{Q}}\|_{\mathcal{H}_\kappa}^2 \right)^2} \right) .$$

This completes the proof. $\qquad \square$

## C.2. Detailed Proofs of Lemma 4

We present the proofs of Lemma 4 as follows.

*Proof.* For simplicity, we let

$$\hat{A} = \sqrt{((4m - 8)\hat{\zeta}_1 + 2\hat{\zeta}_2)/(m - 1)} \quad \text{and} \quad A = \sqrt{4E[H_{1,2} H_{1,3}] - 4(E[H_{1,2}])^2} ,$$

and

$$\hat{B} = (m^2 - m)^{-1} \sum_{i \neq j} 4K - \kappa(\mathbf{x}_i, \mathbf{x}_j) - \kappa(\mathbf{y}_i, \mathbf{y}_j) \quad \text{and} \quad B = 4K - \|\boldsymbol{\mu}_{\mathbb{P}}\|_{\mathcal{H}_\kappa}^2 - \|\boldsymbol{\mu}_{\mathbb{Q}}\|_{\mathcal{H}_\kappa}^2 .$$

Build on these results, we can bound the bias as follows:

$$
\begin{aligned}
\left| E[\sigma_{X,Y}^2] - \sigma_{\mathbb{P},\mathbb{Q}}^2 \right| &= \left| E\left[ \frac{\hat{A}^2}{\hat{B}^2} \right] - \frac{A^2}{B^2} \right| = \left| E\left[ \frac{\hat{A}^2}{\hat{B}^2} \right] - E\left[ \frac{\hat{A}^2}{B^2} \right] + E\left[ \frac{\hat{A}^2}{B^2} \right] - \frac{A^2}{B^2} \right| \\
&= \left| E\left[ \frac{\hat{A}^2}{\hat{B}^2} \right] - E\left[ \frac{\hat{A}^2}{B^2} \right] \right| \\
&\leq E\left[ \left| \frac{\hat{A}^2}{\hat{B}^2} - \frac{\hat{A}^2}{B^2} \right| \right] \\
&= E\left[ \left| \frac{\hat{A}^2(B - \hat{B})(B + \hat{B})}{\hat{B}^2 B^2} \right| \right] \\
&\leq C * E\left[ \left| B - \hat{B} \right| \right]
\end{aligned}
$$

---

[6]The exact delta-method variance of NAMMD generally involves the joint first-order fluctuation of both the MMD numerator and the normalizing denominator. In this proof and in our calibration, we use the standard MMD numerator variance scaled by the squared population denominator. This variance form is valid under the first-order variance simplification condition $\text{Var}(h_1 - \epsilon g_1) = \text{Var}(h_1)$, where $h_1$ and $g_1$ are the first-order projections of the numerator $U$-statistic with kernel $H_{i,j}$ and the denominator $U$-statistic with kernel $G_{i,j} = 4K - \kappa(\boldsymbol{x}_i, \boldsymbol{x}_j) - \kappa(\boldsymbol{y}_i, \boldsymbol{y}_j)$, respectively. Without this simplification, the full delta-method variance can be obtained by replacing $H_{i,j}$ with $H_{i,j} - \epsilon G_{i,j}$ in the $U$-statistic variance components.

where $C > 0$ is a constant that ensures $\frac{\hat{A}^2(B+\hat{B})}{\hat{B}^2 B^2} \leq C$, and it exists since the kernel is bounded. The second equation is based on the unbiased variance estimator of the $U$-statistic, i.e. $\hat{A}$. Based on the large deviation bound for $B$, we have

$$\Pr\left(\left|B - \hat{B}\right| \geq t\right) \leq 2\exp\left(-mt^2/4K^2\right)$$

and

$$
\begin{aligned}
C * E\left[\left|B - \hat{B}\right|\right] &= C * \int_0^\infty \Pr\left(\left|B - \hat{B}\right| \geq t\right) dt \\
&\leq C * \int_0^\infty 2\exp\left(-mt^2/4K^2\right) dt \\
&= C * \int_0^\infty 2\exp\left(-u\right)\frac{K}{\sqrt{m}\sqrt{u}} du \\
&= C * \frac{2K\sqrt{\pi}}{\sqrt{m}} = O\left(\frac{1}{\sqrt{m}}\right) .
\end{aligned}
$$

This completes the proof. $\qquad\square$

### C.3. Detailed Proofs of Lemma 5

*Proof.* Recall our NAMMD distance as follows:

$$\text{NAMMD}(\mathbb{P}, \mathbb{Q}; \kappa) = \frac{\|\boldsymbol{\mu}_{\mathbb{P}} - \boldsymbol{\mu}_{\mathbb{Q}}\|^2_{\mathcal{H}_\kappa}}{4K - \|\boldsymbol{\mu}_{\mathbb{P}}\|^2_{\mathcal{H}_\kappa} - \|\boldsymbol{\mu}_{\mathbb{Q}}\|^2_{\mathcal{H}_\kappa}} = \frac{\text{MMD}^2(\mathbb{P}, \mathbb{Q}; \kappa)}{4K - \|\boldsymbol{\mu}_{\mathbb{P}}\|^2_{\mathcal{H}_\kappa} - \|\boldsymbol{\mu}_{\mathbb{Q}}\|^2_{\mathcal{H}_\kappa}} .$$

Given two i.i.d. samples $X = \{\boldsymbol{x}_1, \boldsymbol{x}_2, ..., \boldsymbol{x}_m\} \sim \mathbb{P}^m$ and $Y = \{\boldsymbol{y}_1, \boldsymbol{y}_2, ..., \boldsymbol{y}_m\} \sim \mathbb{Q}^m$, we have the empirical estimator as follows

$$
\begin{aligned}
\widehat{\text{NAMMD}}(X, Y; \kappa) &= \frac{\sum_{i \neq j} \kappa(\boldsymbol{x}_i, \boldsymbol{x}_j) + \kappa(\boldsymbol{y}_i, \boldsymbol{y}_j) - \kappa(\boldsymbol{x}_i, \boldsymbol{y}_j) - \kappa(\boldsymbol{y}_i, \boldsymbol{x}_j)}{\sum_{i \neq j} 4K - \kappa(\boldsymbol{x}_i, \boldsymbol{x}_j) - \kappa(\boldsymbol{y}_i, \boldsymbol{y}_j)} \\
&= \frac{\widehat{\text{MMD}}^2(X, Y; \kappa)}{1/(m^2 - m)\sum_{i \neq j} 4K - \kappa(\boldsymbol{x}_i, \boldsymbol{x}_j) - \kappa(\boldsymbol{y}_i, \boldsymbol{y}_j)} .
\end{aligned}
$$

We denote by

$$
\begin{aligned}
A &= |\widehat{\text{NAMMD}}(X, Y; \kappa) - \text{NAMMD}(\mathbb{P}, \mathbb{Q}; \kappa)| \\
&= \left|\frac{\widehat{\text{MMD}}^2(X, Y; \kappa) - \text{MMD}^2(\mathbb{P}, \mathbb{Q}; \kappa) + \text{MMD}^2(\mathbb{P}, \mathbb{Q}; \kappa)}{1/(m^2 - m)\sum_{i \neq j} 4K - \kappa(\boldsymbol{x}_i, \boldsymbol{x}_j) - \kappa(\boldsymbol{y}_i, \boldsymbol{y}_j)} - \frac{\text{MMD}^2(\mathbb{P}, \mathbb{Q}; \kappa)}{4K - \|\boldsymbol{\mu}_{\mathbb{P}}\|^2_{\mathcal{H}_\kappa} - \|\boldsymbol{\mu}_{\mathbb{Q}}\|^2_{\mathcal{H}_\kappa}}\right| .
\end{aligned}
$$

Given this, we let

$$B = \left|\frac{\widehat{\text{MMD}}^2(X, Y; \kappa) - \text{MMD}^2(\mathbb{P}, \mathbb{Q}; \kappa)}{1/(m^2 - m)\sum_{i \neq j} 4K - \kappa(\boldsymbol{x}_i, \boldsymbol{x}_j) - \kappa(\boldsymbol{y}_i, \boldsymbol{y}_j)}\right| ,$$

and

$$C = \left|\frac{\text{MMD}^2(\mathbb{P}, \mathbb{Q}; \kappa)}{1/(m^2 - m)\sum_{i \neq j} 4K - \kappa(\boldsymbol{x}_i, \boldsymbol{x}_j) - \kappa(\boldsymbol{y}_i, \boldsymbol{y}_j)} - \frac{\text{MMD}^2(\mathbb{P}, \mathbb{Q}; \kappa)}{4K - \|\boldsymbol{\mu}_{\mathbb{P}}\|^2_{\mathcal{H}_\kappa} - \|\boldsymbol{\mu}_{\mathbb{Q}}\|^2_{\mathcal{H}_\kappa}}\right| .$$

It is easy to see that $A \leq B + C$ and we have

$$\Pr\left(A \geq t\right) \leq \Pr\left(B + C \geq t\right) \leq \Pr\left(B \geq b\right) + \Pr\left(C \geq c\right) ,$$

for $b + c = t$ with $t > 0$ and $b, c \geq 0$.

Based on the large deviation bound for U-statistic (Theorem 11), we have

$$\Pr(B \ge b) \le \Pr\left(\left|\widehat{\mathrm{MMD}}^2(X,Y;\kappa) - \mathrm{MMD}^2(\mathbb{P},\mathbb{Q};\kappa)\right|/2K \ge b\right) \le 2\exp\left(-mb^2/4\right),$$

In a similar manner, we have

$$
\begin{aligned}
&\Pr(C \ge c)\\
&\le \Pr\left(\frac{\mathrm{MMD}^2(\mathbb{P},\mathbb{Q};\kappa)|\sum_{i\ne j}(\kappa(\boldsymbol{x}_i,\boldsymbol{x}_j)+\kappa(\boldsymbol{y}_i,\boldsymbol{y}_j))/(m^2-m)) - \|\boldsymbol{\mu}_\mathbb{P}\|_{\mathcal{H}_\kappa}^2 - \|\boldsymbol{\mu}_\mathbb{Q}\|_{\mathcal{H}_\kappa}^2|}{(1/(m^2-m)\sum_{i\ne j}4K - \kappa(\boldsymbol{x}_i,\boldsymbol{x}_j)-\kappa(\boldsymbol{y}_i,\boldsymbol{y}_j))\cdot(4K - \|\boldsymbol{\mu}_\mathbb{P}\|_{\mathcal{H}_\kappa}^2 - \|\boldsymbol{\mu}_\mathbb{Q}\|_{\mathcal{H}_\kappa}^2)} \ge c\right)\\
&\le \Pr\left(\left|\sum_{i\ne j}\frac{\kappa(\boldsymbol{x}_i,\boldsymbol{x}_j)}{m(m-1)} + \frac{\kappa(\boldsymbol{y}_i,\boldsymbol{y}_j)}{m(m-1)} - \|\boldsymbol{\mu}_\mathbb{P}\|_{\mathcal{H}_\kappa}^2 - \|\boldsymbol{\mu}_\mathbb{Q}\|_{\mathcal{H}_\kappa}^2\right| \frac{\mathrm{MMD}^2(\mathbb{P},\mathbb{Q};\kappa)}{4K^2} \ge c\right)\\
&\le \Pr\left(\left|\sum_{i\ne j}\frac{\kappa(\boldsymbol{x}_i,\boldsymbol{x}_j)}{m(m-1)} + \frac{\kappa(\boldsymbol{y}_i,\boldsymbol{y}_j)}{m(m-1)} - \|\boldsymbol{\mu}_\mathbb{P}\|_{\mathcal{H}_\kappa}^2 - \|\boldsymbol{\mu}_\mathbb{Q}\|_{\mathcal{H}_\kappa}^2\right|/2K \ge c\right)\\
&\le 2\exp\left(-mc^2\right)
\end{aligned}
$$

For simplicity, let $b = 2t/3$ and $c = t/3$, we have

$$
\begin{aligned}
\Pr\left(A \ge t\right) &\le \Pr\left(B \ge 2t/3\right) + \Pr\left(C \ge t/3\right)\\
&= 4\exp\left(-mt^2/9\right).
\end{aligned}
$$

This completes the proof. $\qquad\square$

## C.4. Detailed Proofs of Theorem 6

We present the proofs of Theorem 6 as follows.

*Proof.* Under null hypothesis $\boldsymbol{H}_0 : \mathrm{NAMMD}(\mathbb{P},\mathbb{Q};\kappa) \le \epsilon$ with $\epsilon \in (0,1)$, the type-I error is

$$\Pr(\mathrm{NAMMD}(X,Y;\kappa) > \tau_\alpha),$$

where $\hat{\tau}_\alpha = \epsilon + \sigma_{X,Y}\mathcal{N}_{1-\alpha}/\sqrt{m}$ (as defined in Eqn. (4)) is the $(1-\alpha)$-quantile of the asymptotic Gaussian distribution in Theorem 2 with $\mathrm{NAMMD}(\mathbb{P},\mathbb{Q};\kappa) = \epsilon$.

Recall that $\sigma_{X,Y}$ is the estimator of $\sigma_{\mathbb{P},\mathbb{Q}}^2$, where

$$\sigma_{\mathbb{P},\mathbb{Q}} = \frac{\sqrt{4E[H_{1,2}H_{1,3}] - 4(E[H_{1,2}])^2}}{4K - \|\boldsymbol{\mu}_\mathbb{P}\|_{\mathcal{H}_\kappa}^2 - \|\boldsymbol{\mu}_\mathbb{Q}\|_{\mathcal{H}_\kappa}^2},$$

and

$$\sigma_{X,Y} = \frac{\sqrt{((4m-8)\hat{\zeta}_1 + 2\hat{\zeta}_2)/(m-1)}}{(m^2-m)^{-1}\sum_{i\ne j}4K - \kappa(\boldsymbol{x}_i,\boldsymbol{x}_j) - \kappa(\boldsymbol{y}_i,\boldsymbol{y}_j)},$$

where $\hat{\zeta}_1$ and $\hat{\zeta}_2$ are standard variance components of the MMD (Serfling, 2009; Sutherland, 2019) (Appendix B.3).

We begin by showing that $\sigma_{X,Y}$ converges to $\sigma_{\mathbb{P},\mathbb{Q}}$. As detailed in Appendix B.3, the terms in the numerator involving $\hat{\zeta}_1$ and $\hat{\zeta}_2$ converge in probability. We now present the convergence of the denominator

$$(m^2-m)^{-1}\sum_{i\ne j}4K - \kappa(\boldsymbol{x}_i,\boldsymbol{x}_j) - \kappa(\boldsymbol{y}_i,\boldsymbol{y}_j),$$

which can be regarded as a $U$-statistic, and it follows that

$$(m^2 - m)^{-1} \sum_{i \neq j} 4K - \kappa(\boldsymbol{x}_i, \boldsymbol{x}_j) - \kappa(\boldsymbol{y}_i, \boldsymbol{y}_j) \xrightarrow{p} 4K - \|\boldsymbol{\mu}_{\mathbb{P}}\|_{\mathcal{H}_\kappa}^2 - \|\boldsymbol{\mu}_{\mathbb{Q}}\|_{\mathcal{H}_\kappa}^2 ,$$

by the large deviation bound (Theorem 11) for $U$-statistic.

Hence, by the continuous mapping theorem, we have that

$$\sigma_{X,Y} \xrightarrow{p} \sigma_{\mathbb{P},\mathbb{Q}} .$$

Next, we prove asymptotic type-I error control based on the convergence of the variance. The null hypothesis $\boldsymbol{H}_0$ : NAMMD$(\mathbb{P}, \mathbb{Q}; \kappa) \leq \epsilon$ with $\epsilon \in (0, 1)$ is composite, covering three cases: 1) NAMMD$(\mathbb{P}, \mathbb{Q}; \kappa) = \epsilon$; 2) NAMMD$(\mathbb{P}, \mathbb{Q}; \kappa) = \epsilon' \in (0, \epsilon)$; 3) NAMMD$(\mathbb{P}, \mathbb{Q}; \kappa) = 0$. We now prove that under three cases the type-I error Pr(NAMMD$(X, Y; \kappa) > \hat{\tau}_\alpha$) are all bounded by $\alpha$.

- Case 1: NAMMD$(\mathbb{P}, \mathbb{Q}; \kappa) = \epsilon$. Since $\hat{\tau}_\alpha = \epsilon + \sigma_{X,Y} \mathcal{N}_{1-\alpha} / \sqrt{m}$ corresponds to the $(1 - \alpha)$-quantile of the asymptotic Gaussian distribution with NAMMD$(\mathbb{P}, \mathbb{Q}; \kappa) = \epsilon$ from Lemma 2, the following equality holds asymptotically

$$\Pr(\text{NAMMD}(X, Y; \kappa) > \hat{\tau}_\alpha) = \alpha.$$

- Case 2: NAMMD$(\mathbb{P}, \mathbb{Q}; \kappa) = \epsilon' \in (0, \epsilon)$. The $(1 - \alpha)$-quantile of the asymptotic Gaussian distribution with NAMMD$(\mathbb{P}, \mathbb{Q}; \kappa) = \epsilon'$ is $\hat{\tau}'_\alpha = \epsilon' + \sigma_{X,Y} \mathcal{N}_{1-\alpha} / \sqrt{m}$ from Lemma 2. Then, the following equality holds asymptotically

$$\Pr(\text{NAMMD}(X, Y; \kappa) > \hat{\tau}'_\alpha) = \alpha,$$

Since $\epsilon' < \epsilon$, we have $\hat{\tau}'_\alpha < \hat{\tau}_\alpha$ and

$$\Pr(\text{NAMMD}(X, Y; \kappa) > \hat{\tau}_\alpha) < \Pr(\text{NAMMD}(X, Y; \kappa) > \hat{\tau}'_\alpha) = \alpha$$

Hence, type-I error is bounded by $\alpha$.

- Case 3: NAMMD$(\mathbb{P}, \mathbb{Q}; \kappa) = 0$. According to the Lemma 5, we have that

$$\Pr(\text{NAMMD}(X, Y; \kappa) > \hat{\tau}_\alpha) < \Pr(\text{NAMMD}(X, Y; \kappa) > \epsilon) \leq 2 \exp(-m\epsilon^2/9) .$$

This probability decays exponentially with the sample size $m$, implying that

$$\Pr(\text{NAMMD}(X, Y; \kappa) > \hat{\tau}_\alpha) \leq \alpha ,$$

holds in the asymptotic manner.

This completes the proof. $\qquad\square$

### C.5. Detailed Proofs of Theorem 7

*Proof.* Under the alternative hypothesis $\boldsymbol{H}_1$ : NAMMD$(\mathbb{P}, \mathbb{Q}; \kappa) > \epsilon$ with $\epsilon \in (0, 1)$, , we need to correctly reject the null hypothesis $\boldsymbol{H}_0$ : NAMMD$(\mathbb{P}, \mathbb{Q}; \kappa) \leq \epsilon$. According to Eqn. 4, we set $\hat{\tau}_\alpha$ as the $(1 - \alpha)$-quantile of the asymptotic null distribution of NAMMD$(\mathbb{P}, \mathbb{Q}; \kappa) = \epsilon$ from Lemma 2 as,

$$\hat{\tau}_\alpha = \epsilon + \frac{\sigma_{X,Y} \mathcal{N}_{1-\alpha}}{\sqrt{m}} ,$$

where the empirical estimator of variance is given by

$$\sigma_{X,Y} = \frac{\sqrt{((4m - 8)\hat{\zeta}_1 + 2\hat{\zeta}_2)/(m - 1)}}{(m^2 - m)^{-1} \sum_{i \neq j} 4K - \kappa(\boldsymbol{x}_i, \boldsymbol{x}_j) - \kappa(\boldsymbol{y}_i, \boldsymbol{y}_j)} ,$$

where $\hat{\zeta}_1$ and $\hat{\zeta}_2$ are standard variance components of the MMD (Serfling, 2009; Sutherland, 2019). We present the details of the estimator in Appendix B.3.

It is easy to see that, for $\kappa(\cdot, \cdot) \leq K$,

$$(m^2 - m)^{-1} \sum_{i \neq j} 4K - \kappa(\boldsymbol{x}_i, \boldsymbol{x}_j) - \kappa(\boldsymbol{y}_i, \boldsymbol{y}_j) \geq 2K \quad \text{and} \quad \hat{\zeta}_1 \leq 4K^2 \quad \text{and} \quad \hat{\zeta}_2 \leq 4K^2 \,,$$

Hence, as we can see,

$$
\begin{aligned}
\sigma_{X,Y} &\leq \frac{\sqrt{(4m - 6)/(m - 1)4K^2}}{2K} \\
&\leq 4K/2K \\
&\leq 2 \,,
\end{aligned}
$$

and we have

$$\hat{\tau}_\alpha \leq \epsilon + \frac{2\mathcal{N}_{1-\alpha}}{\sqrt{m}} \,.$$

In a similar manner, to ensure the rejection, we have

$$\widehat{\mathrm{NAMMD}}(X, Y; \kappa) > \epsilon + \frac{2\mathcal{N}_{1-\alpha}}{\sqrt{m}}.$$

To derive the bound, the following holds with at least probability $1 - \upsilon$,

$$\widehat{\mathrm{NAMMD}}(X, Y; \kappa) \geq \mathrm{NAMMD}(\mathbb{P}, \mathbb{Q}; \kappa) - \sqrt{\frac{9\log 2/\upsilon}{m}} \,,$$

then, we have

$$\mathrm{NAMMD}(\mathbb{P}, \mathbb{Q}; \kappa) - \sqrt{\frac{9\log 2/\upsilon}{m}} > \epsilon + \frac{2\mathcal{N}_{1-\alpha}}{\sqrt{m}} \,,$$

which leads to

$$m \geq \frac{\left(2 * \mathcal{N}_{1-\alpha} + \sqrt{9\log 2/\upsilon}\right)^2}{(\mathrm{NAMMD}(\mathbb{P}, \mathbb{Q}; \kappa) - \epsilon)^2} \,.$$

This completes the proof. $\qquad\square$

## C.6. Detailed Proofs and Explanations of Theorem 9

### C.6.1. DETAILED PROOFS OF THEOREM 9

Given Definition 8, we assume $\mathbb{P}_1$ and $\mathbb{Q}_1$ are known, and $X$ and $Y$ are two i.i.d. samples drawn from $\mathbb{P}_2$ and $\mathbb{Q}_2$. The goals of distribution closeness testing are to correctly reject null hypotheses with calculated statistics $\widehat{\mathrm{NAMMD}}(X, Y; \kappa)$ and $\widehat{\mathrm{MMD}}(X, Y; \kappa)$.

For simplicity, we let

$$
\begin{aligned}
\mathrm{NORM}(\mathbb{P}_1, \mathbb{Q}_1; \kappa) &= 4K - \|\boldsymbol{\mu}_{\mathbb{P}_1}\|_{\mathcal{H}_\kappa}^2 - \|\boldsymbol{\mu}_{\mathbb{Q}_1}\|_{\mathcal{H}_\kappa}^2 \,, \\
\mathrm{NORM}(\mathbb{P}_2, \mathbb{Q}_2; \kappa) &= 4K - \|\boldsymbol{\mu}_{\mathbb{P}_2}\|_{\mathcal{H}_\kappa}^2 - \|\boldsymbol{\mu}_{\mathbb{Q}_2}\|_{\mathcal{H}_\kappa}^2 \,,
\end{aligned}
$$

and rewrite the empirical estimator with $X$ and $Y$ as follows

$$\widehat{\mathrm{NORM}}(X, Y; \kappa) = 1/(m^2 - m) \sum_{i \neq j} 4K - \kappa(\boldsymbol{x}_i, \boldsymbol{x}_j) - \kappa(\boldsymbol{y}_i, \boldsymbol{y}_j) \,.$$

*Proof.* [7]

Let $\tau_\alpha^M$ and $\tau_\alpha^N$ be the true $(1-\alpha)$-quantiles of asymptotic null distributions of $\sqrt{m}\widehat{\text{MMD}}$ and $\sqrt{m}\widehat{\text{NAMMD}}$, respectively. Specifically, from Lemma 12, we have

$$\tau_\alpha^M = \sqrt{m}\text{MMD}^2(\mathbb{P}_1, \mathbb{Q}_1; \kappa) + \sigma_M \mathcal{N}_{1-\alpha} \ ,$$

where

$$\sigma_M^2 := 4E[H_{1,2}H_{1,3}] - 4(E[H_{1,2}])^2 \ , \tag{7}$$

and $H_{i,j} = \kappa(\boldsymbol{x}_i, \boldsymbol{x}_j) + \kappa(\boldsymbol{y}_i, \boldsymbol{y}_j) - \kappa(\boldsymbol{x}_i, \boldsymbol{y}_j) - \kappa(\boldsymbol{y}_i, \boldsymbol{x}_j)$, and the expectation are taken with respect to $\boldsymbol{x}_1, \boldsymbol{x}_2, \boldsymbol{x}_3 \overset{\text{i.i.d.}}{\sim} \mathbb{P}_2$ and $\boldsymbol{y}_1, \boldsymbol{y}_2, \boldsymbol{y}_3 \overset{\text{i.i.d.}}{\sim} \mathbb{Q}_2$.

In a similar manner, from Lemma 2, we have

$$
\begin{aligned}
\tau_\alpha^N &= \sqrt{m}\text{NAMMD}(\mathbb{P}_1, \mathbb{Q}_1; \kappa) + \sigma_{\mathbb{P}_2, \mathbb{Q}_2} \mathcal{N}_{1-\alpha} \\
&= \frac{\sqrt{m}\text{MMD}^2(\mathbb{P}_1, \mathbb{Q}_1; \kappa)}{4K - \|\boldsymbol{\mu}_{\mathbb{P}_1}\|_{\mathcal{H}_\kappa}^2 - \|\boldsymbol{\mu}_{\mathbb{Q}_1}\|_{\mathcal{H}_\kappa}^2} + \frac{\sigma_M \mathcal{N}_{1-\alpha}}{4K - \|\boldsymbol{\mu}_{\mathbb{P}_2}\|_{\mathcal{H}_\kappa}^2 - \|\boldsymbol{\mu}_{\mathbb{Q}_2}\|_{\mathcal{H}_\kappa}^2} \\
&= \frac{\sqrt{m}\text{MMD}^2(\mathbb{P}_1, \mathbb{Q}_1; \kappa)}{\text{NORM}(\mathbb{P}_1, \mathbb{Q}_1; \kappa)} + \frac{\sigma_M \mathcal{N}_{1-\alpha}}{\text{NORM}(\mathbb{P}_2, \mathbb{Q}_2; \kappa)} \ ,
\end{aligned}
$$

It is easy to see that $\sqrt{m}\widehat{\text{MMD}}(X, Y; \kappa) > \tau_\alpha^M$ is equivalent to

$$\sqrt{m}\widehat{\text{MMD}}(X, Y; \kappa) - \sqrt{m}\text{MMD}^2(\mathbb{P}_1, \mathbb{Q}_1; \kappa) > \sigma_M \mathcal{N}_{1-\alpha} \ , \tag{8}$$

and in a similar manner, $\sqrt{m}\widehat{\text{NAMMD}}(X, Y; \kappa) > \tau_\alpha^N$ is equivalent to

$$\frac{\text{NORM}(\mathbb{P}_2, \mathbb{Q}_2; \kappa)}{\widehat{\text{NORM}}(X, Y; \kappa)}\sqrt{m}\widehat{\text{MMD}}(X, Y; \kappa) - \frac{\text{NORM}(\mathbb{P}_2, \mathbb{Q}_2; \kappa)}{\text{NORM}(\mathbb{P}_1, \mathbb{Q}_1; \kappa)}\sqrt{m}\text{MMD}^2(\mathbb{P}_1, \mathbb{Q}_1; \kappa) > \sigma_M \mathcal{N}_{1-\alpha} \ , \tag{9}$$

Hence, to ensure

$$\sqrt{m}\widehat{\text{MMD}}(X, Y; \kappa) > \tau_\alpha^M \quad \Rightarrow \quad \sqrt{m}\widehat{\text{NAMMD}}(X, Y; \kappa) > \tau_\alpha^N \ , \tag{10}$$

we must verify that, according to Eqn. 8 and 9,

$$\left(\frac{\text{NORM}(\mathbb{P}_2, \mathbb{Q}_2; \kappa)}{\widehat{\text{NORM}}(X, Y; \kappa)} - 1\right)\sqrt{m}\widehat{\text{MMD}}(X, Y; \kappa) \geq \left(\frac{\text{NORM}(\mathbb{P}_2, \mathbb{Q}_2; \kappa)}{\text{NORM}(\mathbb{P}_1, \mathbb{Q}_1; \kappa)} - 1\right)\sqrt{m}\text{MMD}^2(\mathbb{P}_1, \mathbb{Q}_1; \kappa) \ . \tag{11}$$

Based on Eqn. 8, the inequality in Eqn. 11 can be adjusted to

$$
\begin{aligned}
&\frac{\text{NORM}(\mathbb{P}_2, \mathbb{Q}_2; \kappa) - \widehat{\text{NORM}}(X, Y; \kappa)}{\widehat{\text{NORM}}(X, Y; \kappa)} \\
&\geq \frac{\text{NORM}(\mathbb{P}_2, \mathbb{Q}_2; \kappa) - \text{NORM}(\mathbb{P}_1, \mathbb{Q}_1; \kappa)}{\text{NORM}(\mathbb{P}_1, \mathbb{Q}_1; \kappa)} \frac{\sqrt{m}\text{MMD}^2(\mathbb{P}_1, \mathbb{Q}_1; \kappa)}{\sqrt{m}\text{MMD}^2(\mathbb{P}_1, \mathbb{Q}_1; \kappa) + \sigma_M \mathcal{N}_{1-\alpha}} \\
&\geq \sqrt{m}\text{NAMMD}(\mathbb{P}_1, \mathbb{Q}_1; \kappa) \frac{\text{NORM}(\mathbb{P}_2, \mathbb{Q}_2; \kappa) - \text{NORM}(\mathbb{P}_1, \mathbb{Q}_1; \kappa)}{\sqrt{m}\text{MMD}^2(\mathbb{P}_1, \mathbb{Q}_1; \kappa) + \sigma_M \mathcal{N}_{1-\alpha}} \ .
\end{aligned}
$$

---

[7]In this proof, $\tau_\alpha^M$ and $\tau_\alpha^N$ are the asymptotic $(1-\alpha)$-quantile of distributions of the MMD and NAMMD estimator, under the null hypothesis $\boldsymbol{H}_0^M : \text{MMD}^2(\mathbb{P}_2, \mathbb{Q}_2; \kappa) \leq \epsilon^M$ and $\boldsymbol{H}_0^N : \text{NAMMD}(\mathbb{P}_2, \mathbb{Q}_2; \kappa) \leq \epsilon^N$ for distribution closeness testing. Here, $\epsilon^M = \text{MMD}^2(\mathbb{P}_1, \mathbb{Q}_1; \kappa)$ and $\epsilon^N = \text{NAMMD}(\mathbb{P}_1, \mathbb{Q}_1; \kappa)$. The respective null distributions for MMD and NAMMD are presented in Lemmas 12 and 2. In practical, since these null distributions are normal, we directly estimate the testing thresholds $\tau_\alpha^M$ and $\tau_\alpha^N$ by computing the variances of the corresponding normal distributions (Jitkrittum et al., 2016; Chwialkowski et al., 2015; Gretton et al., 2012a; Biggs et al., 2023).

Given this, we have

$$\text{NORM}(\mathbb{P}_2, \mathbb{Q}_2; \kappa)$$
$$\geq \left(1 + \sqrt{m}\text{NAMMD}(\mathbb{P}_1, \mathbb{Q}_1; \kappa)\frac{\text{NORM}(\mathbb{P}_2, \mathbb{Q}_2; \kappa) - \text{NORM}(\mathbb{P}_1, \mathbb{Q}_1; \kappa)}{\sqrt{m}\text{MMD}^2(\mathbb{P}_1, \mathbb{Q}_1; \kappa) + \sigma_M \mathcal{N}_{1-\alpha}}\right)\widehat{\text{NORM}}(X, Y; \kappa)$$
$$\geq (1 - \Delta)\widehat{\text{NORM}}(X, Y; \kappa),$$

where we let, for simplicity

$$\Delta = \sqrt{m}\text{NAMMD}(\mathbb{P}_1, \mathbb{Q}_1; \kappa)\frac{\|\boldsymbol{\mu}_{\mathbb{P}_2}\|^2_{\mathcal{H}_\kappa} + \|\boldsymbol{\mu}_{\mathbb{Q}_2}\|^2_{\mathcal{H}_\kappa} - \|\boldsymbol{\mu}_{\mathbb{P}_1}\|^2_{\mathcal{H}_\kappa} - \|\boldsymbol{\mu}_{\mathbb{Q}_1}\|^2_{\mathcal{H}_\kappa}}{\sqrt{m}\text{MMD}^2(\mathbb{P}_1, \mathbb{Q}_1; \kappa) + \sigma_M \mathcal{N}_{1-\alpha}}.$$

Here, by assuming $\|\boldsymbol{\mu}_{\mathbb{P}_1}\|^2_{\mathcal{H}_\kappa} + \|\boldsymbol{\mu}_{\mathbb{Q}_1}\|^2_{\mathcal{H}_\kappa} < \|\boldsymbol{\mu}_{\mathbb{P}_2}\|^2_{\mathcal{H}_\kappa} + \|\boldsymbol{\mu}_{\mathbb{Q}_2}\|^2_{\mathcal{H}_\kappa}$, we have $\Delta \in (0, 1/2)$.

As we can see, $\text{NORM}(\mathbb{P}_2, \mathbb{Q}_2; \kappa) \geq (1 - \Delta)\widehat{\text{NORM}}(X, Y; \kappa)$ is equivalent to

$$(1 - \Delta)\widehat{\text{NORM}}(X, Y; \kappa) - (1 - \Delta)\text{NORM}(\mathbb{P}_2, \mathbb{Q}_2; \kappa) \leq \Delta \cdot \text{NORM}(\mathbb{P}_2, \mathbb{Q}_2; \kappa),$$

which is

$$\widehat{\text{NORM}}(X, Y; \kappa) - \text{NORM}(\mathbb{P}_2, \mathbb{Q}_2; \kappa) \leq \frac{\Delta}{1 - \Delta}\text{NORM}(\mathbb{P}_2, \mathbb{Q}_2; \kappa).$$

Using the large deviation bound as follows

$$P\left(\widehat{\text{NORM}}(X, Y; \kappa) - \text{NORM}(\mathbb{P}_2, \mathbb{Q}_2; \kappa) \geq t\right) \leq \exp(-mt^2/4K^2),$$

with $t > 0$, the Eqn. 10 holds with probability at least

$$1 - \exp\left(-m\left(\frac{\Delta}{1 - \Delta}\text{NORM}(\mathbb{P}_2, \mathbb{Q}_2; \kappa)\right)^2/4K^2\right).$$

This completes the proof of first part.

From Lemma 12, we have the test power of MMD test as follows

$$p_M = \Pr\left(\sqrt{m}\widehat{\text{MMD}}^2(X, Y; \kappa) \geq \tau_\alpha^M\right),$$

with

$$\Pr\left(\sqrt{m}\widehat{\text{MMD}}^2(X, Y; \kappa) \geq \tau_\alpha^M\right) - \Phi\left(\frac{\sqrt{m}\text{MMD}^2(\mathbb{P}_2, \mathbb{Q}_2; \kappa) - \tau_\alpha^M}{\sigma_M}\right) \to 0,$$

which is equivalent to

$$\Phi\left(\frac{\sqrt{m}(\text{MMD}^2(\mathbb{P}_2, \mathbb{Q}_2; \kappa) - \text{MMD}^2(\mathbb{P}_1, \mathbb{Q}_1; \kappa)) - \sigma_M \mathcal{N}_{1-\alpha}}{\sigma_M}\right).$$

The test power of NAMMD test is given by, according to Lemma 2,

$$p_N = \Pr\left(\sqrt{m}\widehat{\text{NAMMD}}(X, Y; \kappa) \geq \tau_\alpha^N\right),$$

with

$$\Pr\left(\sqrt{m}\widehat{\text{NAMMD}}(X, Y; \kappa) \geq \tau_\alpha^N\right) - \Phi\left(\frac{\sqrt{m}\text{NAMMD}(\mathbb{P}_2, \mathbb{Q}_2; \kappa) - \tau_\alpha^N}{\sigma_{\mathbb{P}_2, \mathbb{Q}_2}}\right) \to 0,$$

which is equivalent to

$$\Phi\left(\frac{\sqrt{m}\left(\text{MMD}^2(\mathbb{P}_2, \mathbb{Q}_2; \kappa) - \dfrac{\text{NORM}(\mathbb{P}_2, \mathbb{Q}_2; \kappa)}{\text{NORM}(\mathbb{P}_1, \mathbb{Q}_1; \kappa)}\text{MMD}^2(\mathbb{P}_1, \mathbb{Q}_1; \kappa)\right) - \sigma_M \mathcal{N}_{1-\alpha}}{\sigma_M}\right).$$

For simplicity, we let

$$A = \frac{\sqrt{m}(\mathrm{MMD}^2(\mathbb{P}_2, \mathbb{Q}_2; \kappa) - \mathrm{MMD}^2(\mathbb{P}_1, \mathbb{Q}_1; \kappa)) - \sigma_M \mathcal{N}_{1-\alpha}}{\sigma_M} \, ,$$

and

$$B = \sqrt{m} \left( 1 - \frac{\mathrm{NORM}(\mathbb{P}_2, \mathbb{Q}_2; \kappa)}{\mathrm{NORM}(\mathbb{P}_1, \mathbb{Q}_1; \kappa)} \right) \frac{\mathrm{MMD}^2(\mathbb{P}_1, \mathbb{Q}_1; \kappa)}{\sigma_M} \, .$$

Similarly, by assuming $\|\boldsymbol{\mu}_{\mathbb{P}_1}\|_{\mathcal{H}_\kappa}^2 + \|\boldsymbol{\mu}_{\mathbb{Q}_1}\|_{\mathcal{H}_\kappa}^2 < \|\boldsymbol{\mu}_{\mathbb{P}_2}\|_{\mathcal{H}_\kappa}^2 + \|\boldsymbol{\mu}_{\mathbb{Q}_2}\|_{\mathcal{H}_\kappa}^2$, we have $B > 0$ with $\mathrm{NORM}(\mathbb{P}_1, \mathbb{Q}_1; \kappa) > \mathrm{NORM}(\mathbb{P}_2, \mathbb{Q}_2; \kappa)$.

As we can see,

$$\varsigma = p_N - p_M = \frac{1}{\sqrt{2\pi}} \int_A^{A+B} e^{-t^2/2} dt \, .$$

*which indicates that the NAMMD-based DCT achieves higher test power than the MMD-based DCT by a gap of $\varsigma$.*

Next, we examine the case where $\varsigma \geq 1/65$. Considering the following inequality holds

$$0 \leq \frac{\sqrt{m}\mathrm{MMD}^2(\mathbb{P}_2, \mathbb{Q}_2; \kappa) - \tau_\alpha^M}{\sigma_M} \leq 0.7 \, ,$$

which is equivalent to

$$0 \leq A \leq 0.7 \, .$$

It follows that

$$m_A^- \leq m \leq m_A^+ \, ,$$

where

$$m_A^- = \left( \frac{\mathcal{N}_{1-\alpha}\sigma_M}{\mathrm{MMD}^2(\mathbb{P}_2, \mathbb{Q}_2; \kappa) - \mathrm{MMD}^2(\mathbb{P}_1, \mathbb{Q}_1; \kappa)} \right)^2 \, ,$$

$$m_A^+ = \left( \frac{(\mathcal{N}_{1-\alpha} + 0.7)\sigma_M}{\mathrm{MMD}^2(\mathbb{P}_2, \mathbb{Q}_2; \kappa) - \mathrm{MMD}^2(\mathbb{P}_1, \mathbb{Q}_1; \kappa)} \right)^2 \, .$$

In a similar manner, let $B \geq 0.05$, we have

$$m \geq m_B \, ,$$

where

$$m_B = \left( 20 \left( 1 - \frac{\mathrm{NORM}(\mathbb{P}_2, \mathbb{Q}_2; \kappa)}{\mathrm{NORM}(\mathbb{P}_1, \mathbb{Q}_1; \kappa)} \right) \frac{\mathrm{MMD}^2(\mathbb{P}_1, \mathbb{Q}_1; \kappa)}{\sigma_M} \right)^{-2} \, .$$

By introducing

$$C_1 \leq m \leq C_2 \, ,$$

with

$$C_1 = \max\left\{ m_A^-, m_B \right\} \qquad \text{and} \qquad C_2 = m_A^+ \, ,$$

it follows that $B \geq 0.05$ and $-0.75 \leq A \leq 0.70$, and the lower bound of the power improvement is given by

$$\varsigma = p_N - p_M \geq \frac{1}{\sqrt{2\pi}} \int_{0.7}^{0.75} e^{-t^2/2} dt \geq 1/65 \, .$$

This completes the proof. □

C.6.2. DETAILED EXPLANATION ON THE CONDITION AND CONSTANTS IN THEOREM 9

In Theorem, the condition

$$\|\boldsymbol{\mu}_{\mathbb{P}_1}\|_{\mathcal{H}_\kappa}^2 + \|\boldsymbol{\mu}_{\mathbb{Q}_1}\|_{\mathcal{H}_\kappa}^2 < \|\boldsymbol{\mu}_{\mathbb{P}_2}\|_{\mathcal{H}_\kappa}^2 + \|\boldsymbol{\mu}_{\mathbb{Q}_2}\|_{\mathcal{H}_\kappa}^2$$

is closely related to the variance of the distributions, as discussed in Section 1. Specifically, the kernel-based variance is defined as

$$\mathrm{Var}(\mathbb{P};\kappa) = E_{\boldsymbol{x}\sim\mathbb{P}}[\kappa(\boldsymbol{x},\boldsymbol{x})] - \|\boldsymbol{\mu}_\mathbb{P}\|_{\mathcal{H}_\kappa}^2 = K - \|\boldsymbol{\mu}_\mathbb{P}\|_{\mathcal{H}_\kappa}^2 ,$$

where $\kappa(\boldsymbol{x},\boldsymbol{x}') = \Psi(\boldsymbol{x} - \boldsymbol{x}') \le K$ with $K > 0$ for a positive-definite $\Psi(\cdot)$ and $\Psi(\boldsymbol{0}) = K$.

Given the variance term, the condition can be equivalently expressed as

$$\mathrm{Var}(\mathbb{P}_1;\kappa) + \mathrm{Var}(\mathbb{Q}_1;\kappa) > \mathrm{Var}(\mathbb{P}_2;\kappa) + \mathrm{Var}(\mathbb{Q}_2;\kappa) .$$

In the NAMMD distance, we incorporate the norms of distributions (i.e., variance information of distributions), and we analyze its advantages through Theorem 9 using the following example.

**Example 1.** From Appendix C.6.1, we have that

$$C_1 = \max\left\{m_A^-, m_B\right\} ,$$

$$C_2 = \left(\frac{(\mathcal{N}_{1-\alpha} + 0.7)\sigma_M}{\mathrm{MMD}^2(\mathbb{P}_2,\mathbb{Q}_2;\kappa) - \mathrm{MMD}^2(\mathbb{P}_1,\mathbb{Q}_1;\kappa)}\right)^2 ,$$

with

$$m_A^- = \left(\frac{\mathcal{N}_{1-\alpha}\sigma_M}{\mathrm{MMD}^2(\mathbb{P}_2,\mathbb{Q}_2;\kappa) - \mathrm{MMD}^2(\mathbb{P}_1,\mathbb{Q}_1;\kappa)}\right)^2 ,$$

$$m_B = \left(20\left(1 - \frac{\mathrm{NORM}(\mathbb{P}_2,\mathbb{Q}_2;\kappa)}{\mathrm{NORM}(\mathbb{P}_1,\mathbb{Q}_1;\kappa)}\right)\frac{\mathrm{MMD}^2(\mathbb{P}_1,\mathbb{Q}_1;\kappa)}{\sigma_M}\right)^{-2} .$$

Consider the reference distribution pair $\mathbb{P}_1 = \mathcal{N}(0, 1.1)$ and $\mathbb{Q}_1 = \mathcal{N}(0, 1.6)$, and the distribution pair $\mathbb{P}_2 = \mathcal{N}(0, 0.5)$ and $\mathbb{Q}_2 = \mathcal{N}(0, 1.0)$ for testing. A Gaussian kernel $\kappa$ with bandwidth 1.0 is employed. Under this setup, it follows that

$$\|\boldsymbol{\mu}_{\mathbb{P}_1}\|_{\mathcal{H}_\kappa}^2 = E_{\boldsymbol{x},\boldsymbol{x}'\sim\mathbb{P}_1}[\exp(-\|\boldsymbol{x} - \boldsymbol{x}'\|_2^2)] = \int_{-\infty}^{\infty} \frac{\exp\left(-z^2\right)\exp\left(-z^2/(2\times 0.02)\right)}{(2\pi\times 0.02)^{0.5}}dz = 0.4303 ,$$

where we denote $z = \boldsymbol{x} - \boldsymbol{x}'$ in the evaluation of the integral; similarly, we obtain that

$$\|\boldsymbol{\mu}_{\mathbb{Q}_1}\|_{\mathcal{H}_\kappa}^2 = E_{\boldsymbol{y},\boldsymbol{y}'\sim\mathbb{Q}_1}[\exp(-\|\boldsymbol{y} - \boldsymbol{y}'\|_2^2)] = \int_{-\infty}^{\infty} \frac{\exp\left(-z^2\right)\exp\left(-z^2/(2\times 2)\right)}{(2\pi\times 2)^{0.5}}dz = 0.3676 ,$$

by denoting $z = \boldsymbol{y} - \boldsymbol{y}'$ in the evaluation of the integral; similarly, we obtain that

$$\langle\boldsymbol{\mu}_{\mathbb{P}_1}, \boldsymbol{\mu}_{\mathbb{Q}_1}\rangle_{\mathcal{H}_\kappa} = E_{\boldsymbol{x}\sim\mathbb{P}_1,\boldsymbol{y}\sim\mathbb{Q}_1}[\exp(-\|\boldsymbol{x} - \boldsymbol{y}\|_2^2)] = \int_{-\infty}^{\infty} \frac{\exp\left(-z^2\right)\exp\left(-z^2/(2\times 2)\right)}{(2\pi\times 2)^{0.5}}dz = 0.3953,$$

by denoting $z = \boldsymbol{x} - \boldsymbol{y}'$ in the evaluation of the integral. Based on these norms, we can calculate that $\mathrm{MMD}^2(\mathbb{P}_1,\mathbb{Q}_1;\kappa) = 0.0073$.

In a similar manner, we have that

$$\|\boldsymbol{\mu}_{\mathbb{P}_2}\|_{\mathcal{H}_\kappa}^2 = 0.5773, \quad \|\boldsymbol{\mu}_{\mathbb{Q}_2}\|_{\mathcal{H}_\kappa}^2 = 0.4472, \quad \langle\boldsymbol{\mu}_{\mathbb{P}_2}, \boldsymbol{\mu}_{\mathbb{Q}_2}\rangle_{\mathcal{H}_\kappa} = 0.5, \quad \mathrm{MMD}^2(\mathbb{P}_2,\mathbb{Q}_2;\kappa) = 0.0245 .$$

For the variance term $\sigma_M$ defined over $\mathbb{P}_2$ and $\mathbb{Q}_2$, which is difficult to compute analytically, we approximate its value using empirical estimates obtained from 1,000 runs with 10,000 samples each. Specifically,

$$\sigma_M^2 = 0.0274 .$$

Finally, we compute $m_A^- = 250.5810$, $m_B = 256.6816$ and

$$C_1 = 256.6816 \quad \text{and} \quad C_2 = 509.2431 .$$

# D. Details of Our Experiments

## D.1. Details of the Experiments in Figure 1

### D.1.1. KEY EXPERIMENTAL PARAMETERS OF THE EXPERIMENTS IN FIGURE 1

Figure 1 is generated from a family of continuous Gaussian mixture distributions constructed so that the population MMD remains fixed while the RKHS norms of the individual mean embeddings vary. Throughout this experiment, we use the Gaussian kernel

$$\kappa(\boldsymbol{x}, \boldsymbol{x}') = \exp\left(-\frac{\|\boldsymbol{x} - \boldsymbol{x}'\|_2^2}{2\gamma^2}\right), \qquad \gamma = 1.$$

The target population MMD value is fixed as

$$\|\boldsymbol{\mu}_{\mathbb{P}} - \boldsymbol{\mu}_{\mathbb{Q}}\|_{\mathcal{H}_\kappa}^2 = 0.005.$$

Each distribution is a mixture of $L = 30$ Gaussian components in dimension $D = 30$. The component covariance is isotropic:

$$\mathcal{N}(\boldsymbol{z}_i, \tau^2 \mathbb{I}_D), \qquad \tau = 0.02.$$

The component centers $\{\boldsymbol{z}_i\}_{i=1}^L$ are placed at the vertices of a regular simplex in $\mathbb{R}^{30}$. For each value of the parameter $\rho$, the pairwise distance between simplex vertices is chosen so that the Gaussian-kernel similarity between distinct smoothed components satisfies

$$\frac{B_\rho}{A} = \rho,$$

where

$$A = \mathbb{E}_{\boldsymbol{x}, \boldsymbol{x}' \sim \mathcal{N}(\boldsymbol{z}_i, \tau^2 I_D)}[\kappa(\boldsymbol{x}, \boldsymbol{x}')] = \left(\frac{\gamma^2}{\gamma^2 + 2\tau^2}\right)^{D/2}$$

is the within-component kernel similarity, and $B_\rho = A\rho$ is the between-component similarity.

For a given $\rho$, the mixture distributions are

$$\mathbb{P}_\rho = \sum_{i=1}^L \boldsymbol{p}_i \mathcal{N}(\boldsymbol{z}_i, \tau^2 I_D), \qquad \mathbb{Q}_\rho = \sum_{i=1}^L \boldsymbol{q}_i \mathcal{N}(\boldsymbol{z}_i, \tau^2 I_D).$$

Their mixture weights are defined by

$$\boldsymbol{p} = \boldsymbol{u} + s\boldsymbol{v}, \qquad \boldsymbol{q} = \boldsymbol{u} - s\boldsymbol{v},$$

where $\boldsymbol{u} = (1/L, \ldots, 1/L)$, and $\boldsymbol{v} \in \mathbb{R}^L$ is a balanced contrast vector satisfying

$$\sum_{i=1}^L \boldsymbol{v}_i = 0, \qquad \|\boldsymbol{v}\|_2 = 1.$$

In the implementation, since $L = 30$, we take

$$\boldsymbol{v} = \frac{1}{\sqrt{L}}(\underbrace{1, \ldots, 1}_{L/2}, \underbrace{-1, \ldots, -1}_{L/2}).$$

The scalar $s$ is chosen as

$$s = \sqrt{\frac{0.005}{4(A - B_\rho)}}.$$

This choice ensures that

$$\|\boldsymbol{\mu}_{\mathbb{P}_\rho} - \boldsymbol{\mu}_{\mathbb{Q}_\rho}\|_{\mathcal{H}_\kappa}^2 = 0.005$$

for all values of $\rho$ used in the experiment.

Under this construction, the squared RKHS norms of the two mean embeddings are equal and given by

$$\|\boldsymbol{\mu}_{\mathbb{P}_\rho}\|_{\mathcal{H}_\kappa}^2 = \|\boldsymbol{\mu}_{\mathbb{Q}_\rho}\|_{\mathcal{H}_\kappa}^2 = \frac{A + (L-1)A\rho}{L} + \frac{0.005}{4}.$$

Thus, increasing $\rho$ increases the individual embedding norms while keeping the MMD fixed. The values of $\rho$ are chosen to cover a range of distribution norms, and the points corresponding exactly to norms $0.4$ and $0.7$ are included for the visualizations in subfigures (a) and (b).

For each value of $\rho$, we draw $m = 50$ samples from $\mathbb{P}_\rho$ and $m = 50$ samples from $\mathbb{Q}_\rho$. The MMD statistic and the NAMMD statistic are computed using the same Gaussian kernel with bandwidth $\gamma = 1$. The reported $p$-values are obtained by a permutation test with 300 random permutations for each repeated trial. To reduce Monte Carlo variability, the whole sampling and testing procedure is repeated 300 times for each $\rho$. Figure 1 reports the median $p$-value across repetitions, with the shaded region corresponding to the interquartile range.

Subfigures (a) and (b) display representative samples from the same Gaussian mixture construction at

$$\|\boldsymbol{\mu}_{\mathbb{P}}\|_{\mathcal{H}_\kappa}^2 = \|\boldsymbol{\mu}_{\mathbb{Q}}\|_{\mathcal{H}_\kappa}^2 = 0.4 \quad \text{and} \quad 0.7,$$

respectively. Since the data are generated in $D = 30$ dimensions, the samples are projected onto two dimensions only for visualization. This projection is used solely for plotting and is not used in computing either the test statistics or the $p$-values.

### D.1.2. RELATIONSHIP BETWEEN THE $p$-VALUE OF THE MMD ESTIMATOR AND THE RKHS NORMS OF THE DISTRIBUTIONS

As shown in Figure 1c, for the distribution pairs considered in this experiment, the empirical $p$-values of the MMD estimator tend to decrease as the RKHS norms of the distributions increase, even though the population MMD value is kept fixed. Specifically, the construction fixes

$$\|\boldsymbol{\mu}_{\mathbb{P}} - \boldsymbol{\mu}_{\mathbb{Q}}\|_{\mathcal{H}_\kappa}^2 = 0.005,$$

while varying

$$\|\boldsymbol{\mu}_{\mathbb{P}}\|_{\mathcal{H}_\kappa}^2 \quad \text{and} \quad \|\boldsymbol{\mu}_{\mathbb{Q}}\|_{\mathcal{H}_\kappa}^2.$$

In this empirical setting, a smaller permutation $p$-value indicates that the observed MMD statistic is less likely under the permutation null distribution. Thus, although the population MMD is fixed, the simulated distribution pairs with larger RKHS norms are observed to be easier to distinguish in the two-sample test.

One possible explanation for this empirical trend is the change in the variability of the MMD estimator. Figure 6 shows that, in the same experiments, the standard deviation of the MMD estimator under the permutation null decreases as the RKHS norms increase. This suggests that, for these simulated distributions, larger RKHS norms are associated with a more concentrated null distribution of the MMD estimator.

This observation is consistent with the RKHS variance identity. For a bounded translation-invariant kernel of the form

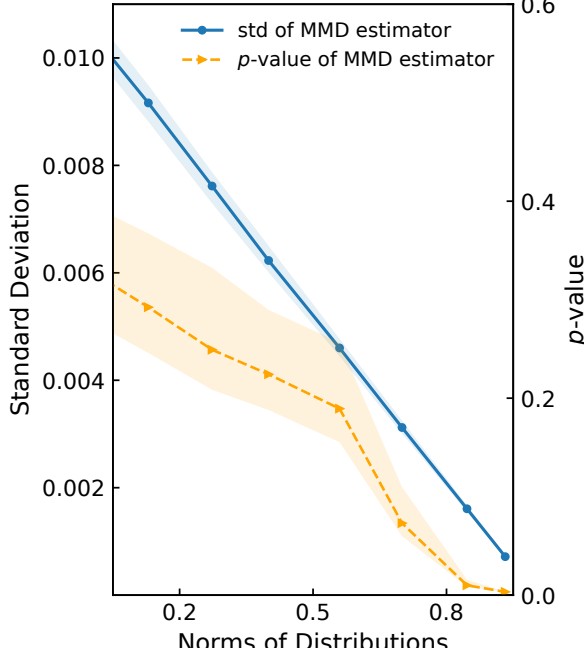

**Figure 6.** Empirical relationship between the $p$-value and the standard deviation of the MMD estimator in two-sample testing, as the RKHS norms of the distributions vary.

$$\kappa(\boldsymbol{x}, \boldsymbol{x}') = \Psi(\boldsymbol{x} - \boldsymbol{x}') \leq K,$$

where $K > 0$, $\Psi$ is positive definite, and $\Psi(\boldsymbol{0}) = K$, the RKHS variance of a distribution can be written as

$$\mathrm{Var}(\mathbb{P}; \kappa) = \mathbb{E}_{\boldsymbol{x} \sim \mathbb{P}} \kappa(\boldsymbol{x}, \boldsymbol{x}) - \|\boldsymbol{\mu}_{\mathbb{P}}\|_{\mathcal{H}_\kappa}^2.$$

For kernels with constant diagonal $\kappa(\boldsymbol{x}, \boldsymbol{x}) = K$, this reduces to

$$\mathrm{Var}(\mathbb{P}; \kappa) = K - \|\boldsymbol{\mu}_{\mathbb{P}}\|_{\mathcal{H}_\kappa}^2,$$

and analogously for $\mathbb{Q}$. Hence, in this class of kernels, increasing the RKHS norm of the mean embedding corresponds to decreasing RKHS variance.

In our simulations, this reduction in variability appears to make the permutation null distribution of the MMD estimator more concentrated around zero. Since the population MMD under the alternative is fixed, the observed MMD statistic can then fall farther into the tail of the null distribution, resulting in smaller empirical permutation $p$-values. We emphasize that this section reports an empirical trend for the constructed Gaussian-mixture examples, rather than a general monotonicity theorem for all distributions or kernels.

### D.2. Details of Experiments with Distributions over Identical Domain

**Data Construction.** Let $\mathbb{P}_n = \{p_1, p_2, ..., p_n\}$ and $\mathbb{Q}_n = \{q_1, q_2, ..., q_n\}$ be two discrete distributions over the same domain $Z = \{\boldsymbol{z}_1, \boldsymbol{z}_2, ..., \boldsymbol{z}_n\} \subseteq \mathbb{R}^d$ such that $\sum_{i=1}^n p_i = 1$ and $\sum_{i=1}^n q_i = 1$. We define the total variation (Canonne et al., 2022) of $\mathbb{P}_n$ and $\mathbb{Q}_n$ as

$$\mathrm{TV}(\mathbb{P}_n, \mathbb{Q}_n) = \sup_{S \subseteq Z} (\mathbb{P}_n(S) - \mathbb{Q}_n(S)) = \frac{1}{2} \sum_{i=1}^n |p_i - q_i| = \frac{1}{2} \|\mathbb{P}_n - \mathbb{Q}_n\|_1 \in [0, 1] .$$

As we can see, the corresponding NAMMD distance can be calculated as

$$
\begin{aligned}
\mathrm{NAMMD}(\mathbb{P}_n, \mathbb{Q}_n; \kappa) &= \frac{\|\boldsymbol{\mu}_{\mathbb{P}_n} - \boldsymbol{\mu}_{\mathbb{Q}_n}\|_{\mathcal{H}_\kappa}^2}{4K - \|\boldsymbol{\mu}_{\mathbb{P}_n}\|_{\mathcal{H}_\kappa}^2 - \|\boldsymbol{\mu}_{\mathbb{Q}_n}\|_{\mathcal{H}_\kappa}^2} \\
&= \frac{\sum_{i,j} p_i p_j \kappa(\boldsymbol{z}_i, \boldsymbol{z}_j) + q_i q_j \kappa(\boldsymbol{z}_i, \boldsymbol{z}_j) - 2 p_i q_j \kappa(\boldsymbol{z}_i, \boldsymbol{z}_j)}{4K - \sum_{i,j} (p_i p_j \kappa(\boldsymbol{z}_i, \boldsymbol{z}_j) + q_i q_j \kappa(\boldsymbol{z}_i, \boldsymbol{z}_j))} .
\end{aligned}
$$

Here, we take the uniform distribution $\mathbb{P}_n = \{1/n, 1/n, ..., 1/n\}$ over sample $Z$, where $p_i = 1/n$ for $i \in \{1, 2, ..., n\}$. We construct discrete distribution $\mathbb{Q}_n$, which is $\epsilon' \in [0, 1]$ total variation away from the uniform distribution $\mathbb{P}_n$, as follows: We initiate the $\mathbb{Q}_n = \mathbb{P}_n$ and randomly split the sample $Z$ into two parts. In the first part, we increase the sample probability of each element by $\epsilon'/n$; and in the second part, we decrease the sample probability of each element by $\epsilon'/n$.

**Testing Threshold for Canonne's test.** Under null hypothesis $H_0' : \mathrm{TV}(\mathbb{P}_n, \mathbb{Q}_n) = \epsilon'$, we set testing threshold $\tau_\alpha'$ as the $(1 - \alpha)$-quantile of the estimated null distribution of Canonne's statistic by resampling method, which repeatedly re-computing the empirical estimator of distance with the samples randomly drawn from $\mathbb{P}_n$ and $\mathbb{Q}_n$.

Specifically, denote by $B$ the iteration number of resampling method. In $b$-th iteration ($b \in [B]$), we randomly draw two samples $X$ and $X'$ from $\mathbb{P}_n$, and two samples $Y$ and $Y'$ from $\mathbb{Q}_n$. The sample sizes are set to be the same as the size of testing samples. Denote by $X_i$ and $X_i'$ the occurrences of $\boldsymbol{z}_i$ in samples $X$ and $X'$ respectively, and let $Y_i$ and $Y_i'$ be the occurrences of $\boldsymbol{z}_i$ in samples $Y$ and $Y'$ respectively. We then calculate the test statistic based on total variation given in Canonne's test as

$$T_b' = \sum_{i=1}^n \frac{(X_i - Y_i)^2 - X_i - Y_i}{\widehat{f}_i},$$

with the term

$$\widehat{f}_i := \max \left\{ |X_i' - Y_i'|, X_i' + Y_i', 1 \right\} .$$

During such process, we obtain $B$ statistics $T_1', T_2', ..., T_B'$ and set testing threshold as

$$\tau_\alpha' = \arg\min_\tau \left\{ \sum_{b=1}^B \frac{\mathbb{I}[T_b' \leq \tau]}{B} \geq 1 - \alpha \right\} .$$

---

**Algorithm 2** Construction of distribution

---

**Input**: Two samples $Z$ and $Z'$, a kernel $\kappa$, step size $\eta$
**Output**: Two samples $Z$ and $Z'$

1: **for** NAMMD$(\mathbb{P}, \mathbb{Q}; \kappa) \neq \epsilon$ **do**
2:    Calculate the objective value $\mathcal{L}(Z, Z' \mid \kappa)$ according to Eqn. 12
3:    Calculate gradient $\nabla \mathcal{L}(Z, Z' \mid \kappa)$
4:    Gradient descend with step size $\eta$ by the Adam method
5: **end for**

---

### D.3. Details of Experiments with Distributions over different Domains

Let $\mathbb{P}$ and $\mathbb{Q}$ be discrete uniform distributions over $Z = \{z_i\}_{i=1}^m$ and $Z' = \{z_i'\}_{i=1}^m$, respectively. As we can see, our NAMMD distance can be calculated as

$$
\begin{aligned}
\text{NAMMD}(\mathbb{P}, \mathbb{Q}; \kappa) &= \frac{\|\boldsymbol{\mu}_{\mathbb{P}} - \boldsymbol{\mu}_{\mathbb{Q}}\|_{\mathcal{H}_\kappa}^2}{4K - \|\boldsymbol{\mu}_{\mathbb{P}}\|_{\mathcal{H}_\kappa}^2 - \|\boldsymbol{\mu}_{\mathbb{Q}}\|_{\mathcal{H}_\kappa}^2} \\
&= \frac{1/m^2 \sum_{i,j} \kappa(z_i, z_j) + \kappa(z_i', z_j') - 2\kappa(z_i, z_j')}{4K - 1/m^2 \sum_{i,j} \left( \kappa(z_i, z_j) + \kappa(z_i', z_j') \right)} .
\end{aligned}
$$

Notably, NAMMD$(\mathbb{P}, \mathbb{Q}; \kappa) = 0$ can be effortlessly achieved by setting $Z = Z'$.

Here, we learn samples $Z$ and $Z'$ given NAMMD$(\mathbb{P}, \mathbb{Q}; \kappa) = \epsilon$ as follows

$$
\mathcal{L}(Z, Z' \mid \kappa) = (\text{NAMMD}(\mathbb{P}, \mathbb{Q}; \kappa) - \epsilon)^2 \tag{12}
$$

We take gradient method (Boyd & Vandenberghe, 2004) for the optimization of Eqn. 12. Algorithm 2 presents the detailed description on optimization. The corresponding calculation of MMD$(\mathbb{P}, \mathbb{Q}; \kappa)$ is given as follows

$$
\begin{aligned}
\text{MMD}^2(\mathbb{P}, \mathbb{Q}; \kappa) &= \|\boldsymbol{\mu}_{\mathbb{P}} - \boldsymbol{\mu}_{\mathbb{Q}}\|_{\mathcal{H}_\kappa}^2 \\
&= 1/m^2 \sum_{i,j} \kappa(z_i, z_j) + \kappa(z_i', z_j') - 2\kappa(z_i, z_j') .
\end{aligned}
$$

### D.4. Details of State-of-the-Art Two-Sample Testing Methods

The details of six state-of-the-art two-sample testing methods used in the experiments (which are summarized in Figure 2) for test power comparison.

- MMDFuse: A fusion of MMD with multiple Gaussian kernels via a soft maximum (Biggs et al., 2023);

- MMD-D: MMD with a learnable Deep kernel (Liu et al., 2020);

- MMDAgg: MMD with aggregation of multiple Gaussian kernels and multiple testing (Schrab et al., 2022);

- AutoTST: Train a binary classifier of AutoML with a statistic about class probabilities (Kübler et al., 2022);

- ME$_{\text{MaBiD}}$: Embeddings over multiple test locations and multiple Mahalanobis kernels (Zhou et al., 2023);

- ACTT: MMDAgg with an accelerated optimization via compression (Domingo-Enrich et al., 2023).

### D.5. Details of our NAMMDFuse

Following the fusing statistics approach (Biggs et al., 2023), we introduce the NAMMDFuse statistic through exponentiation of NAMMD with samples $X$ and $Y$ as follows

$$
\widehat{\text{FUSE}}(X, Y) = \frac{1}{\lambda} \log \left( E_{\kappa \sim \pi(\langle X, Y \rangle)} \left[ \exp \left( \lambda \frac{\widehat{\text{NAMMD}}(X, Y; \kappa)}{\sqrt{\widehat{N}(X, Y)}} \right) \right] \right)
$$

where $\lambda > 0$ and $\widehat{N}(X, Y) = \frac{1}{m(m-1)} \sum_{i \neq j}^{m} \kappa(\mathbf{x}_i, \mathbf{x}_j)^2 + \kappa(\mathbf{y}_i, \mathbf{y}_j)^2$ is permutation invariant. $\pi(\langle X, Y \rangle)$ is the prior distribution on the kernel space $\mathcal{K}$. In experiments, we set the prior distribution $\pi(\langle X, Y \rangle)$ and the kernel space $\mathcal{K}$ to be the same for MMDFuse.

### D.6. Details of Different Kernels

The details of the various kernels used in the experiments (which are summarized in Table 13) for test power comparison in two-sample testing, employing the same kernel for NAMMD and MMD.

- Gaussian: $G(\boldsymbol{x}, \boldsymbol{y}) = \exp(-\|\boldsymbol{x} - \boldsymbol{y}\|^2 / 2\gamma^2)$ for $\gamma > 0$ (Gretton et al., 2012b);

- Laplace: $L(\boldsymbol{x}, \boldsymbol{y}) = \exp(-\|\boldsymbol{x} - \boldsymbol{y}\|_1 / \gamma)$ for $\gamma > 0$ (Biggs et al., 2023);

- Deep: $D(\boldsymbol{x}, \boldsymbol{y}) = [(1 - \lambda)G(\phi_\omega(\boldsymbol{x}), \phi_\omega(\boldsymbol{y})) + \lambda]G(\boldsymbol{x}, \boldsymbol{y})$ for $\lambda > 0$ and network $\phi_\omega$ (Liu et al., 2020);

- Mahalanobis: $M(\boldsymbol{x}, \boldsymbol{y}) = \exp\left(-(\boldsymbol{x} - \boldsymbol{y})^T M(\boldsymbol{x} - \boldsymbol{y}) / 2\gamma^2\right)$ for $\gamma > 0$ and $M \succ 0$ (Zhou et al., 2023).

### D.7. Details of Confidence and Accuracy Gaps

We can test the confidence gap between source dataset $S$ and target dataset $T$ for a model $f$. Let $f(x)$ represent the probability assigned by the model $f$ to the true label. We define the confidence gap as

$$|E_{\boldsymbol{x} \in S}[1 - f(\boldsymbol{x})] - E_{\boldsymbol{x} \in T}[1 - f(\boldsymbol{x})]| . \tag{13}$$

A smaller gap indicates similar model performance in the source and target dataset.

In a similar manner, we can also define the accuracy gap as follows

$$|E_{\boldsymbol{x} \in S}[f(\boldsymbol{x}; y_{\boldsymbol{x}})] - E_{\boldsymbol{x} \in T}[f(\boldsymbol{x}; y_{\boldsymbol{x}})]| , \tag{14}$$

where $f(\boldsymbol{x}; y_{\boldsymbol{x}}) = 1$ if the model $f$ correctly predicts the true label $y_{\boldsymbol{x}}$, and $f(\boldsymbol{x}; y_{\boldsymbol{x}}) = 0$ otherwise.

We present the confidence and accuracy gaps between the original ImageNet and its variants in Table 6, with the values computed using the pre-trained ResNet50 model.

**Table 6.** Confidence and accuracy gaps between the original ImageNet and its variants.

|  | ImageNetsk | ImageNetr | ImageNetv2 | ImageNeta |
|---|---|---|---|---|
| Accuracy Gap | 0.529 | 0.564 | 0.751 | 0.827 |
| Confidence Gap | 0.504 | 0.549 | 0.684 | 0.764 |

## E. Additional Experimental Results

### E.1. Comparison with Total Variation: Sensitivity to Sample Structure.

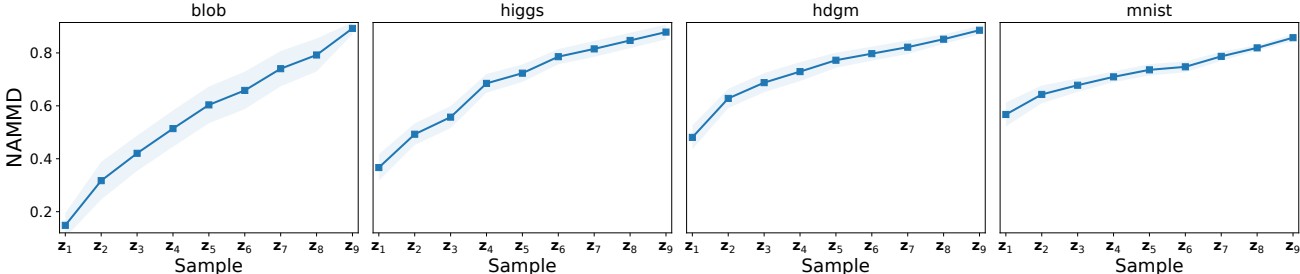

**Figure 7.** The NAMMD distance between $\delta(\boldsymbol{z}_0)$ and $\delta(\boldsymbol{z}_i)$ with $i \in \{1, 2, \ldots, 9\}$.

We demonstrate that our NAMMD better captures the differences between distributions by exploiting intrinsic structures. For each dataset, we sample ten elements and randomly selecting one element to serve as the base $\boldsymbol{z}_0$. The remaining

elements are sorted as $z_1, z_2, ..., z_9$ with $\|z_0 - z_1\|^2 \geq \|z_0 - z_2\|^2 \geq \cdots \geq \|z_0 - z_9\|^2$. For each element $z_i$, we construct the Dirac distribution $\delta_{z_i}$ with support only at element $z_i$, and we calculate the distance $\text{NAMMD}(\delta_{z_0}, \delta_{z_i}, \kappa)$. We repeat this 10 times, using a Gaussian kernel with $\gamma = 1$ for blob, higgs, and hdgm, and $\gamma = 10$ for mnist.

From Figure 7, it is evident that our $\text{NAMMD}(\delta_{z_0}, \delta_{z_i}, \kappa)$ distance increases as $\|z_0 - z_i\|^2$ decrease for all datasets. This is different from previous total variation $\text{TV}(\delta_{z_0}, \delta_{z_i}) = 1$ for $i \in \{1, 2, ..., 9\}$, which merely measures the difference between probability mass functions of two distributions. In comparison, our NAMMD distance can effectively capture intrinsic structures and complex patterns in real-word datasets by leveraging kernel trick.

### E.2. Comparisons on NAMMD and Wasserstein on Discrete Distributions

**Table 7.** Comparisons of test power (mean±std) between NAMMD and Wasserstein under different $\epsilon'$ values.

| Dataset | $\epsilon' = 0.1$ | | $\epsilon' = 0.3$ | | $\epsilon' = 0.5$ | | $\epsilon' = 0.7$ | |
|---|---|---|---|---|---|---|---|---|
| | NAMMD | Wasser. | NAMMD | Wasser. | NAMMD | Wasser. | NAMMD | Wasser. |
| blob | **.968**±.022 | .295±.047 | **.912**±.053 | .343±.037 | **.960**±.020 | .265±.023 | **.961**±.029 | .549±.051 |
| cifar10 | **.919**±.017 | .765±.020 | **.923**±.021 | .721±.022 | **.997**±.002 | .733±.024 | **.999**±.001 | .718±.026 |
| hdgm | **.942**±.023 | .866±.023 | **.946**±.017 | .823±.028 | **.965**±.014 | .821±.027 | **.989**±.004 | .788±.024 |
| higgs | **.908**±.050 | .612±.059 | **.947**±.027 | .427±.045 | **.962**±.023 | .548±.046 | **.995**±.005 | .548±.053 |
| mnist | **.931**±.024 | .867±.020 | **.965**±.007 | .818±.023 | **.997**±.001 | .823±.027 | **1.00**±.000 | .822±.034 |
| Average | **.934**±.027 | .681±.034 | **.939**±.025 | .626±.031 | **.976**±.012 | .638±.030 | **.989**±.008 | .685±.038 |

We compare the test power of DCT using our NAMMD against the Wasserstein distance on *discrete distributions supported on the same finite set*, following exactly the experimental setup of Table 1. As shown in Table 7, NAMMD-based DCT consistently achieves higher test power than the Wasserstein-based approach across different datasets and $\epsilon'$ levels.

### E.3. Comparisons on respectively selected kernels for MMD and NAMMD.

**Table 8.** Comparisons of test power (mean±std) on distribution closeness testing with respect to different NAMMD values, and the bold denotes the highest mean between tests with our NAMMD and original MMD. Notably, different selected kernels are applied for NAMMD and MMD respectively in this table.

| Dataset | $\epsilon = 0.1$ | | $\epsilon = 0.3$ | | $\epsilon = 0.5$ | | $\epsilon = 0.7$ | |
|---|---|---|---|---|---|---|---|---|
| | MMD | NAMMD | MMD | NAMMD | MMD | NAMMD | MMD | NAMMD |
| blob | .939±.009 | **.983**±.004 | .968±.007 | **.991**±.002 | .952±.010 | **.999**±.001 | .934±.010 | **1.00**±.000 |
| higgs | .914±.051 | **.972**±.009 | .934±.056 | **.976**±.007 | .967±.021 | **.994**±.002 | .949±.036 | **.1.00**±.000 |
| hdgm | .925±.071 | **.976**±.005 | .915±.069 | **.978**±.004 | .913±.058 | **.984**±.004 | .938±.052 | **1.00**±.000 |
| mnist | .951±.006 | **.962**±.005 | .955±.032 | **.961**±.021 | .935±.049 | **.967**±.036 | .977±.011 | **.992**±.002 |
| cifar10 | .976±.012 | **.987**±.006 | .971±.007 | **.988**±.003 | .991±.004 | **1.00**±.000 | **1.00**±.000 | **1.00**±.000 |
| Average | .941±.030 | **.976**±.006 | .949±.034 | **.979**±.007 | .952±.028 | **.989**±.009 | .960±.022 | **.998**±.000 |

Similar to Table 2 (where the experiments are performed using the same kernel for both MMD and NAMMD), we conduct experiments with different selected kernels for NAMMD and MMD. For MMD, the kernel selection remains the same as in the experiments in Table 2, and we denote the kernel for MMD as $\kappa^M$. However, for NAMMD, we select the kernel $\kappa^N$ similar to the experiments in Table 2, but with an additional regularization term related to the norms of the original distributions in the dataset (i.e., $4K - \|\mu_{\mathbb{P}}\|^2_{\mathcal{H}_\kappa} - \|\mu_{\mathbb{Q}}\|^2_{\mathcal{H}_\kappa}$) during the optimization. Notably, these kernel selection methods are heuristic for distribution closeness testing, as obtaining a test power estimator for distribution closeness testing with multiple distribution pairs and selecting an optimal global kernel for distribution closeness testing based on the estimator remain open questions and poses a significant challenge. We use $\kappa^N$ for the construction distribution pairs $(\mathbb{P}_1, \mathbb{Q}_1)$ and $(\mathbb{P}_2, \mathbb{Q}_2)$. Following Definition 8, we perform NAMMD distribution closeness testing with $\kappa^N$ and MMD distribution closeness testing with $\kappa^M$ respectively. Table 8 summarizes the average test powers and standard deviations of NAMMD distribution closeness testing and MMD distribution closeness testing. It is evident that our NAMMD test achieves better performance than the MMD test, and this improvement when using different selected kernels for NAMMD and MMD can be explained by the analysis for distribution closeness testing based on Theorem 9.

## E.4. Type-I Error Experiments

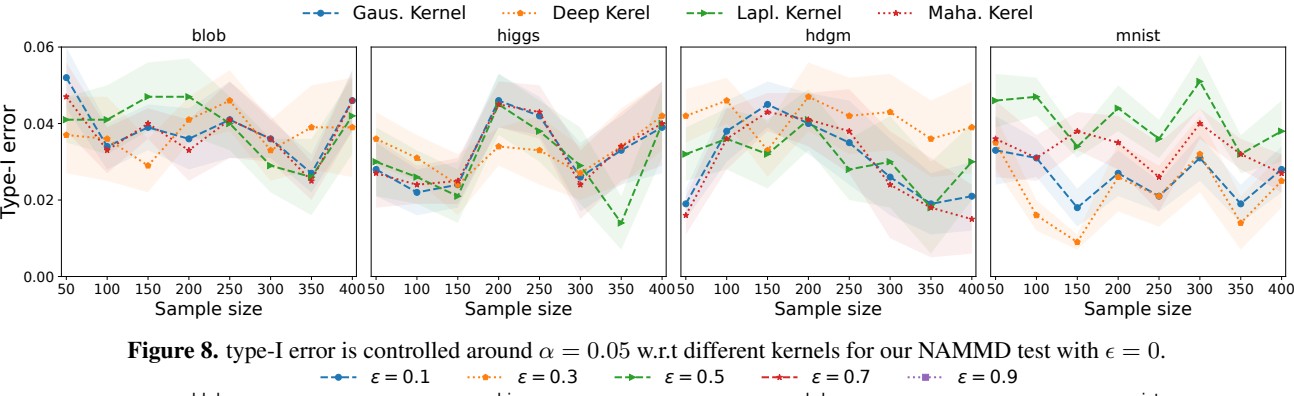

**Figure 8.** type-I error is controlled around $\alpha = 0.05$ w.r.t different kernels for our NAMMD test with $\epsilon = 0$.

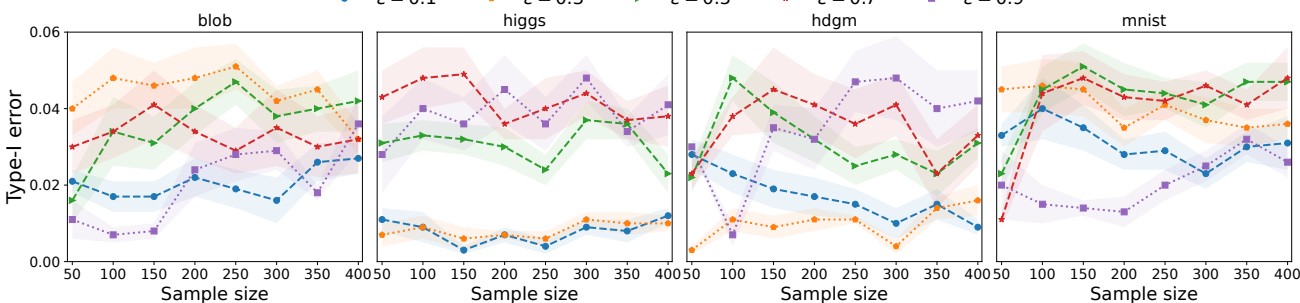

**Figure 9.** The type-I error is controlled around $\alpha = 0.05$ w.r.t different $\epsilon \in (0, 1)$ for our NAMMD test.

From Figure 8, it is evident that the type-I error of our NAMMD test is controlled around $\alpha = 0.05$ with respect to different kernels and datasets in two-sample testing (i.e. distribution closeness testing with $\epsilon = 0$) by using permutation tests. In a similar manner, Figure 9 shows that the type-I error of our NAMMD test is controlled around $\alpha = 0.05$ with respect to different $\epsilon \in (0, 1)$ and datasets in distribution closeness testing, where we derive the testing threshold based on asymptotic distribution. These results are nicely in accordance with Theorem 6.

**Table 9.** type-I error (mean±std) across different sample sizes on ImageNet-SK.

|  | 500 | 1000 | 1500 | 2000 |
|---|---|---|---|---|
| MMD | .049±.012 | .051±.007 | .040±.015 | .054±.012 |
| NAMMD | .047±.013 | .052±.006 | .042±.012 | .047±.012 |

**Table 10.** type-I error (mean±std) across different sample sizes on ImageNet-V2.

|  | 500 | 1000 | 1500 | 2000 |
|---|---|---|---|---|
| MMD | .051±.007 | .044±.005 | .042±.007 | .032±.004 |
| NAMMD | .047±.006 | .044±.004 | .040±.008 | .036±.006 |

**Table 11.** type-I error (mean±std) across different sample sizes on ImageNet-A.

|  | 500 | 1000 | 1500 | 2000 |
|---|---|---|---|---|
| MMD | .047±.005 | .030±.005 | .030±.004 | .039±.004 |
| NAMMD | .047±.005 | .027±.003 | .037±.006 | .040±.006 |

**Table 12.** type-I error (mean±std) across different sample sizes on ImageNet-R.

|  | 500 | 1000 | 1500 | 2000 |
|---|---|---|---|---|
| MMD | .039±.005 | .051±.006 | .042±.006 | .040±.006 |
| NAMMD | .041±.006 | .045±.006 | .048±.005 | .047±.007 |

Following Definition 8, we take ImageNet as $\mathbb{P}_1$ and $\mathbb{P}_2$, and sequentially assign each of its variants (ImageNet-SK, ImageNet-V2, ImageNet-A, and ImageNet-R) as both $\mathbb{Q}_1$ and $\mathbb{Q}_2$. In this setup, the null hypothesis holds because the distributional distance between $(\mathbb{P}_2, \mathbb{Q}_2)$ is not substantially larger than that between $(\mathbb{P}_1, \mathbb{Q}_1)$, and therefore the alternative hypothesis (i.e., $d(\mathbb{P}_2, \mathbb{Q}_2) > \epsilon$ with $\epsilon = d(\mathbb{P}_1, \mathbb{Q}_1)$) does not arise. From Table 9 to Table 12, both NAMMD- and MMD-based DCT maintain type-I error around the significance level $\alpha = 0.05$.

### E.5. Comparisons on various kernels.

For further comparison, we evaluate our NAMMD test (with $\epsilon = 0$) against the MMD test in terms of test power with the same kernel. We perform this experiments across four frequently used kernels (Appendix D.6): 1). Gaussian kernel (Gretton et al., 2012b); 2). Laplace kernel (Biggs et al., 2023); 3). Deep kernel (Liu et al., 2020); 4). Mahalanobis kernel (Zhou et al., 2023). Following (Zhou et al., 2023; Liu et al., 2020), we learn kernels on a subset of each available dataset for 2000 epochs, and then test on 100 random same size subsets from remaining dataset. The ratio is set to $1 : 1$ for training and test sample sizes. We repeat such process 10 times for each dataset. For our NAMMD test, the null hypothesis is NAMMD$(\mathbb{P}, \mathbb{Q}, \kappa) = 0$, and we apply permutation test.

**Table 13.** Comparisons of test power (mean±std) on two-sample testing with the same kernel, and the bold denotes the highest mean between our NAMMD test and the original MMD test.

| Dataset | Gaus. Kernel | | Maha. Kernel | | Deep Kernel | | Lapl. Kernel | |
|---|---|---|---|---|---|---|---|---|
|  | MMD | NAMMD | MMD | NAMMD | MMD | NAMMD | MMD | NAMMD |
| blob | .600±.090 | **.616±.090** | **1.00±.000** | **1.00±.000** | .859±.084 | **.863±.083** | .359±.088 | **.364±.088** |
| higgs | .563±.073 | **.566±.075** | .904±.087 | **.905±.086** | .796±.091 | **.797±.091** | .556±.062 | **.581±.062** |
| hdgm | .707±.042 | **.713±.041** | .801±.097 | **.805±.095** | .332±.087 | **.334±.086** | .090±.012 | **.100±.013** |
| mnist | .405±.019 | **.411±.020** | .970±.013 | **.975±.012** | .462±.100 | **.467±.098** | .873±.016 | **.881±.010** |
| cifar10 | .219±.017 | **.222±.020** | .984±.007 | **.987±.006** | .997±.003 | **1.00±.000** | .998±.002 | **1.00±.000** |
| Average | .499±.048 | **.506±.049** | .932±.041 | **.934±.040** | .689±.073 | **.692±.072** | .575±.036 | **.585±.035** |

Table 13 summarizes the average of test powers and standard deviations of our NAMMD test and the MMD test with the same kernel. NAMMD test achieves better performance than original MMD test as for Gaussian, Laplace, Mahalanobis and Deep kernels. It is because scaling maximum mean discrepancy with the norms of mean embeddings improves the effectiveness of NAMMD test in two-sample testing.

### E.6. Sensitivity to Kernel Bandwidth

We further evaluate bandwidth sensitivity in a two-dimensional Gaussian DCT setting. Each pair is given by $\mathbb{P} = \mathcal{N}(-\boldsymbol{\delta}/2, \sigma^2 I_d)$ and $\mathbb{Q} = \mathcal{N}(\boldsymbol{\delta}/2, \sigma^2 I_d)$, with $d = 2$ and $\|\boldsymbol{\delta}\|_2 = 0.20$. The boundary pair defining the tolerance uses $\sigma^2 = 0.30$, while the alternative pair uses $\sigma^2 = 0.10$. For each bandwidth, we calibrate the rejection threshold using 1000 independent repetitions from the boundary pair and estimate power using 1000 independent repetitions from the alternative pair, with sample size $n = 150$ and significance level $\alpha = 0.05$. The Gaussian kernel bandwidth is varied over $0.1h_0, 0.3h_0, 0.5h_0, h_0, 2.0h_0$, where $h_0$ denotes the median-heuristic bandwidth.

**Table 14.** Power of MMD and NAMMD under different Gaussian kernel bandwidths. Here, $h_0$ denotes the median-heuristic bandwidth.

| Method | $0.1h_0$ | $0.3h_0$ | $0.5h_0$ | $h_0$ | $2.0h_0$ |
|---|---|---|---|---|---|
| MMD | .792 ± .013 | .900 ± .009 | .772 ± .013 | .330 ± .015 | .037 ± .006 |
| NAMMD | .800 ± .013 | .924 ± .008 | .858 ± .011 | .519 ± .016 | .076 ± .008 |

As shown in Table 14, both MMD and NAMMD are sensitive to bandwidth choices: the empirical power decreases when

the bandwidth is too small or too large. These results suggest that NAMMD has a bandwidth-sensitivity pattern similar to MMD. Hence, following prior work, we use the data-adaptive bandwidth selection strategy in Appendix B.5.

### E.7. Sensitivity to RKHS Norm Estimation Error

Since NAMMD rescales MMD using the RKHS norms of the two distributions, we further evaluate its sensitivity to RKHS norm estimation error using Gaussian distributions with known population quantities. Specifically, we consider $\mathbb{P} = \mathcal{N}(-\boldsymbol{\delta}/2, \sigma^2 I_d)$ and $\mathbb{Q} = \mathcal{N}(\boldsymbol{\delta}/2, \sigma^2 I_d)$ under the Gaussian kernel with bandwidth 1, where $\|\boldsymbol{\delta}\|_2 = 0.5$. For each dimension $d \in \{2, 200, 20000\}$ and each target RKHS norm level $\|\boldsymbol{\mu}_{\mathbb{P}}\|^2_{\mathcal{H}_\kappa} = \|\boldsymbol{\mu}_{\mathbb{Q}}\|^2_{\mathcal{H}_\kappa} \in \{0.2, 0.4, 0.6, 0.8\}$, we choose $\sigma^2$ according to the closed-form RKHS norm of a Gaussian distribution. We then estimate MMD and NAMMD from two samples with $n = 10$ observations per distribution. We report the relative root mean squared error (relative RMSE), defined as the root mean squared estimation error divided by the corresponding population value. The results are averaged over 300 independent repetitions.

**Table 15.** Relative RMSE of MMD and NAMMD under different RKHS norm levels and dimensions.

| Dimension | Method | $\text{norm}^2 = 0.2$ | $\text{norm}^2 = 0.4$ | $\text{norm}^2 = 0.6$ | $\text{norm}^2 = 0.8$ |
|-----------|--------|------|------|------|------|
| $d = 2$ | MMD | 7.462 | 2.311 | 1.172 | 0.633 |
| $d = 2$ | NAMMD | 7.586 | 2.382 | 1.189 | 0.642 |
| $d = 200$ | MMD | 0.216 | 0.141 | 0.103 | 0.060 |
| $d = 200$ | NAMMD | 0.217 | 0.142 | 0.105 | 0.061 |
| $d = 20000$ | MMD | 0.019 | 0.013 | 0.010 | 0.006 |
| $d = 20000$ | NAMMD | 0.019 | 0.013 | 0.010 | 0.006 |

As shown in Table 15, NAMMD has relative RMSE comparable to that of MMD across the tested norm levels and dimensions. This suggests that, in these settings, the additional use of RKHS norm estimates does not make NAMMD substantially more sensitive to estimation error than MMD.

### E.8. Runtime comparison.

**Table 16.** Comparisons of runtime (seconds) on two-sample testing with permutation test, corresponding to the experiments shown in Figure 2.

| Samp. Size | ACTT | AutoTST | MEmabid | MMD-D | MMDAgg | MMDFuse | NAMMDFuse |
|-----------|------|---------|---------|-------|--------|---------|-----------|
| 50 | 35.918 | 681.669 | 45.945 | 303.413 | 13.621 | 9.340 | 9.602 |
| 100 | 42.035 | 707.498 | 53.125 | 308.542 | 14.742 | 11.686 | 12.107 |
| 150 | 44.429 | 707.368 | 82.473 | 446.897 | 16.037 | 13.298 | 13.744 |
| 200 | 44.981 | 734.686 | 83.341 | 448.066 | 17.031 | 14.015 | 14.388 |
| 250 | 45.129 | 731.910 | 83.877 | 451.478 | 20.730 | 16.573 | 16.921 |
| 300 | 46.402 | 750.984 | 158.909 | 747.656 | 21.316 | 19.404 | 19.989 |
| 350 | 46.077 | 809.401 | 159.829 | 743.727 | 22.301 | 23.441 | 23.439 |
| 400 | 46.994 | 847.017 | 232.811 | 1025.473 | 23.655 | 27.632 | 27.845 |

Table 16 presents the time costs of the proposed permutation-based method, NAMMDFuse (which aggregates multiple NAMMD statistics with different kernels, as shown in Appendix D.5). NAMMDFuse exhibits similar time costs to MMDFuse and is significantly faster than most baseline methods. Recall that the NAMMD is defined as:

$$
\begin{aligned}
\text{NAMMD}(\mathbb{P}, \mathbb{Q}; \kappa) &= \frac{\|\boldsymbol{\mu}_{\mathbb{P}} - \boldsymbol{\mu}_{\mathbb{Q}}\|^2_{\mathcal{H}_\kappa}}{4K - \|\boldsymbol{\mu}_{\mathbb{P}}\|^2_{\mathcal{H}_\kappa} - \|\boldsymbol{\mu}_{\mathbb{Q}}\|^2_{\mathcal{H}_\kappa}} \\
&= \frac{\|\boldsymbol{\mu}_{\mathbb{P}}\|^2_{\mathcal{H}_\kappa} + \|\boldsymbol{\mu}_{\mathbb{Q}}\|^2_{\mathcal{H}_\kappa} - 2\langle \boldsymbol{\mu}_{\mathbb{P}}, \boldsymbol{\mu}_{\mathbb{Q}}\rangle_{\mathcal{H}_\kappa}}{4K - \|\boldsymbol{\mu}_{\mathbb{P}}\|^2_{\mathcal{H}_\kappa} - \|\boldsymbol{\mu}_{\mathbb{Q}}\|^2_{\mathcal{H}_\kappa}} \\
&= \frac{E_{\boldsymbol{x},\boldsymbol{x}'\sim\mathbb{P}^2}[\kappa(\boldsymbol{x}, \boldsymbol{x}')] + E_{\boldsymbol{y},\boldsymbol{y}'\sim\mathbb{Q}^2}[\kappa(\boldsymbol{y}, \boldsymbol{y}')] - 2E_{\boldsymbol{x}\sim\mathbb{P}, \boldsymbol{y}\sim\mathbb{Q}}[\kappa(\boldsymbol{x}, \boldsymbol{y}')]}{4K - E_{\boldsymbol{x},\boldsymbol{x}'\sim\mathbb{P}^2}[\kappa(\boldsymbol{x}, \boldsymbol{x}')] - E_{\boldsymbol{y},\boldsymbol{y}'\sim\mathbb{Q}^2}[\kappa(\boldsymbol{y}, \boldsymbol{y}')]} .
\end{aligned}
$$

where the kernel $\kappa(\boldsymbol{x}, \boldsymbol{x}') = \Psi(\boldsymbol{x} - \boldsymbol{x}')$ is positive-definite with $\Psi(\boldsymbol{0}) = K$ and $\Psi(\boldsymbol{x} - \boldsymbol{x}') \leq K$ for all $\boldsymbol{x}, \boldsymbol{x}'$, and $K > 0$. Notably, the scaling term of NAMMD, $4K - \|\boldsymbol{\mu}_{\mathbb{P}}\|^2_{\mathcal{H}_\kappa} - \|\boldsymbol{\mu}_{\mathbb{Q}}\|^2_{\mathcal{H}_\kappa}$, can be efficiently computed using intermediate quantities

from MMD, thus incurring negligible additional cost. Overall, the computational overhead introduced by NAMMD is minimal. In the formulation of NAMMD, all computations in the RKHS can be expressed as inner products, often computed via pairwise distances. This avoids the need to explicitly compute RKHS embeddings and helps reduce computational complexity. During the permutation test, we precompute the pairwise inner products and reuse them by rearranging the indices to obtain permutation results, eliminating the need to recompute them for each permutation. This strategy can be implemented efficiently.

For DCT, the runtime has a similar structure. The dominant cost of both MMD-based DCT and NAMMD-based DCT is the computation of the kernel matrices $\mathbf{K_{XX}}$, $\mathbf{K_{YY}}$, and $\mathbf{K_{XY}}$, together with the corresponding $U$-statistic quantities used to estimate the test statistic and its asymptotic variance. NAMMD does not require additional kernel evaluations beyond those used by MMD. Its denominator, $4K - \|\boldsymbol{\mu}_\mathbb{P}\|^2_{\mathcal{H}_\kappa} - \|\boldsymbol{\mu}_\mathbb{Q}\|^2_{\mathcal{H}_\kappa}$, is computed from the same within-sample kernel averages used in MMD. Similarly, the asymptotic threshold for NAMMD-based DCT is obtained from the estimated asymptotic variance and a normal quantile, and the required variance estimators are computed from the same precomputed kernel matrices. Thus, compared with MMD-based DCT, NAMMD-based DCT mainly introduces additional matrix reductions and arithmetic operations, rather than additional kernel computations. Consequently, the computational overhead of NAMMD-based DCT over MMD-based DCT is small in practice, and no repeated permutation procedure is required for obtaining DCT threshold.

# F. Limitation Statement

Our analysis in this paper focuses on kernels of the form $\kappa(\boldsymbol{x}, \boldsymbol{x}') = \Psi(\boldsymbol{x} - \boldsymbol{x}') \leq K$ with a positive-definite $\Psi(\cdot)$ and $\Psi(\mathbf{0}) = K$, including Laplace (Biggs et al., 2023), Mahalanobis (Zhou et al., 2023) and Deep kernels (Liu et al., 2020) (frequently used in kernel-based hypothesis testing). For these kernels, *a larger norm of mean embedding $\|\boldsymbol{\mu}_\mathbb{P}\|^2_{\mathcal{H}_\kappa}$ indicates a smaller variance $Var(\mathbb{P}; \kappa) = K - \|\boldsymbol{\mu}_\mathbb{P}\|^2_{\mathcal{H}_\kappa}$, which corresponds to a more tightly concentrated distribution in the RKHS.* Leveraging this property, we gain the insight that two distributions can be separated more effectively at the same MMD distance with larger norms as discussed in Appendix D.1.2. Hence, we scale MMD using $4K - \|\boldsymbol{\mu}_\mathbb{P}\|^2_{\mathcal{H}_\kappa} - \|\boldsymbol{\mu}_\mathbb{Q}\|^2_{\mathcal{H}_\kappa}$, making the new NAMMD increase with the norms $\|\boldsymbol{\mu}_\mathbb{P}\|^2_{\mathcal{H}_\kappa}$ and $\|\boldsymbol{\mu}_\mathbb{Q}\|^2_{\mathcal{H}_\kappa}$. Figure 1c and 1d demonstrate that our NAMMD exhibits a stronger correlation with the $p$-value in testing, while MMD is held constant. We also prove that scaling improves NAMMD's effectiveness as a closeness measure in Theorem 9.

However, all these improvements rely on the property that *"A larger norm of mean embedding $\|\boldsymbol{\mu}_\mathbb{P}\|^2_{\mathcal{H}_\kappa}$ indicates a smaller variance $Var(\mathbb{P}; \kappa) = K - \|\boldsymbol{\mu}_\mathbb{P}\|^2_{\mathcal{H}_\kappa}$, which corresponds to a more tightly concentrated distribution $\mathbb{P}$".* The proposed method may not work well for kernels where the embedding norm of distribution may increases as the data variance increases. For these kernels, the "less informative" of MMD still arises when assessing the closeness levels for multiple distribution pairs with the same kernel, i.e., MMD value can be the same for many pairs of distributions that have different norms in the same RKHS. We will demonstrate this by further considering two other types of kernels as follows.

**Unbounded kernels for bounded data**: For polynomial kernels of the form

$$\kappa(\mathbf{x}, \mathbf{x}') = (\mathbf{x}^T \mathbf{x}' + c)^d \,,$$

We define $\mathbb{P}_1 = \{\frac{1}{4}, \frac{3}{4}\}$ and $\mathbb{Q}_1 = \{\frac{1}{2}, \frac{1}{2}\}$ be discrete distributions over vector domains $\{(\sqrt{c}, ..., 0), (-\sqrt{c}, ..., 0)\}$, respectively. Furthermore, we define $\mathbb{P}_2 = \{\frac{3}{4}, \frac{1}{4}\}$ and $\mathbb{Q}_2 = \{1, 0\}$ be discrete distributions over domains $\{(\sqrt{c}, ..., 0), (-\sqrt{c}, ..., 0)\}$. It is evident that

$$\text{MMD}^2(\mathbb{P}_1, \mathbb{Q}_1; \kappa) = \text{MMD}^2(\mathbb{P}_2, \mathbb{Q}_2; \kappa) = \frac{1}{8}(2c)^d \,,$$

with different norms for distributions pairs $\|\boldsymbol{\mu}_{\mathbb{P}_1}\|^2_{\mathcal{H}_\kappa} + \|\boldsymbol{\mu}_{\mathbb{Q}_1}\|^2_{\mathcal{H}_\kappa} = \frac{9}{8}(2c)^d$, and $\|\boldsymbol{\mu}_{\mathbb{P}_2}\|^2_{\mathcal{H}_\kappa} + \|\boldsymbol{\mu}_{\mathbb{Q}_2}\|^2_{\mathcal{H}_\kappa} = \frac{13}{8}(2c)^d$. Specifically, we have $\|\boldsymbol{\mu}_{\mathbb{P}_1}\|^2_{\mathcal{H}_\kappa} = \frac{5}{8}(2c)^d$, $\|\boldsymbol{\mu}_{\mathbb{Q}_1}\|^2_{\mathcal{H}_\kappa} = \frac{1}{2}(2c)^d$, $\|\boldsymbol{\mu}_{\mathbb{P}_2}\|^2_{\mathcal{H}_\kappa} = \frac{5}{8}(2c)^d$ and $\|\boldsymbol{\mu}_{\mathbb{Q}_2}\|^2_{\mathcal{H}_\kappa} = (2c)^d$.

In a similar manner, for matrix products kernels of the form

$$\kappa(\mathbf{x}, \mathbf{x}') = (\mathbf{x}^T M \mathbf{x}' + c)^d \,,$$

and denote by $M_{11}$ the element in the first row and first column of the matrix $M$. We define $\mathbb{P}_1 = \{\frac{1}{4}, \frac{3}{4}\}$ and $\mathbb{Q}_1 = \{\frac{1}{2}, \frac{1}{2}\}$ over vector domains $\{(\sqrt{c/M_{11}}, ..., 0), (-\sqrt{c/M_{11}}, ..., 0)\}$, respectively. Furthermore, we define $\mathbb{P}_2 = \{\frac{3}{4}, \frac{1}{4}\}$ and $\mathbb{Q}_2 = \{1, 0\}$ over domains $\{(\sqrt{c/M_{11}}, ..., 0), (-\sqrt{c/M_{11}}, ..., 0)\}$. We obtain the same results as for polynomial kernels.

**Kernels with a positive limit at infinity**: Using the kernel as $\kappa(\mathbf{x}, \mathbf{x}') = \exp(-\frac{\|\mathbf{x}-\mathbf{x}'\|^2}{2\gamma})$ when $\|\mathbf{x} - \mathbf{x}'\|_\infty < K$, and otherwise $\kappa(\mathbf{x}, \mathbf{x}')$ with positive constants $K$ and $c$. We define $\mathbb{P}_1 = \{\frac{1}{4}, \frac{3}{4}\}$ and $\mathbb{Q}_1 = \{\frac{3}{4}, \frac{1}{4}\}$ over vector domains $\{(K, ..., 0), (4K, ..., 0)\}$, respectively. Furthermore, we define $\mathbb{P}_2 = \{\frac{1}{2}, \frac{1}{2}\}$ and $\mathbb{Q}_2 = \{1, 0\}$ over domains $\{(K, ..., 0), (4K, ..., 0)\}$. It is evident that

$$\text{MMD}^2(\mathbb{P}_1, \mathbb{Q}_1; \kappa) = \text{MMD}^2(\mathbb{P}_2, \mathbb{Q}_2; \kappa) = \frac{1}{2}(1 - c) \,,$$

with different norms for pairs $\|\boldsymbol{\mu}_{\mathbb{P}_1}\|_{\mathcal{H}_\kappa} + \|\boldsymbol{\mu}_{\mathbb{Q}_1}\|^2_{\mathcal{H}_\kappa} = \frac{5+3c}{4}$, and $\|\boldsymbol{\mu}_{\mathbb{P}_2}\|^2_{\mathcal{H}_\kappa} + \|\boldsymbol{\mu}_{\mathbb{Q}_2}\|^2_{\mathcal{H}_\kappa} = \frac{3+c}{2}$. Specifically, we have $\|\boldsymbol{\mu}_{\mathbb{P}_1}\|^2_{\mathcal{H}_\kappa} = \frac{5+3c}{8}$, $\|\boldsymbol{\mu}_{\mathbb{Q}_1}\|^2_{\mathcal{H}_\kappa} = \frac{5+3c}{8}$, $\|\boldsymbol{\mu}_{\mathbb{P}_2}\|^2_{\mathcal{H}_\kappa} = \frac{1+c}{2}$ and $\|\boldsymbol{\mu}_{\mathbb{Q}_2}\|^2_{\mathcal{H}_\kappa} = 1$.

For these kernels, the relationship between the norm of mean embedding and the variance of distribution is not monotonic, where a smaller norm of mean embedding may indicate a smaller variance or a larger variance, depending on the properties of the data distributions. Hence, when using these kernels for distribution closeness testing, mitigating the issue (i.e., MMD being the same for multiple pairs of distributions with different norms in the same RKHS) by incorporating norms of distributions becomes more challenging, potentially leading to a more complex distance design.

