# OpenReview forum: "Are Two Datasets Close Enough With Statistical Significance? A Kernel Distributional Closeness Testing Approach"
_ICML.cc/2026/Conference — ICML 2026 regular_

### Official Review · Reviewer_d8VP · 2026-02-13

**Soundness:** 3
**Presentation:** 4
**Significance:** 3
**Originality:** 4
**Overall Recommendation:** 5
**Confidence:** 3

**Summary:**

The article's broad area is about statistical hypothesis testing and kernel-based methods for comparing probability distributions, with a focus on extending distribution closeness testing (DCT) to complex data domains.
The submission addresses the following issue:
Existing DCT frameworks are largely tailored to discrete distributions and struggle to handle high-dimensional or structured data such as images.
To overcome this limitation, the authors introduce the norm-adaptive maximum mean discrepancy (NAMMD), a modification of MMD that rescales discrepancies using RKHS norms to better reflect relative closeness between distribution pairs.
Extensive experiments on synthetic and real datasets demonstrate empirical advantages in both DCT and two-sample testing settings.

**Compliance With Llm Reviewing Policy:**

Affirmed.

**Final Justification:**

Overall, the article's broad area consists of statistical hypothesis testing and distribution comparison, and the submission addresses a pertinent issue in extending distribution closeness testing (DCT) to complex, continuous data.

The paper makes a clear and well-motivated contribution: identifying a genuine limitation of MMD in the DCT setting and proposing a principled fix (NAMMD) with solid theoretical backing and thorough experiments. The theoretical results --- type-I error control, sample complexity bounds, and the power advantage over MMD-based DCT --- are rigorous, and the empirical evaluation across synthetic and real-world benchmarks is convincing.

The rebuttal addressed my concerns.

**Key Questions For Authors:**

Do the authors have any ideas toward a kernel selection criterion specifically optimized for comparing closeness across multiple distribution pairs, rather than a single pair as in TST?

How sensitive is NAMMD to estimation error in RKHS norms, especially in small-sample or high-dimensional regimes?

While the reference-pair strategy is intuitive, could the authors provide more systematic guidance or heuristics for selecting $\epsilon$ in applications where no obvious reference pair exists?

**Limitations:**

Yes.

**Strengths And Weaknesses:**

Strengths:

The paper articulates well why classical DCT methods are insufficient for complex data and provides an intuitive explanation of MMD’s limitations in ranking closeness across distribution pairs.

NAMMD is a natural and interpretable modification of MMD.
The normalization by RKHS norms is well justified empirically and theoretically.

The comparison theorem relating NAMMD and MMD test power is particularly insightful.

Weaknesses:

The advantage of NAMMD over MMD relies on certain norm conditions.
While discussed, the practical prevalence and interpretability of these conditions perhaps could be elaborated a bit further :)

---

> ### Author Rebuttal · Authors · 2026-03-31
>
> We sincerely thank the reviewer for the careful reading and constructive feedback. Below, we address the concerns in turn.
>
> >[W1] The advantage of NAMMD over MMD relies on certain norm conditions. While discussed, the practical prevalence and interpretability of these conditions perhaps could be elaborated a bit further :)
>
> [W1A] The norm condition is introduced for the setting where $\mathrm{MMD}(P_1,Q_1) <\mathrm{MMD}(P_2,Q_2)$, namely
> $$
> ||\mu_{P_1}||\_{\mathcal H_\kappa} + ||\mu_{Q_1}||\_{\mathcal H_\kappa}
> <
> ||\mu_{P_2}||\_{\mathcal H_\kappa} + ||\mu_{Q_2}||\_{\mathcal H_\kappa}.
> $$
> This condition is natural because the RKHS norms contribute directly to the MMD through
> $$
> MMD^2(P,Q)=||\mu_P||\_{\mathcal H_\kappa}^2+||\mu_Q||\_{\mathcal H_\kappa}^2-2\langle \mu_P,\mu_Q\rangle_{\mathcal H_\kappa}.
> $$
> In particular, when the inner-product terms are comparable across pairs, larger RKHS norms typically correspond to larger MMD values. We will further clarify the practical meaning and prevalence of this condition in the revision.
>
>
> >[Q1] Do the authors have any ideas toward a kernel selection criterion specifically optimized for comparing closeness across multiple distribution pairs, rather than a single pair as in TST?
>
> [Q1A] One possible direction is to select the kernel by optimizing an aggregate criterion over multiple candidate pairs, for example a weighted sum of NAMMD-based margins, so that the resulting kernel better preserves the relative ordering of closeness across pairs. However, the precise formulation of such a criterion, along with its finite-sample estimation and theoretical guarantees, remains open. We view this as an interesting direction for future work and will briefly discuss it in the revision.
>
>
> >[Q2] How sensitive is NAMMD to estimation error in RKHS norms, especially in small-sample or high-dimensional regimes?
>
> [Q2A] We further evaluate the sensitivity of NAMMD to RKHS norm estimation error. In this experiment, we fix the mean gap between the two Gaussian distributions and vary the variance to control the RKHS norm. The results suggest that NAMMD is not substantially more sensitive than MMD. Specifically, even with a small sample size of 10, the relative RMSE of NAMMD remains very close to that of MMD, where relative RMSE is the root mean squared estimation error divided by the corresponding population value.
>
> We report the relative RMSE below over 300 independent repetitions.
>
> | dimension | method | norm$^2$=0.2 | norm$^2$=0.4 | norm$^2$=0.6 | norm$^2$=0.8 |
> | --------- | ------ | -----------: | -----------: | -----------: | -----------: |
> | $d=2$     | MMD    |        7.462 |        2.311 |        1.172 |        0.633 |
> | $d=2$     | NAMMD  |        7.586 |        2.382 |        1.189 |        0.642 |
> |           |        |              |              |              |              |
> | $d=200$   | MMD    |        0.216 |        0.141 |        0.103 |        0.060 |
> | $d=200$   | NAMMD  |        0.217 |        0.142 |        0.105 |        0.061 |
> |           |        |              |              |              |              |
> | $d=20000$ | MMD    |        0.019 |        0.013 |        0.010 |        0.006 |
> | $d=20000$ | NAMMD  |        0.019 |        0.013 |        0.010 |        0.006 |
>
> As shown above, NAMMD and MMD have similar relative RMSE across all tested norm levels and dimensions. We will include these results in the revision.
>
> >[Q3] While the reference-pair strategy is intuitive, could the authors provide more systematic guidance or heuristics for selecting $\epsilon$ in applications where no obvious reference pair exists?
>
> [Q3A] When no obvious reference pair is available, one practical heuristic is to construct a synthetic pair by adding controlled perturbations (e.g., Gaussian noise) to one distribution. The resulting pairs provide a range of discrepancies that can be used to calibrate $\epsilon$. In practice, $\epsilon$ may be chosen as the largest perturbation level still regarded as negligible for the application. We will clarify this in the revision.

---

> > ### Author Rebuttal · Reviewer_d8VP · 2026-04-01
> >
> > My concerns have been adequately addressed.

---

> > > ### Author Response · Authors · 2026-04-01
> > >
> > > Dear Reviewer d8VP,
> > >
> > > It is glad to know that your concerns have been clarified.
> > >
> > > Many thanks for your firm support regarding our submission!
> > >
> > > Best,
> > >
> > > Authors of Submission13794

---

### Official Review · Reviewer_dMwS · 2026-03-04

**Soundness:** 3
**Presentation:** 3
**Significance:** 2
**Originality:** 2
**Overall Recommendation:** 4
**Confidence:** 3

**Summary:**

The paper proposes a maximum mean discrepancy (MMD)-based criterion for distribution closeness testing (DCT), namely norm-adaptive MMD (NAMMD), designed to handle complex data. The key idea is to rescale the MMD statistic using the RKHS norms of the distributions, which aims to improve robustness when MMD alone may fail to distinguish distributions. The effectiveness of NAMMD is evaluated through both theoretical analysis and empirical experiments.

**Compliance With Llm Reviewing Policy:**

Affirmed.

**Final Justification:**

Most of my concerns have been addressed, and the idea of NAMMD is interesting to me. However, some clarifications and explanations provided in the rebuttal are needed to be added in the revision. Overall, I remain positive about the paper.

**Key Questions For Authors:**

1. The paper mentions that “NAMMD-based DCT performs better than MMD-based DCT with the optimal kernel” (Lines 321–329, Page 6). However, the paper does not appear to provide supporting theoretical or empirical evidence for this claim. Could the authors clarify what is meant by “optimal kernel” in this context and provide justification for this statement?
2. How sensitive is NAMMD to the choice of kernel type and kernel bandwidth? It is well known that the performance of MMD can be sensitive to the kernel bandwidth. Does NAMMD exhibit similar sensitivity?
3. Is NAMMD always expected to outperform MMD in distribution closeness testing, or are there scenarios where MMD may still be preferable?

**Limitations:**

Yes, the authors included the limitation discussions in the appendix.

**Strengths And Weaknesses:**

- Strengths:
1. The paper provides a solid theoretical analysis of the proposed statistic. In particular, when using the same kernel, NAMMD retains the desirable properties of MMD in cases where MMD performs well, while potentially alleviating some failure cases of MMD. These properties are supported by both theoretical results and experimental evaluations, which makes the idea interesting.
2. The paper is generally well-written and clearly presented. The motivation of the work and the definition of NAMMD are easy to follow, and the structure of the paper is clear.


- Weaknesses:
1. The proposed method appears to require either a predefined threshold or a reference distribution pair to determine the decision threshold of DCT. As a result, the interpretation of the test seems to depend on comparisons with reference distributions rather than an intrinsic threshold derived from the statistic itself. It would be helpful if the authors could clarify whether this reliance on reference pairs is standard practice in DCT, how this pair is selected and how sensitive the method is to the choice of such reference pairs.
2. The paper focuses on MMD-based discrepancy measures. While MMD is widely used for sample-based distribution comparison in complex data settings, it is not the only possible choice. Other integral probability metrics or divergences (e.g., Wasserstein distance or energy distance) may also be used for distribution comparison. It would strengthen the paper if the authors could discuss why the proposed normalization is specifically designed for MMD and whether similar ideas could be extended to other discrepancy measures.
3. Some of the baselines used in the experiments (e.g., MMDFuse) were originally developed for two-sample testing. Although these methods produce discrepancy statistics similar to those used in DCT, it would be helpful if the authors could clarify how such two-sample testing methods are adapted or interpreted in the DCT setting.

---

> ### Author Rebuttal · Authors · 2026-03-31
>
> We sincerely thank the reviewer for the careful reading and constructive feedback. Below, we address the concerns in turn.
>
> >[W1] predefined threshold and reference pair
>
> [W1A] In DCT, the threshold $ϵ$ is application-dependent, rather than determined only by the statistic. To the best of our knowledge, prior DCT works mostly treat $ϵ$ as user-specified and do not discuss in detail how to choose it in practice. In our paper, the reference pair is only an intuitive and practical way to choose $ϵ$.
>
> In practice, one can choose a reference pair that is already considered acceptably close for the task. For example, although ImageNet and Pascal VOC are not identical, a model trained on ImageNet can still perform well on Pascal VOC. In this case, the NAMMD value between these two distributions gives a meaningful tolerance level $ϵ$.
>
> To clarify this point, we provide three examples in Section 5.2. In each case, the reference pair reflects a meaningful task-specific tolerance level. Specifically, we use downstream accuracy for ImageNet and its variants, confidence margin for ImageNet versus ImageNetv2 when labels are limited, and perturbation severity for adversarial CIFAR10. In all cases, NAMMD-based DCT correctly reflects the relative closeness of different distribution pairs.
>
> >[W2] why MMD and extension
>
> [W2A] Our focus on MMD is mainly motivated by the DCT setting, where a tractable asymptotic null distribution is needed to construct testing thresholds. In DCT, the null hypothesis allows $P \neq Q$ as long as their discrepancy is below $ϵ$. Therefore, the permutation calibration used in TST no longer applies, because under the DCT null the two distributions are not required to be equal, so the samples are generally not exchangeable.
>
> From this perspective, other discrepancy measures with tractable asymptotic null distributions can also be used for DCT. Examples include sliced Wasserstein distance [1] and energy distance, which is a special case of MMD with a distance-induced kernel [2].
>
> Our adjustment for MMD uses the RKHS norm, with
> $$
> ||μ_P||\_{H_κ}^2=K-Var(P;κ).
> $$
> This suggests adjusting the discrepancy by the concentration of the distributions in RKHS. Extending this idea to other discrepancies would require an analogous notion of concentration. We will clarify this point in the revision.
>
> [1] Rodríguez-Vítores et al. (2025). An improved central limit theorem for the empirical sliced Wasserstein distance. arXiv.
>
> [2] Sejdinovic et al. (2013). Equivalence of distance-based and RKHS-based statistics in hypothesis testing. Annals of Statistics.
>
> >[W3] TST methods in the DCT
>
> [W3A] As discussed in [W2A], the key issue is not whether a method was originally proposed for TST, but whether its discrepancy statistic admits valid calibration under the DCT null $H_0:d(P,Q)\leqϵ$. If so, then the method can be adapted to DCT by following the same procedure as in our paper. We will clarify this point in the revision.
>
> >[Q1] clarify about “optimal kernel”
>
> [Q1A] Thank you for pointing this out. In Theorem 9, we show that, under a certain norm condition, NAMMD has higher power than MMD when both use the same kernel $κ$. By “optimal kernel,” we mean a kernel that maximizes DCT power. Let $κ\_{MMD}$ denote an optimal kernel for MMD. Then NAMMD with $κ\_{MMD}$ already outperforms MMD with $κ\_{MMD}$. Hence, if $κ\_{NAMMD}$ is optimal for NAMMD, NAMMD with $κ\_{NAMMD}$ must also outperform MMD with its optimal kernel. We agree that this term was not formally defined in the current draft and will revise it for precision.
>
> >[Q2] kernel type and bandwidth
>
> [Q2A] Table 15 (Appendix D.5) compares NAMMD and MMD under several kernels, including Gaussian, Laplace, deep, and Mahalanobis. NAMMD remains competitive and often outperforms MMD under the same kernel.
>
> For bandwidth, NAMMD shows sensitivity similar to MMD. In DCT on BLOB, we vary the Gaussian kernel bandwidth over $0.1h_0, 0.3h_0, 0.5h_0, h_0, 2h_0$, where $h_0$ is the median-heuristic bandwidth, and report the empirical power over 1000 repetitions below.
>
> |power|$.1h_0$|$.3h_0$|$.5h_0$|$h_0$|$2.0h_0$|
> |-|-|-|-|-|-|
> |MMD|$.792\pm.013$|$.900\pm.009$|$.772\pm.013$|$.330\pm.015$|$.037\pm.006$|
> |NAMMD|$.800\pm.013$|$.924\pm.008$|$.858\pm.011$|$.519\pm.016$|$.076\pm.008$|
>
> Both methods are sensitive to bandwidth: power drops when it is too small or too large. We therefore follow prior work and use a data-adaptive bandwidth selection strategy (Appendix B.5). We will add this discussion and the results in the revision.
>
> >[Q3] scenarios where MMD may still be preferable
>
> [Q3A] NAMMD is not always expected to outperform MMD. Definition 8 considers a setting where the limitation of MMD does not arise. Given this, Theorem 9 shows that NAMMD still has higher power than MMD under a certain norm condition. When this condition does not hold, the advantage is no longer guaranteed, and MMD may still be preferable. We will clarify this point in the revision.

---

> > ### Author Rebuttal · Reviewer_dMwS · 2026-04-02
> >
> > Most of my concerns have been addressed, and I remain positive about the paper. However, some clarifications and explanations discussed in the rebuttal should still be incorporated.
> >
> > Minor: In “[Q2A] Table 15 (Appendix D.5),” I could not find Table 15. Did you mean Table 13?

---

> > > ### Author Response · Authors · 2026-04-02
> > >
> > > Dear Reviewer dMwS,
> > >
> > > It is glad to know that your concerns have been addressed. We will incorporate the clarifications and explanations discussed in the rebuttal into the revised paper. We also sincerely thank you for your positive assessment and support.
> > >
> > > You are correct that we meant Table 13, not Table 15. We apologize for this error.
> > >
> > > Best,
> > >
> > > Authors of Submission13794

---

### Official Review · Reviewer_2u62 · 2026-03-05

**Soundness:** 2
**Presentation:** 2
**Significance:** 2
**Originality:** 3
**Overall Recommendation:** 3
**Confidence:** 5

**Summary:**

This paper introduces a ''norm-adaptive maximum mean discrepancy'' measure. For a bounded kernel $\kappa \le K$, the latter reads
$$\frac{\|\mu_P -\mu_Q\|^2_{\mathcal{H}_ \kappa}}{ 4K - \|\mu_P \|^2_{\mathcal{H}_ \kappa}- \|\mu_Q\|^2_{\mathcal{H}_\kappa} },$$
where $\mu_P$ is the kernel mean embedding of the measure $P$. The authors introduce this measure with the aim of performing ``tolerant testing'', i.e., the null hypothesis is that two distributions do not diverge by more than $\epsilon$ for a specified metric. The paper proposes both theoretical and numerical results suggesting that this discrepancy is helpful in a variety of cases.

**Compliance With Llm Reviewing Policy:**

Affirmed.

**Final Justification:**

The discussion section revealed how important the understanding gaps are, culminating in the authors first saying that power of the test is not important and then acknowledging that a more powerful test is an important goal.

The authors acknowledged that their motivating figure might be mathematically flawed and tried to save it by appealing to intuition.


A complete overhaul appears necessary. The authors should streamline the message and clearly state one or two hypotheses that they want to test. Either one tests for a fixed threshold, or one compares two distances. These two problems are related but not the same. The authors should be careful about what is random and what not.
They should avoid creating strawmen; the best example being their claimed "issue" that many pairs of distributions are at the same distance of one another.

The introduction of NAMMD is interesting per se. This object should be studied carefully, as it is not a metric anymore.   A statistical key finding is that their normalisation can increase the power. It is a clean statistical statement that deserves to be written down in a concise, rigorous paper.

When writing such a paper, it is also important to acknowledge that testing for differences between distribution has a long history. If people solved the one-dimensional problem 30 years, it is a good indication that the citing papers focussed on more than one dimension in recent years.

**Key Questions For Authors:**

1) To me, the writing is statistically inaccurate and the problem that the authors raise appears not to be an actual one. It is clear that for any metric space, many points can be at the same distance of a given reference point. There is no notion of informativeness in MMD; this is a distance (if the kernel is characteristic) and thus only tells   On line 86 you speak of p-values of an estimator, even though p-values are related to tests. In your case, $\mathbb{P}$ and $\mathbb{Q}$ are fixed, nonrandom  distributions (as you use them to define the hypotheses), either they satisfy the null or they don't. The limitations that you aim at showing are thus very confusing to me.
You aim at driving the point home with Figure 1 but the latter speaks of p-values of estimators. Also, it seems off to look at TST to derive conclusions about DCT.
2) In finite sample experiments you show that your test has a better power than Cannone's.  Picking a single competitor is quite weak evidence. Also, you carry these tests on discrete distributions, the precise setting you introduce as too restrictive in the Introduction.
3) In Section 4, you rely on auxiliary measures to find a suitable parameter epsilon. In practice, these measures will be replaced by empirical measures and the variability associated to that should be discussed.
4) Overall, I think that the message of the paper should be clearer. I have the impression that what you are trying to say is that the MMD test statistics might have a high variance, which impacts the power of tests and that your scaling helps in that matter. If this is your conclusion, you should streamline the paper. Also, using Slutsky's theorem as you do might be too rough an analysis to show what you aim at showing.

**Limitations:**

yes

**Strengths And Weaknesses:**

Soundness: The paper is reasonable.The proofs look correct but require greater care in the formulation of the statements. The numerical results seem promising but their scope is limited.

Presentation: At first glance the presentation looks ok. Yet, if one dives deeper, there are many issues. A central element of the paper should be the presentation of the testing procedure for NAMMD. There, $\zeta_1 $ and $\zeta_2$ play an important role and should be introduced more carefully, describing that they require estimation. The $\approx$ sign in the derivation of these quantities is not acceptable.
The paper states that few methods are available for continuous data omitting, thereby omitting completely optimal transport based approaches, yet they use such methods in Appendix E.2.
Lemma~12 does not discuss the assumptions needed to apply the results by Serfling.
In the numerical section, the authors introduce a variant of their method, which they describe so briefly in Section D.5 that it is unclear why it is needed and what it brings. Is this change made because the usual NAMMD wasn't working well enough?

The details of the experiments should be presented more clearly. I looked after the number of replications for Section E.4 and could not find them. The graph should also present the 0.05 line to help compare. There are still many points for which the (pointwise?) confidence interval goes above 5%.  Also your results are asymptotic and you apply your test to sample sizes around 2000 on Figure 3, why don't you cover that range? Could it be that you have a better rejection power because the test is not well-calibrated? A better presentation of the results and experiments would make sure that the reader can't doubt.

Significance: I am not completely convinced by the actual problem that the authors seem to identify and tackle. Figure 1 is central to showing that there is a problem. Appendix D.1.1 reads as if they were using a sample size of 4 to produce that figure, casting critical shadows on the claim.

Originality: To the best of my knowledge, the paper is original.

---

> ### Author Rebuttal · Authors · 2026-03-31
>
> We thank the reviewer for careful reading and helpful comments. We respond below and kindly refer reviewer to the anonymous supplementary for additional results:
> https://anonymous.4open.science/r/ICML2026-Submission13794-700E/ICML2026-Submission13794.md
>
> >[W1] $ζ_1$ and $ζ_2$
>
> [W1A] The notation for $ζ_1$ and $ζ_2$ should be clarified. In Eq. (2), they should be $\hat ζ_1$ and $\hat ζ_2$, since Eq. (2) is an empirical estimator rather than a population quantity. We will revise this.
>
> This issue comes from a notation error in the appendix, where we used same symbols for both population quantities and their empirical estimators, together with $\approx$. We will revise this.
>
> >[W2] continuous data and optimal transport
>
> [W2A] In introduction, our point was not that there are few methods for continuous distributions in general, but that this is the case in DCT. In Section 2 and Section B.5, we discuss discrepancy measures for continuous distributions, including analogues of total variation, KL divergence, and IPMs such as Wasserstein-1, an optimal-transport-based distance, etc. In Appendix E.2, we use Wasserstein distance only for comparison in DCT. To our knowledge, Wasserstein-based methods have not been systematically developed for DCT. We will revise wording and dicuss optimal transport.
>
> >[W3] assumptions of Lemma 12
>
> [W3A] Assumptions of Serfling’s result are: (i) $E[h^2] < \infty$, and (ii) $ζ_1 > 0$. In our setting, the first holds because kernel is bounded, and the second holds under $P\neq Q$. We will add these assumptions to Lemma 12.
>
> >[W4] NAMMDFuse
>
> [W4A] In our main experiments, NAMMD already performs competitively. We include NAMMDFuse not because NAMMD performs poorly, but to show that NAMMD is compatible with existing kernel selection methods, e.g., multi-kernel fusion framework of Biggs et al. (2023).
>
> >[W5] Type-I error
>
> [W5A] Each point is based on 1000 independent repetitions. Some pointwise confidence intervals slightly exceeding $.050$ are expected from finite-sample variability and do not indicate systematic miscalibration.
>
> We also conducted additional experiments with a wider range of sample sizes; see Tables 1-4 in the anonymous supplementary. We will include them in revision, together with the $.050$ reference line.
>
> >[W6] Figure 1 and sample size
>
> [W6A] When the MMD value is fixed at $.150$, the issue in Figure 1 already appears at sample size $4$ in dimension 2. We agree, however, that sample size $4$ alone is not fully convincing. We therefore consider same setting with MMD $.050$ and sample size $50$; see Table 5 in the anonymous supplementary. The same trend holds: when RKHS norm increases with fixed MMD, MMD-based TST $p$-value decreases.
>
> >[Q1] informativeness and Figure 1
>
> [Q1A] Here, “informativeness” is task-specific: in DCT, we want metric to better reflect underlying closeness levels of differnet distribuiton pairs, namely how easily two distributions can be distinguished, for which TST $p$-value provides one indicator. In theory, $ϵ$, $P$, and $Q$ are fixed. In practice, however, $ϵ$ is often chosen from a reference pair, so the task becomes comparing closeness of test and reference pairs.
>
> We also show the same phenomenon directly in DCT. Specifically, we fix $ϵ=.050$, $MMD(P,Q)=.100$, and sample size $50$, where $P$ and $Q$ are 2-d Gaussian distributions; see Table 6 in the anonymous supplementary. The results below show the same trend: when the RKHS norm increases with fixed MMD, MMD-based DCT $p$-value decreases.
>
> >[Q2] comparison on discrete distributions
>
> [Q2A] We include Canonne’s method because it is a representative DCT baseline for discrete distributions. This provides a direct comparison with an existing DCT method in traditional setting. We also include continuous experiments, where NAMMD-based DCT compares favorably with MMD-based DCT. Our point in Introduction was that many DCT methods are developed for discrete settings and may be less applicable to continuous scenarios.
>
> >[Q3] reference-pair variability
>
> [Q3A] In practice, the auxiliary measure used to choose $ϵ$ would be replaced by its empirical counterpart, which introduces additional variability. We will clarify this point in revision. In Section 5.2, we show that a stable reference pair is often available. We use simple application-specific quantities to choose reference pair. In each case, NAMMD-based DCT correctly reflects relative closeness between reference and test pairs.
>
> >[Q4] message and theory
>
> [Q4A] As discussed in [W6A] and [Q1A], Figure 1 should be revised. Our main point is that, under both TST and DCT, same MMD value can correspond to different distribution pairs and levels of statistical distinguishability, as reflected by $p$-values. The variance or power improvement from incorporating RKHS norm is a consequence, not the main message. Regarding theory, we also provide support through Lemma 4, Lemma 5, and Theorem 7 without Slutsky's theorem, together with empirical evidence.

---

> > ### Author Rebuttal · Reviewer_2u62 · 2026-04-03
> >
> > About optimal transport and equivalent distribution testing:
> > One of the very first papers on optimal transport (in dimension 1 though) was already doing "significant difference testing" (Nonparametric validation of similar distributions, Munk & Czado, 1998).
> > There are many instances extremely close to what you consider. It is sometimes called "tolerant testing", "equivalent testing".  Further, what you consider is first and foremost a statistical problem.  It is worrisome to see that none of the four references in the literature review (l.38 second column) come from a statistical journal.
> >
> >
> > Your answer to Q1:
> > It confirms what I fear, you do not seem to have a clear idea of what you are doing. The sentence "we want metric to better reflect underlying closeness levels of differet distribuiton pairs" is completely flawed. There is no better or worse closeness. A metric provides a number for each pair of distributions. If you change the metric you change the meaning of closeness! "How easily two distributions can be distinguished" means comparing the power of test in the problem of testing P=Q, which is coherent with the TST part after but not with you claiming that you think about DCT. Indeed, the first word of that sentence is "In DCT, ...".
> >
> > A p-value is a random variable. It depends on the metric you choose and the sample size. Per se, it does not indicate closeness of two distributions. Even further, you say that you focus on DCT but the sentence  "In practice, however, $ϵ$ is often chosen from a reference pair, so the task becomes comparing closeness of test and reference pairs." means that the problem you describe is an ordering problem of distances.
> >
> > The latter problem seems to be what you actually want to tackle given the context provided.
> >
> > You provide Table 6. In there, the p-values are similar for both MMD and NAMMD. If you are interested in a testing problem you are not showing that the test gives better power. The only point that can be made is that NAMMD increases (very slightly thought) with the RKHS norm.
> >
> > About Q4: Again your answer is flawed. For a fixed metric, it is expected that the power curves of test will be different for different pairs of distributions, even if they are at the same distance.
> >
> > Your paper is about distribution shifts. You rightfully claim that what matters in that context is not the distributions being equal as small differences might not be harmful to the application. We conclude that the problem is to **identify whether a pair of test and training distributions are close enough not to cause issues in a downstream task**.
> >
> > The main objective of the paper should then be comparing two distance conditionally on the fact that the reference pairs of measures $\mathbb{P}_1,\mathbb{Q}_1$ is such that $\mathbb{Q}_1$ displayed a good performance on a task where the training was done on $\mathbb{P}_1$.
> >
> >
> > Additional questions:
> > 1) If testing is the objective, what is the interest of NAMMD? The additional tables with p-values that you provided in this review round don't show major differences in p-values.
> > If you want to distinguish something statistically, you want a high power. A high power is not a mere side-effect.
> > 2) To show that NAMMD is the actual quantity to consider, you need to show that a distribution close to another one in the NAMMD distance is going to deliver what you call a good performance for the downstream task of interest.

---

> > > ### Author Response · Authors · 2026-04-06
> > >
> > > We thank reviewer for the new comments. It is clear that the reviewer only has concerns regarding Q1A and Q4A in previous rebuttal, and suggests that we cite references from Statistics in the Introduction.
> > >
> > > Based on the comments, we now understand the huge divergence in scores (5,4,4,2). This seems to come from different writing, saying, reading styles. We will explain this first and clarify that there are **no flaws** in our paper.
> > >
> > > >re: Q1A and a flaw in Figure 1
> > >
> > > It **was** not clear why Figure 1 would cause so many confusions when we first read the comments (as other reviewers like it, scores are 5,4,4). Now, we get a clearer reason: reviewer may expect Figure 1 to directly provide a mathematically grounded design objective, but that is not the purpose of Figure 1.
> > >
> > > Figure 1 is intended to convey **just one message**: even when MMD is fixed, distinguishability (reflected by p-values) can differ across distribution pairs. This empirical observation **intuitively** motivates us to seek a statistic whose value increases as the p-value decreases (that's it, no more thinking regarding Figure 1). NAMMD is proposed from this **intuitive motivation**, not from a mathematically grounded one. In machine learning community, it is common to begin from intuition and then justify the design through theory or experiments, as we do in this paper.
> > >
> > > If we view Figure 1 as a mathematically-grounded motivation, it might contain flaws. **However, if we view Figure 1 as an intuitive motivation, there are no flaws.** The related concern instead reflects a difference in how the purpose of Figure 1 is understood. **All other reviewers also agree that the presentation and motivation are very clear,** e.g., *"provides an intuitive explanation of MMD’s limitations in ranking closeness across distribution pairs"* (by reviewer d8VP).
> > >
> > > There could be another presentation style, perhaps as you suggest: *"MMD might have a high variance, which impacts the power of tests and that your scaling helps in that matter."* However, it might introduce more prior knowledge for general machine learning audience at the outset than Figure 1 does.
> > >
> > > Yet, we do agree that your statistical perspective is valuable, and we will include it to indicate that there are other ways to make the discussion more statistically grounded. We will also clarify that the paper is written from a machine learning perspective and set the boundary of its claims. We do not want to overclaim or create noise in the field. Please let us know your thoughts in the final comment.
> > >
> > > We also hope you can **reconsider the score**. A score of 2 suggests technical flaws, weak evaluation, inadequate reproducibility, or writing so poor that the key claims cannot be understood. We do not believe this applies to our paper. There are no technical flaws, and the evaluation is strong.
> > >
> > > >re: flaw in Q4A
> > >
> > > This concern also relates to Figure 1, whose purpose we clarified above.
> > >
> > > >re: statistical reference
> > >
> > > We agree that this paper is an important prior work. Yet, it is developed for **1D continuous distributions**, relying on inverse CDFs, quantile-process asymptotics, and order statistics, and thus does not directly address the **high-dimensional complex-data setting** in our paper. Our contribution is to extend DCT to complex data through kernel discrepancies. We also cite related work from the tolerant/equivalent testing literature, e.g., **The Price of Tolerance in Distribution Testing**. We will clarify these connections explicitly in revision.
> > >
> > > >re: distribution shift
> > >
> > > Our paper is indeed a testing paper, and distribution-shift situation is only a practical showcase of DCT, which is itself a contribution since this application to machine learning scenarios has not been discussed before.
> > >
> > > >re: additional question 1
> > >
> > > We agree that, by introducing **NAMMD**, an important goal is to obtain a **more powerful DCT**. In **Table 6** (simple 2D Gaussian), this advantage not appear very large. In this toy setting (used for motivation in Figure 1, **not the main experiment**), the scale of **NAMMD** is small, so its numerical variation is also small; the key point is that the variation is **consistent with the p-value trend**, which reflects changing **statistical distinguishability** even when **MMD** is fixed. The advantage of **NAMMD** becomes much clearer in later experiments on higher-dimensional settings, e.g., **CIFAR10** and **ImageNet**. This is supported by both experiments and our **theoretical analysis**.
> > >
> > > >re: additional question 2
> > >
> > > We provide **three representative applications** in Section 5.2, each using a reference pair to encode a downstream-relevant tolerance level. Concretely, we use accuracy margin for ImageNet and its variants, confidence margin for classes of ImageNet versus ImageNetv2, and perturbation level for adversarial CIFAR10. In each case, NAMMD-based DCT correctly reflects the relative closeness among different distribution pairs and performs better than MMD.

---

### Official Review · Reviewer_RCqX · 2026-03-12

**Soundness:** 3
**Presentation:** 4
**Significance:** 4
**Originality:** 3
**Overall Recommendation:** 4
**Confidence:** 5

**Summary:**

This submission deals with testing the closeness of two distributions. The authors proposed a norm-adaptive maximum mean discrepancy (NAMMD) for distribution closeness test statistics (Eq. (1)). The research problem was motivated by observing that the existing approach, MMD, may lead to different p-values with the same MMD value from different pairs of distributions, which indicates that the closeness of different pairs can be different. Applying RKHS norms to MMD to form a new NAMMD statistic that better measures the closeness between two distributions (datasets). The asymptotics of NAMMD under the null hypothesis are established, the distribution closeness test (DCT) statistics are derived, and Type 1 errors are controlled. Numerical experiments are conducted to validate NAMMD's advantages and demonstrate its practical effectiveness across various data scenarios. While NAMMD shows promising performance, its effectiveness may vary depending on the data structure and sample size.

**Compliance With Llm Reviewing Policy:**

Affirmed.

**Key Questions For Authors:**

I see that two populations have the same sample size, which can be restricted. I'd like to know whether the authors can develop and apply subsampling methods from the larger population, perform many NAMMD DCTs, and report the final test results.

i.i.d. samples have been assumed; I'd like to know whether the proposed method can be extended to non-i.i.d. samples.

Do the orders of components in each population matter? I.e., if I randomly shuffle component orders in one of two populations, will the test results be significantly changed?

**Limitations:**

I see that two populations have the same sample size, which can be restricted. I'd like to know whether the authors can develop and apply subsampling methods from the larger population, perform many NAMMD DCTs, and report the final test results.

i.i.d. samples have been assumed, which limits its applications.

**Strengths And Weaknesses:**

Soundness. Figure 1 clearly shows that the proposed NAMMD DCT outperforms the benchmark. Lemma 2 (asymptotics), Lemmas 4-5 (consistency), and Theorem 9 (higher testing power) are established to justify the proposed DCT statistic.

Presentation. The paper is clearly written.

Significance. TST has been a phenomenon in the literature; DCT may be more practically relevant, as the identity can be easily rejected in most scenarios.

Originality. The existing method, MMD, is efficient at testing whether two samples are from the same population (distribution), but has limitations for assessing the closeness of the two distributions when they are not identical, which is more common in practice. NAMMD uses kernel functions to adjust the MMD in Eq. (1), thereby enabling DCT within the MMD framework.

Weaknesses: I see that two populations have the same sample size, which can be restricted. I'd like to know whether the authors can develop and apply subsampling methods from the larger population, perform many NAMMD DCTs, and report the final test results.

---

> ### Author Rebuttal · Authors · 2026-03-31
>
> We sincerely thank the reviewer for the careful reading and constructive feedback. Below, we address the concerns in turn.
>
> >[W1+Q1]  I see that two populations have the same sample size, which can be restricted. I'd like to know whether the authors can develop and apply subsampling methods from the larger population, perform many NAMMD DCTs, and report the final test results.
>
> [W1Q1A] The proposed method is not restricted to the equal-sample setting. As discussed in Appendix B.2, it extends naturally to the unequal-sample case, with similar asymptotic properties and the same testing procedure. To make this clearer, we will add a remark in the revision explicitly stating that our method is not limited to the equal-sample setting.
>
> We agree that repeatedly subsampling from the larger population, conducting multiple NAMMD-based DCTs, and aggregating the results is a reasonable practical strategy. However, this would no longer be a single test and would require an additional multiple-testing correction or p-value combination step to ensure valid overall Type I error control.
>
>
> >[Q2] i.i.d. samples have been assumed; I'd like to know whether the proposed method can be extended to non-i.i.d. samples.
>
> [Q2A] Our current theory is developed under the i.i.d. assumption and do not directly extend to non-i.i.d. data. Prior MMD work has extended kernel two-sample testing to dependent data through dependence-aware calibration methods such as the wild bootstrap [1]. We believe NAMMD could potentially be extended in a similar direction, but this would require new theory and is beyond the scope of the current paper. We will clarify this limitation in the revision.
>
> [1] Chwialkowski, K., Sejdinovic, D., & Gretton, A. (2014). A wild bootstrap for degenerate kernel tests. NeurIPS.
>
> >[Q3] Do the orders of components in each population matter? I.e., if I randomly shuffle component orders in one of two populations, will the test results be significantly changed?
>
> [A3] The proposed method is permutation-invariant with respect to the sample ordering within each population. In the equal-sample setting,
> $$\widehat{\mathrm{NAMMD}}(X,Y;\kappa)=\frac{\sum_{i\ne j} H_{i,j}}{\sum_{i\ne j}\bigl[4K-\kappa(x_i,x_j)-\kappa(y_i,y_j)\bigr]},
> $$
> where
> $$
> H_{i,j}=\kappa(x_i,x_j)+\kappa(y_i,y_j)-\kappa(x_i,y_j)-\kappa(y_i,x_j).
> $$
> Since both the numerator and denominator are symmetric sums over all sample pairs, the statistic depends only on the sample sets, not their ordering. Hence, randomly shuffling the samples within either population does not change the NAMMD value or the test result.

---

> > ### Author Rebuttal · Reviewer_RCqX · 2026-04-03
> >
> > As the iid assumption as a limitation remains, my given score reflects my assessment.

---

> > > ### Author Response · Authors · 2026-04-03
> > >
> > > Dear Reviewer RCqX,
> > >
> > > Thank you for the follow-up. We are very glad to know that your concerns have been addressed.
> > >
> > > We would like to briefly clarify one point regarding the i.i.d. assumption. In hypothesis testing, the i.i.d. setting is a standard and natural starting point, especially for developing new theory. Since this work is, to the best of our knowledge, the first kernel-based DCT motivated by continuous distributions, we believe it is appropriate to first establish the method and theory in the i.i.d. setting. As noted in the rebuttal, we will discuss the potential extension to the non-i.i.d. case as future work and incorporate this discussion into the revision.
> > >
> > > Best,
> > >
> > > Authors of Submission 13794

---

### Decision · Program_Chairs · 2026-04-30

**Decision:**

Accept (regular)

**Comment:**

This paper studies distribution closeness testing for complex data and proposes NAMMD for this setting. The reviewers agreed that the paper addresses an interesting problem and makes a meaningful technical contribution. Several reviewers were positive about both the theoretical development and the empirical evaluation. The rebuttal also clarified a number of concrete issues.
At the same time, some concerns remained after the discussion. These centered on the paper’s framing and statistical interpretation. In particular, the reviewers did not fully agree on whether the target problem, the motivation, and the main claims are stated with sufficient clarity and precision in the current version. Even so, most reviewers remained positive and viewed these issues as revisable.
For these reasons, I recommend weak acceptance. The paper has clear merit, and the remaining concerns appear addressable in revision. I encourage the authors to sharpen the framing, clarify the statistical interpretation, and incorporate the rebuttal clarifications into the final version.